# A Generalized Geometric Theoretical Framework of Centroid Discriminant Analysis for Linear Classification of Multi-dimensional Data

**Yue Wu**[1,2]    **Jialin Zhao**[1,3]    **Carlo Vittorio Cannistraci**[1,2,3,*]
[1] Center for Complex Network Intelligence (CCNI) [†]
[2] Department of Biomedical Engineering, Tsinghua University
[3] Department of Computer Science & Technology, Tsinghua University

## Abstract

With the advent of the neural network era, traditional machine learning methods have increasingly been overshadowed. Nevertheless, continuing research on the role of geometry for learning in data science is crucial to envision and understand new principles behind the design of efficient machine learning. Linear classifiers are favored in certain tasks due to their reduced susceptibility to overfitting and their ability to provide interpretable decision boundaries. However, achieving both scalability and high predictive performance in linear classification remains a persistent challenge. Here, we propose a theoretical framework named geometric discriminant analysis (GDA). GDA includes the family of linear classifiers that can be expressed as a function of a centroid discriminant basis (CDB0) - the connection line between two centroids - adjusted by geometric corrections under different constraints. We demonstrate that linear discriminant analysis (LDA) is a subcase of the GDA theoretical framework, and we show its convergence to CDB0 under certain conditions. Then, based on the GDA framework, we propose an efficient linear classifier named centroid discriminant analysis (CDA) which is defined as a special case of GDA under a 2D plane geometric constraint. CDA training is initialized starting from CDB0 and involves the iterative calculation of new adjusted centroid discriminant lines whose optimal rotations on the associated 2D planes are searched via Bayesian optimization. CDA has good scalability (quadratic time complexity) which is lower than LDA and support vector machine (SVM) (cubic complexity). Results on 27 real datasets across classification tasks of standard images, medical images and chemical properties, offer empirical evidence that CDA outperforms other linear methods such as LDA, SVM and logistic regression (LR) in terms of scalability, performance and stability. Furthermore, we show that linear CDA can be generalized to nonlinear CDA via kernel method, demonstrating improvements on the linear version with tests on three challenging datasets of images and chemical data. GDA's general validity as a new theoretical framework may inspire the design of new classifiers under the definition of different geometric constraints, paving the way towards a deeper understanding of the role of geometry in learning from data.

## 1 Introduction

Linear classifiers are often favored over nonlinear models, such as neural networks, for certain tasks due to their comparable performance in high-dimensional data spaces, faster training speeds, reduced tendency to overfit, and greater interpretability in decision-making (Varoquaux et al. (2017); Yuan et al. (2012)). Notably, linear classifiers have demonstrated performance on par with convolutional

---

[*]Corresponding author: kalokagathos.agon@gmail.com
[†]Research Center in Tsinghua Laboratory of Brain and Intelligence (THBI), Department of Psychological and Cognitive Sciences, Tsinghua University

neural networks (CNNs) in medical classification tasks, such as predicting Alzheimer's disease from structural or functional brain MRI images (Schulz et al. (2020); Varoquaux & Cheplygina (2022)).

These linear classifiers can be categorized into several types based on the principles they use to define the decision boundary or classification discriminant, as described below, where $N$ is the number of samples, $M$ is the number of features and $k$ denotes an iteration term:

- The minimum distance classifier (MDC) (Duda et al. (2001)) which is a prototype-based classifier that assigns points according to the perpendicular bisector boundary between the centroids of two groups. This classifier has $O(NM)$ training time complexity.
- Fisher's linear discriminant analysis (LDA, specifically refer to Fisher's LDA in this study) is a variance-based classifier which can be trained in cubic time complexity $O(NM^2 + M^3)$. While faster implementations like spectral regression discriminant analysis (SRDA) (Cai et al. (2008)) claim lower training time complexity, their efficiency depends on specific conditions, such as a sufficiently small iterative term and sparsity in the data. These constraints limit SRDA's applicability in real-world classification tasks.
- Support vector machine (SVM) (Cortes & Vapnik (1995)) with a linear kernel is a maximum-margin classifier, which has a training time complexity of $O(N^3)$. Fast implementations include Liblinear (referred to as fast SVM) and SVM-SGD, which use coordinate descent and stochastic gradients respectively, achieving quasi-quadratic time complexity $O(kNM)$.
- Perceptron (Minsky & Papert (1969)) is a misclassification-triggered ruled-based classifier. Its training time complexity is $O(kNM)$.
- Logistic regression (LR) (Panda et al. (2022)) is a statistics-based classifier. It can be trained using either maximum likelihood estimation (MLE) or iteratively reweighted least squares, with time complexity of $O(NM^2 + M^3)$ and $O(NM^2)$ respectively, and with $O(kNM)$ using the same coordinate descent technique in Liblinear.

Among linear classifiers, MDC offers the lowest training time complexity but suffers from limited performance due to its overly simplified decision boundary. Widely used methods such as LDA and SVM are often favored for their strong predictive capabilities. However, these methods can be computationally expensive, particularly for large-scale datasets. Hence, achieving both high scalability and strong predictive performance remains a challenging tradeoff, highlighting the need for new approaches that balance these competing demands.

To address this challenge, this paper makes the following **3 key contributions**:

- **A geometric theoretical framework for classifiers:** (See Appendix. A.1 for explanation of the term theoretical framework.) This study introduces a geometric framework, geometric discriminant analysis (GDA), to unify certain linear classifiers under a common theoretical model. GDA leverages a special type of centroid discriminant basis (CDB0), a vector connecting the centroids of two classes, which serves as the foundation for constructing classifier decision boundaries. The GDA framework adjusts the CDB0 through geometric corrections under various constraints, enabling the derivation of classifiers with desirable properties. Notably, we show that: MDC is a special case of GDA, where geometric corrections are not applied to CDB0; linear discriminant analysis (LDA) is a special case of GDA, where the CDB0 is corrected by maximizing the projection variance ratio.

- **A high-performance and scalable linear geometric classifier:** Building upon the GDA framework, we propose centroid discriminant analysis (CDA), a novel geometric classifier that iteratively adjusts the CDB through performance-dependent rotations on 2D planes. These rotations are optimized via Bayesian optimization, enhancing the decision boundary's adaptability efficiently. CDA achieves lower training time complexity (quadratic) and is more efficient than LDA and SVM. Experimental evaluations on 27 real-world datasets of standard images, medical images and chemical property data, reveal that CDA consistently outperforms LDA, SVM and LR in predictive performance, scalability, and stability.

- **Nonlinear geometric classification via kernel method:** For complex data where linear models are not enough, CDA supports nonlinear classification via kernel method. We demonstrated with challenging image and chemical datasets that kernel CDA improved over linear CDA and outperformed kernel SVM and kernel LDA. While the core CDA algorithm is efficient, the kernelized version inherits the standard computational bottleneck

of kernel matrix construction. Nonetheless, while kernel CDA offers greater expressiveness and improved capability, linear CDA remains highly valuable for real-world tasks due to its superior training efficiency, interpretability, and reduced risk of overfitting.

More importantly, we emphasize that CDA not only achieves robust predictive performance but also offers superior computational efficiency. Unlike traditional methods such as LDA and SVM, which typically exhibit cubic time complexity, CDA operates with quadratic complexity, resulting in significantly faster runtimes in practice. These advantages make CDA particularly attractive for real-world applications, where scalability, interpretability, and efficiency are essential. As linear classifiers remain widely used across scientific domains for their transparency and speed, CDA represents a valuable advancement for practitioners seeking reliable and computationally lightweight solutions. Lastly, the GDA theoretical framework, from which CDA is derived, may inspire new classifiers under the definition of different geometric constraints.

## 2 GEOMETRIC DISCRIMINANT ANALYSIS (GDA)

In this study, we propose a generalized geometric theoretical framework for centroid-based linear classifiers. In geometry, the generalized definition of centroid is the weighted average of points. For binary classification problem, training a linear classifier involves finding a discriminant (perpendicular to the decision boundary) and a bias. In GDA, we focus on the centroid discriminant basis (CDB) which is defined as the unit vector from the centroid of negative class to positive class. Specifically, we focus on a particular discriminant termed as CDB0, which is constructed from centroids with uniform sample weights. GDA theoretical framework incorporates all the classifiers whose classification discriminant is CDB0 adjusted by geometric corrections on CDB0, which is described in details in the following using an instance with LDA. Moreover, in the GDA theoretical framework, the classifier discriminants are scaling-invariant, which is explained in Appendix. A.2. Thus, throughout, $\gamma \neq 0$ denotes a generic constant independent of the variable of interest. The bias is also excluded from the analysis since it can be naturally determined after the discriminant is fixed (see Appendix. A.3).

Without loss of generality, assume a two-dimensional space (see Appendix. E for any-dimensional proofs). We derive the GDA theoretical framework as a generalization of Fisher's LDA (hereafter referred to simply as LDA) and includes it under a certain geometric constraint. In LDA, the linear discriminant (LD) is derived as the maximization of between-class variance to within-class variance:

$$S = \frac{\sigma_{\mathrm{b}}^2}{\sigma_{\mathrm{w}}^2} = \frac{(\boldsymbol{w}^{\mathrm{T}}\boldsymbol{\nu}_1 - \boldsymbol{w}^{\mathrm{T}}\boldsymbol{\nu}_0)^2}{\boldsymbol{w}^{\mathrm{T}}\boldsymbol{\Sigma}_0\boldsymbol{w} + \boldsymbol{w}^{\mathrm{T}}\boldsymbol{\Sigma}_1\boldsymbol{w}} = \frac{(\boldsymbol{w}^{\mathrm{T}}\boldsymbol{\nu}_1 - \boldsymbol{w}^{\mathrm{T}}\boldsymbol{\nu}_0)^2}{\boldsymbol{w}^{\mathrm{T}}(\boldsymbol{\Sigma}_0 + \boldsymbol{\Sigma}_1)\boldsymbol{w}} \tag{1}$$

where $\boldsymbol{\nu}_0$ and $\boldsymbol{\nu}_1$ are the means of negative and positive class, $\boldsymbol{\Sigma}_0$ and $\boldsymbol{\Sigma}_1$ are their covariance matrices, $\boldsymbol{w}$ is the projection coefficient. The maximum is obtained when $N\boldsymbol{\Sigma}\boldsymbol{w} = \boldsymbol{\nu}_1 - \boldsymbol{\nu}_0$ (Fisher (1936)), or $N\boldsymbol{\Sigma}\gamma\boldsymbol{w}_{\mathrm{LD}} = \boldsymbol{\nu}_1 - \boldsymbol{\nu}_0$, where $\boldsymbol{w}_{\mathrm{LD}}$ is the normalized LDA discriminant in the GDA theoretical framework with a normalizing constant $\gamma$, $N$ is the total number of samples. $\boldsymbol{\Sigma}$ is the sum of within-class covariance matrices of each class $\boldsymbol{\Sigma}_0 + \boldsymbol{\Sigma}_1$:

$$\boldsymbol{\Sigma} = \begin{bmatrix} \sigma0_{xx}^2 & \sigma0_{xy}^2 \\ \sigma0_{yx}^2 & \sigma0_{yy}^2 \end{bmatrix} + \begin{bmatrix} \sigma1_{xx}^2 & \sigma1_{xy}^2 \\ \sigma1_{yx}^2 & \sigma1_{yy}^2 \end{bmatrix} = \begin{bmatrix} \sigma_{xx}^2 & \sigma_{xy}^2 \\ \sigma_{yx}^2 & \sigma_{yy}^2 \end{bmatrix} \tag{2}$$

And the inverse of the covariance matrix $\boldsymbol{\Sigma}$ is:

$$\boldsymbol{\Sigma}^{-1} = \frac{\mathrm{adj}(\boldsymbol{\Sigma})}{|\boldsymbol{\Sigma}|} = \frac{1}{\sigma_{xx}^2\sigma_{yy}^2 - \sigma_{xy}^2\sigma_{yx}^2} \begin{bmatrix} \sigma_{yy}^2 & -\sigma_{xy}^2 \\ -\sigma_{yx}^2 & \sigma_{xx}^2 \end{bmatrix} = \frac{\sigma_{yy}^2}{\sigma_{xx}^2\sigma_{yy}^2 - \sigma_{xy}^2\sigma_{yx}^2} \begin{bmatrix} 1 & -\frac{\sigma_{xy}^2}{\sigma_{yy}^2} \\ -\frac{\sigma_{yx}^2}{\sigma_{yy}^2} & \frac{\sigma_{xx}^2}{\sigma_{yy}^2} \end{bmatrix} \tag{3}$$

where $\mathrm{adj}(\boldsymbol{\Sigma})$ is the adjugate of $\boldsymbol{\Sigma}$. Let $\Delta\boldsymbol{\mu}$ denote the unit vector of $\Delta\boldsymbol{\nu}$, then $\boldsymbol{\nu}_1 - \boldsymbol{\nu}_0 = \gamma(\boldsymbol{\mu}_1 - \boldsymbol{\mu}_0) = \gamma\Delta\boldsymbol{\mu} = \gamma[\Delta\mu_x, \Delta\mu_y]^{\mathrm{T}}$, and $\boldsymbol{w}_{\mathrm{LD}} = \gamma\boldsymbol{\Sigma}^{-1}(\boldsymbol{\nu}_1 - \boldsymbol{\nu}_0) = \gamma\boldsymbol{\Sigma}^{-1}(\boldsymbol{\mu}_1 - \boldsymbol{\mu}_0)$. Further,

$$\boldsymbol{w}_{\mathrm{LD}} = \frac{\gamma\sigma_{yy}^2}{\sigma_{xx}^2\sigma_{yy}^2 - \sigma_{xy}^2\sigma_{yx}^2} \begin{bmatrix} 1 & -\frac{\sigma_{xy}^2}{\sigma_{yy}^2} \\ -\frac{\sigma_{yx}^2}{\sigma_{yy}^2} & \frac{\sigma_{xx}^2}{\sigma_{yy}^2} \end{bmatrix} \begin{bmatrix} \Delta\mu_x \\ \Delta\mu_y \end{bmatrix} = \gamma \left( \begin{bmatrix} 1 & 0 \\ 0 & 1 \end{bmatrix} + \begin{bmatrix} 0 & -\frac{\sigma_{xy}^2}{\sigma_{yy}^2} \\ -\frac{\sigma_{yx}^2}{\sigma_{yy}^2} & \frac{\sigma_{xx}^2}{\sigma_{yy}^2} - 1 \end{bmatrix} \right) \begin{bmatrix} \Delta\mu_x \\ \Delta\mu_y \end{bmatrix}$$

$$= \gamma([\Delta\mu_x, \Delta\mu_y]^{\mathrm{T}} + \boldsymbol{C}_{\mathrm{correction}}[\Delta\mu_x, \Delta\mu_y]^{\mathrm{T}}) \tag{4}$$

where $C_{\text{correction}}$ is a correction matrix that acts as the second-order term associated with the sum of covariance matrices. Since $w_{\text{CDB0}}$ is the unit vector constructed from centroids with uniform sample weights (i.e., arithmetic means), it can be written as $w_{\text{CDB0}} = [\Delta\mu_x, \Delta\mu_y]^{\text{T}}$. As in the covariance matrix, $\sigma_{xy}^2 = \sigma_{yx}^2$, and let $c_{xy} = \sigma_{xy}^2/\sigma_{yy}^2$, $c_{xx/yy} = \sigma_{xx}^2/\sigma_{yy}^2 - 1$, the linear discriminant in Eq. 4 can be compressed into the following general form (Fig. 4a-c, general case):

$$w_{\text{GD}} = w_{\text{LD}} = \gamma(\begin{bmatrix} \Delta\mu_x \\ \Delta\mu_y \end{bmatrix} + \begin{bmatrix} 0 & -c_{xy} \\ -c_{xy} & c_{xx/yy} \end{bmatrix} \begin{bmatrix} \Delta\mu_x \\ \Delta\mu_y \end{bmatrix}) = \gamma(w_{\text{CDB0}} + C_{\text{correction}} w_{\text{CDB0}}). \quad (5)$$

Since from Eq. 5, $w_{\text{LD}}$ can be decomposed into the basis $w_{\text{CDB0}}$ and a correction, we write $w_{\text{GD}}$ to indicate that $w_{\text{LD}}$ is a geometrical discriminant (GD), a discriminant geometrically modified from $w_{\text{CDB0}}$. This geometrical modification can be intuitively interpreted as rotations on $w_{\text{CDB0}}$.

Starting from Eq. 5, we have the following special cases to consider, which represent different forms of geometrical modification applied to $w_{\text{CDB0}}$:

**Special case 1.** If we assume two variables have the same variance ($\sigma_{xx}^2 = \sigma_{yy}^2$), then $c_{xx/yy} = 0$. Eq. 5 becomes (Fig. 4, special case 1):

$$w_{\text{LD}} = \gamma(\begin{bmatrix} \Delta\mu_x \\ \Delta\mu_y \end{bmatrix} + \begin{bmatrix} 0 & -c_{xy} \\ -c_{xy} & 0 \end{bmatrix} \begin{bmatrix} \Delta\mu_x \\ \Delta\mu_y \end{bmatrix}) = \gamma(w_{\text{CDB0}} + C_{\text{correction}} w_{\text{CDB0}}) \quad (6)$$

From Eq. 6, there are two special cases:

**Special case 1.1.** If two covariance matrices are same ($\Sigma_0 = \Sigma_1$), then the following also holds: $c_{xy} = \frac{\sigma_{xy}^2}{\sigma_{yy}^2} = \frac{\mathbb{E}[(x-\mu_x)(y-\mu_y)]}{\sigma_x \sigma_y} = r_{xy}$ where $r_{xy}$ is the Pearson correlation coefficient (PCC) between x and y of the samples in each class. Equation 6 becomes (Fig. 4, special case 1.1):

$$w_{\text{LD}} = \gamma(\begin{bmatrix} \Delta\mu_x \\ \Delta\mu_y \end{bmatrix} + \begin{bmatrix} 0 & -r_{xy} \\ -r_{xy} & 0 \end{bmatrix} \begin{bmatrix} \Delta\mu_x \\ \Delta\mu_y \end{bmatrix}) = \gamma(w_{\text{CDB0}} + C_{\text{correction}} w_{\text{CDB0}}) \quad (7)$$

**Special case 1.1.1.** From Eq. 7, if there is no correlation between x and y variables (e.g., $r_{xy} = 0$), then the equation becomes (Fig. 4, special case 1.1.1):

$$w_{\text{LD}} = \gamma([\Delta\mu_x, \Delta\mu_y]^{\text{T}}) = [\Delta\mu_x, \Delta\mu_y]^{\text{T}} = w_{\text{CDB0}} \quad (8)$$

which is equivalent to MDC method except for the bias. The second equal mark is from the fact that $[\Delta\mu_x, \Delta\mu_y]^{\text{T}}$ is already a unit vector, thus $\gamma = 1$.

**Special case 1.2.** From Eq. 6, if two classes are symmetric about one variable, i.e., $\sigma0_{xy}^2 = -\sigma1_{xy}^2 \neq 0$, then $c_{xy} = \frac{\sigma_{xy}^2}{\sigma_{yy}^2} = 0$. In this case, the obtained discriminant is the same as in Special case 1.1.1 (Fig. 4, special case 1.2):

$$w_{\text{LD}} = \gamma[\Delta\mu_x, \Delta\mu_y]^{\text{T}} = [\Delta\mu_x, \Delta\mu_y]^{\text{T}} = w_{\text{CDB0}} \quad (9)$$

Fig. 4c shows how LDA applies geometric modifications to $w_{\text{CDB0}}$ as a shape adjustment for different data shapes. When two covariance matrices are similar (Special case 1.1), the correction term acts only when variables $x$ and $y$ have certain extent of correlation and increases with this correlation (Fig. 4, third row). Interestingly, when further assumptions are made that two variables have no correlation (Special case 1.1.1), or when two classes are symmetric about one variable (Special case 1.2), we can see from Eq. 8-9 that $w_{\text{LD}}$ approaches $w_{\text{CDB0}}$, which is indeed what we observe in the last two rows of Fig. 4. Videos showing these special cases using 2D simulated data can be found from this link[1].

The demonstrations of all these cases for higher-dimensional space are in the Appendix. E.

From the above derivation, Eq. 5-9 show that the solution of LDA can be represented by $w_{\text{CDB0}}$ superimposed with a geometric correction on $w_{\text{CDB0}}$, and the geometric correction term is obtained under the constraint of Eq. 1, which solves the maximization of the projection variance ratio. Without loss of generality, the conclusion can be extended to other linear classifiers with different constraints imposed on the geometric correction term, for instance MDC, where the correction terms are all zero.

---

[1] https://drive.google.com/drive/folders/1E3QqNzkBz7hdTWpBhEIA75pxlAY4xxw3?usp=sharing

Here, we propose the generalized GDA theoretical framework in which not only the geometric correction term can impose any constraint based on different principles, but also any number of correction terms can act together on $\boldsymbol{w}_{\text{CDB0}}$ to create the classification discriminant, given by:

$$\boldsymbol{w}_{\text{GD}} = \gamma(\boldsymbol{w}_{\text{CDB0}} + \boldsymbol{C}_1 \boldsymbol{w}_{\text{CDB0}} + \boldsymbol{C}_2 \boldsymbol{w}_{\text{CDB0}} + \cdots + \boldsymbol{C}_n \boldsymbol{w}_{\text{CDB0}}) \tag{10}$$

in which LDA only involves the first order base term $\boldsymbol{w}_{\text{CDB0}}$ and the second order correction $\boldsymbol{C}_1$, while higher order terms $\boldsymbol{C}_2, \boldsymbol{C}_3, \ldots$ become 0. Fig. 1 shows that LDA is derived by the general GD equation imposing one variance-based geometric correction in Fig. 1b, shown in Fig. 1a using an instance of 2d artificial data with specific covariance.

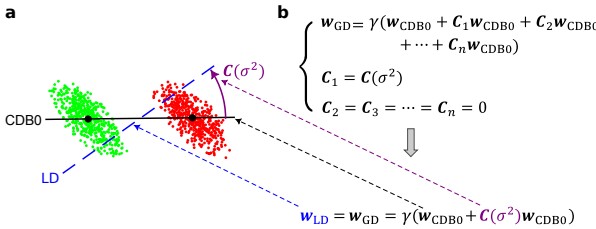

Figure 1: (a) A 2D instance showing that LD (blue dashed) can be constructed from CDB0 (black solid) using covariance correction which is a geometrical correction. (b) The evolution from CDB0 to LD. The first equation is the general form of geometrical discriminants (GD) that can be constructed by CDB0 with geometric corrections. If only one correction exists and this correction is a covariance-related matrix, LD can be derived from the general expression. $\gamma$ denotes a normalizing factor. LD: Linear Discriminant; CDB0: Centroid Discriminant Basis 0.

Since GDA is a framework rather than a specific algorithm, it does not itself provide global theoretical guarantees on performance or convergence. Instead, such guarantees must be established for each concrete classifier instantiated within the framework. Therefore, in Section D we provide a section on CDA convergence proof, where we provide a formal mathematical proposition showing that, under the defined fitness function, CDA is guaranteed to converge. This convergence relies solely on the monotonicity and boundedness of the objective sequence and therefore holds independently of the specific 1D optimizer employed (e.g., BO, Brent's method, etc).

## 3 CENTROID DISCRIMINANT ANALYSIS (CDA)

Based on the GDA theoretical framework, in this study we propose an efficient geometric centroid-based linear classifier named centroid discriminant analysis (CDA), then introduce the extension to nonlinear classification via kernel method in section 5. As described by Eq. 10 in the GDA theory, the model of a geometric classifier can be expressed by the basis CDB0 imposed by geometric corrections on this basis. We first give the conclusion that the final discriminant of CDA after $n$ rotations is in the form $\boldsymbol{w}_{\text{CDA}}^{(n)} = \gamma(\boldsymbol{w}_{\text{CDB0}} + \boldsymbol{C}_1 \boldsymbol{w}_{\text{CDB0}}) = \boldsymbol{w}_{\text{GD}}$, where $\boldsymbol{C}_1 = \prod^n \boldsymbol{A}_{\text{cda}} - \boldsymbol{I}$ is the geometric correction operator term, $\boldsymbol{I}$ is the identity operator, and $\boldsymbol{A}_{\text{cda}}$ is the operator of a single CDA rotation). The complete derivation of CDA in GDA theoretical framework can be found in Appendix. F.

From the geometric point of view, CDA is built on the basis CDB0 in the GDA theoretical framework, then subjected to a series of geometric constraints guiding the rotations of the discriminant CDB in high-dimensional spaces. CDA follows a performance-dependent training that, starting from CDB0 involves the iterative calculation of new adjusted centroid discriminant lines whose optimal rotations on the associated 2D planes are searched via Bayesian optimization. The following parts together with Figure. 5 describe the CDA workflow, mechanisms and principles in detail.

**Performance-associated CDB Classifier**: In GDA theoretical framework, CDB is defined as the unit vector pointing from the geometric centroid of the negative class to positive class. The geometric centroid is defined in a general sense which considers the weights of the data. The space of CDB consists of unit vectors obtained with all possible weights.

Apart from the discriminant line, a bias is further required to realize classification by offering a decision boundary for data projected onto the discriminant. We perform a search for this bias by

checking the middle points of every two consecutive sorted projections. Thus, given $N$ samples, there are $N-1$ candidates. We name the best candidate as optimal operating point (OOP) and select OOP according to a performance-score, defined as $(Fscore^{\text{pos}} + Fscore^{\text{neg}} + ACscore)/3$ (see Appendix. G for definitions), where pos and neg refer to evaluating the metric for each class. The performance score is a comprehensive metric that simultaneously considers sensitivity, recall, and specificity, providing a fair evaluation that accounts for biased models trained on imbalanced data, owing to the more conservative AC-score metric (Wu & Cannistraci (2025)). With this OOP search strategy, any vector in the space of CDB is associated with a performance-score. The OOP search can be performed efficiently in $O(N\log N)$ time (see Alg. 3 for pseudocode). Importantly, during each rotation within the 2D plane, CDA explicitly selects the direction that maximizes the performance score and progressively refines this choice through continuous rotations. Because the optimization target is transparent at every step, CDA offers an inherently explainable learning process.

**CDA as Consecutive Geometric Rotations of CDB in 2D planes**: Our idea is to start the optimization path from CDB0, continuously rotate the classification discriminant on 2D planes on which there is a high probability of better performance. To construct such a 2D plane with another vector, a key observation is that samples whose projections onto CDB are close to the decision boundary (i.e. OOP) should have more weights, because these samples are prone to overlapping with samples from the other class, causing misclassification. Thus, we compute another CDB using centroids with shifted sample weights toward OOP (see next part). On the plane formed by these two CDBs, the best rotation is found by Bayesian optimization. For clarity, we have the following definition: the first vector in each rotation is termed CDB1, the second vector in each rotation is termed CDB2, the CDB searched with the best performance is termed CDA, and at the end of each rotation CDA becomes CDB1 for the next rotation. CDA rotation refers to this rotation process. Figure 5a shows the diagram for the first CDA rotation, where CDB1 equals to CDB0 calculated using uniform sample weights. The CDA training stops when meeting any of the two criteria: (1) Reach the maximum 50 iterations. (2) the coefficient of variation (CV) of the last 10 performance-score is less than a threshold, indicating that the training has converged (see Alg. 1,2 and 4 for pseudocode).

**Sample Weights Update Strategy**: Given CDB1 and the associated OOP in each CDA rotation, the distance of all projections to OOP can be obtained by $d_i = |q_i - oop|, \ \forall i \in \{1, 2, \ldots, N\}$, which is then reversed by $\boldsymbol{d}_{\text{r}} = |\boldsymbol{d} - \min(\boldsymbol{d}) - \max(\boldsymbol{d})|$, in alignment with the purpose of giving larger weights to points close to the decision boundary. Since only the relative information of sample weights is important, they are L2-normalized and decay smoothly from previous sample weights by $\boldsymbol{\alpha} = \boldsymbol{\alpha} \odot \boldsymbol{d}_{\text{r}}/\|\boldsymbol{\alpha} \odot \boldsymbol{d}_{\text{r}}\|_2$, where $\odot$ indicates the Hadamard (element-wise) product. Specifically, in the first CDA rotation, the CDB1 is obtained with uniform sample weights, which corresponds to CDB0 in the GDA theoretical framework.

**Rotation Optimization**: The CDA rotation aims to search for a unit vector CDB with the best performance-score in a 2d plane spanned by CDB1 and CDB2. There are several optimizers that can realize this single-parameter search of rotation angle. Based on our findings in Section L, we make the following recommendations. For maximum speed, Fibonacci search is the preferred choice, and BO is a strong second option. Be aware that BO seems to prefer log-transform of the data. For large-scale datasets, Fibonacci search remains the most efficient and still reliable method. In the main article, we proceed with BO due to its high performance. BO is a statistical-based technique to estimate the global minimum of a function with as fewer evaluations as possible. CDA leverages this high-efficiency characteristic of BO to achieve fast training. BO has $O(Z^3)$ time complexity when it searches a single parameter, where $Z$ is the number of sampling-and-evaluations. CDA employs a strategy that grows BO sampling times from 4 to the maximum 10 with CDA iterations (see Appendix C.1). Figure 5b shows an instance of BO working process. Alg. 5 shows the pseudocode for the CDA rotation as the black box function to optimize by BO.

**Finalization**: As a finalization step, on the best plane CDA refines the discriminant with a null-model statistical test using 100 random CDB lines drawn from the plane (see Appendix. C.2 and Alg. 6).

**Multiclass Prediction**: Error-correcting output codes (ECOC) (Allwein et al. (2001)) is a vote/penalization-based method to make multiclass predictions from trained binary classifiers. To predict the class of a new sample, multiple scores can be obtained from the set of trained binary classifiers. These scores are interpreted as they either vote for or are against a particular class. Depending on the coding matrix and the loss function chosen, ECOC takes the class with the highest overall reward or lowest overall penalty as the predicted class. In this study, the coding matrix for

one-versus-one training scheme is selected, as it creates more linear separability (Acevedo et al. (2022; 2024)). For the type of loss function, hinge-loss is chosen, as our internal tests suggested that this loss leads to the best classification performance for all tested linear classifiers.

# 4    EXPERIMENTAL EVIDENCES ON LINEAR CLASSIFICATION OF REAL DATA

In this section, we compared the proposed CDA with other linear classifiers including LDA, SVM and LR. The 5 tested SVM variants include the original SVM, dual and primal fast SVM implemented by Liblinear, SVM-SGD, and one fast SVM with BO hyperparameter search. LR uses fast implementation by Liblinear (fast SVM and fast LR refers to the dual version unless otherwise specified). Additionally, we include baseline method, CDB0 (equivalent to MDC equipped with OOP bias), to quantify the improvements made by CDA over a centroid-based classification. Their theoretical time complexity is shown in Fig. 2a. Experiment details are in Appendix J. An additional comparison with Partial Least Squares Discriminant Analysis (PLSDA, Barker & Rayens (2003)) - a widely used and competitive linear method especially in chemometrics, genomics, and other high-dimensional applications - with both binary and inherent-multiclass versions can be accessed in Section T).

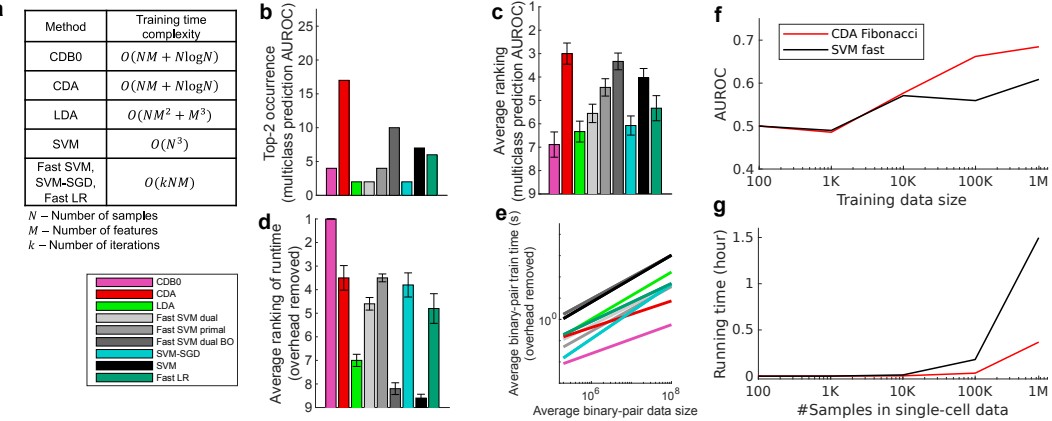

Figure 2: (a) Theoretical training time complexity of linear classifiers. (b-e) Performance on 27 real datasets. (b) Top-2 occurrences and (c) average ranking of classifiers according to multiclass AUROC. Error bars represent standard errors. (d) Average ranking of training time on large datasets with average class-pair data size $N \times M > 10^7$. (e) Lines of linear regression between training times and dataset sizes averaged across binary-pairs in each dataset, in the log-log scale to reveal the scalability. (f-g) Large-scale data test. (f) Test set AUROC and (g) training time on large-scale single-cell data with varying sizes of training samples by CDA-Fibonacci and fast SVM. CDB0: Centroid Discriminant Basis 0; CDA: Centroid Discriminant Analysis; LDA: Linear Discriminant Analysis; SVM: Support Vector Machine; LR: Logistic Regression; BO: Bayesian Optimization.

**Classification Performance on Real Data**: We assessed linear classification performance across 27 datasets, including standard image classification (Coates et al. (2011); Cohen et al. (2017); Hull (1994); Krizhevsky (2009); LeCun et al. (1998); Netzer & Wang (2011); Nilsback & Zisserman (2008); Stallkamp et al. (2011); Xiao et al. (2017)), medical image classification (Yang et al. (2023)), and chemical property prediction tasks (Wu et al. (2018)). These datasets represent a broad range of real-world applications and varying data sizes, enabling evaluations for both training speed and predictive performance. Image data were flattened to 1D vectors; chemical formulas were processed by simplified molecular input line entry system (SMILES) tokenized encoding (see Appendix K). Each dataset was split into a 4:1 ratio for training and test sets. The final model for each method was an unweighted ensemble of the five cross-validated models on the training set. We used one-versus-one training scheme and use external framework to realize multiclass prediction, instead of the native multiclass approaches (See reasons in Section T).

Fig. 2b-e show the test set multiclass prediction performance. In Fig. 2b, CDA achieved a top-2 occurrence of AUROC on 17 out of 27 datasets, outperforming all the other linear classifiers, indicating its stability and competitiveness. Fig. 2c shows that CDA achieved the highest average ranking around 3.3, followed by fast SVM BO and SVM, however, their extremely low average

ranking of training speed in Fig. 2d indicates the impracticality for large-scale datasets (Panel (d) shows results on large datasets with average class-pair data size $N \times M > 10^7$, because in practice we care more about the time consumption on large datasets rather than small ones). See Section N for other metrics and Section S for complete performance tables. These results of CDA is with BO optimizer and log-transform to stabilize training, while the comparison for different optimizers are in Section L, and for the efficient variant CDA-Fibonacci see Fig. N.

In Fig. 2e we performed linear regressions between training times and data sizes averaged across binary pairs within each dataset with overhead deducted (See Fig. 8a for original data points, and Fig. 8b-c for results with overhead kept). The regression results show that CDB0 exhibits the best scalability due to its simplicity. SVM primal and SVM-SGD are fast for small data size, however, CDA outperforms them when data size gets large. CDA not only improves significantly over CDB0 in performance but also retains similar scalability. This improvement is driven by three key components: generalized centroids with non-uniform weights, sample weight shift strategy, and rotations by Bayesian optimization. The weighted average number of iterations required by CDA per dataset was 29.33 - a small constant that does not contribute to time complexity. In contrast, the iterations required by SVM variants increase significantly with more challenging datasets to achieve a reasonable classification performance. Considering the diversity of tested datasets, CDA demonstrates itself as a generic classifier with strong performance and scalability, making it applicable to large-scale classification tasks across various domains.

We further test CDA on a large-scale dataset, the 1.3-million-cell mouse brain dataset (10x Genomics (2017)), with the largest two classes (See Section. H for details). We applied CDA-Fibonacci as a faster version for large data (See Section L). The results show that with growing sample sizes, CDA outperforms fast SVM on both classification AUROC (Fig. 2f and Appendix Fig. 6), and training speed (Fig. 2g). These results indicate that linear CDA is even more efficient and scalable than the flagship SVM method in efficiency. Hence, CDA as a fast approach has the potential to drive large-scale real-world applications in fields such as biomedicine and autonomous driving.

In addition, we made a comparison between CDA with two prevalent neural network architectures - MLP and ResNet (See Appendix. U.1). In addition, we found that it is feasible to incorporate neural networks in CDA, using their extracted features to train CDA, which improves linear CDA performance. Importantly, CDA can be used to initialze linear layers or MLP in the final layers of a neural network architecture, outperforming their randomly initialized versions (See Appendix U.2).

**CDA's Convergence Property**: We analyzed the relationship between the actual stopping iteration of CDA and the average binary classification performance (ps-score) across different datasets and iteration limits. The results revealed two distinct regimes. In the right half of Fig. 3, for tasks converging before 50 iterations, we observed a significant negative correlation (Pearson's R = –0.48), indicating that datasets with lower performance required more iterations to converge. This highlights the importance of allowing at least 50 iterations, as early stopping before this point may prevent convergence for more challenging tasks. In contrast, for tasks exceeding 50 iterations, the correlation was weak and positive (Pearson's R = 0.184), suggesting that beyond this threshold, the number of iterations no longer plays a significant role in ensuring convergence. This indicates that once 50 iterations are reached, CDA can stabilize regardless of the underlying task performance. To further validate this observation, in the left half of Fig. 3, we compared the distributions of ps-scores obtained under maximum iteration limits of 50 and 150. The estimated probability densities were nearly identical, and the Wilcoxon signed-rank test confirmed no significant difference between the two conditions (p=0.398). This result supports the conclusion that extending the maximum number of iterations beyond 50 does not provide systematic benefits in terms of classification performance. We also provide a similar convergence analysis for fast version CDA-Fibonacci in Appendix O.

## 5 PRELIMINARY TEST ON NONLINEAR KERNEL CDA

When data exhibit strong nonlinearity, linear classifiers often underperform. To address this problem, models must exploit these nonlinear patterns to enhance expressiveness and improve classification performance. In fact, linear CDA can be naturally extended to a nonlinear variant using kernel method while retaining efficiency (see Appendix C.4). The main computational bottleneck lies in the kernel matrix computation, which is a common cost shared by all kernel-based classifiers.

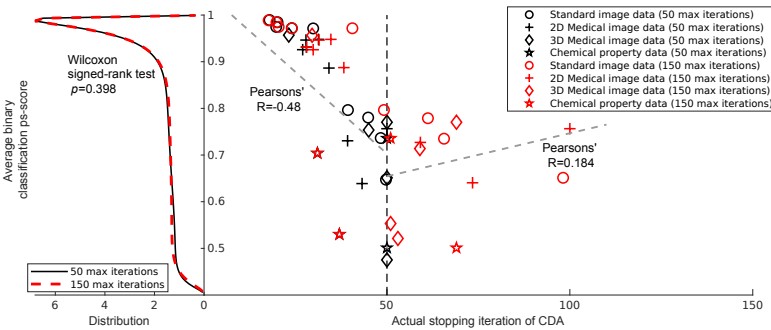

Figure 3: CDA-BO convergence. The right half shows relationship between classification performance and the actual stopping iteration of CDA across datasets under two maximum iteration limits (50 and 150). A negative correlation was observed for tasks converging before 50 iterations (Pearson's R = –0.48), while a weak positive correlation was found for tasks converging after 50 iterations (Pearson's R = 0.184). The gray dash lines show linear regression of points till 50 and points between 51 and 150. The black vertical line indicates the 50-iteration threshold. In the left half, probability density distributions of ps-scores under 50 and 150 maximum iterations show substantial overlap.

To compare the performance of nonlinear CDA and other nonlinear classifiers, we performed tests on 3 challenging datasets, image dataset SVHN, chemical property dataset ClinTox and medical image dataset fracturemnist3d (see Appendix K for data processing). We choose these datasets because they are difficult to classify with linear classifiers, and we are interested in to what extent nonlinear classifiers improve over linear ones. We used a subset of SVHN samples (24,000) due to time limitation to test with kernel method. We compared linear/Gaussian CDA (nCDA), SVM/nSVM and LDA/nLDA. For kernel methods, we performed hyperparameter search for the gaussian parameter $\sigma$. The data were divided into train, validation and test set, where on the validation set the hyperparameter was tuned. The training specifics are in Appendix J.2. The multiclass test set results (on ClinTox binary results since it is a binary task) in Table 1 show that kernel CDA outperforms on 2 out of 3 datasets, while the performance gap compared to the best one (nLDA) is not large. Importantly, kernel CDA improves on linear CDA substantially, suggesting the potential of applying kernel methods on linear CDA for complex data. Despite the fact that gaussian CDA improves substantially over linear CDA in tasks such as the SVHN image classification, we emphasize that linear CDA is still very useful and important, with high efficiency and low resource requirement in computation.

Table 1: Test set classification performance.

| Dataset | Method | AUROC | AUPR | Fscore | ACscore |
|---|---|---|---|---|---|
| SVHN subset (image) | CDA | 0.615±0.02 | 0.63±0.02 | 0.619±0.02 | 0.423±0.05 |
| | nCDA | 0.777±0.01 | 0.782±0.01 | 0.78±0.01 | 0.731±0.02 |
| | SVM | 0.555±0.01 | 0.568±0.007 | 0.551±0.006 | 0.273±0.05 |
| | nSVM | 0.736±0.02 | 0.776±0.009 | 0.756±0.008 | 0.654±0.03 |
| | nLDA | **0.786±0.01** | **0.79±0.01** | **0.789±0.01** | **0.743±0.02** |
| ClinTox (chemical) | CDA | 0.567 | 0.561 | 0.56 | 0.351 |
| | nCDA | **0.625** | **0.627** | **0.627** | **0.46** |
| | SVM | 0.565 | 0.578 | 0.575 | 0.294 |
| | nSVM | 0.500 | 0.481 | 0.480 | 0.000 |
| | nLDA | 0.605 | 0.612 | 0.611 | 0.409 |
| Fracture 3D (medical image) | CDA | 0.518±0.01 | 0.526±0.02 | 0.486±0.02 | 0.279±0.05 |
| | nCDA | **0.625±0.04** | 0.62±0.04 | **0.607±0.04** | **0.577±0.08** |
| | SVM | 0.576±0.009 | 0.579±0.008 | 0.577±0.008 | 0.505±0.05 |
| | nSVM | 0.608±0.06 | 0.591±0.07 | 0.586±0.07 | 0.44±0.2 |
| | nLDA | 0.608±0.03 | **0.626±0.02** | 0.605±0.02 | 0.491±0.1 |

## 6  CONCLUSIONS AND DISCUSSIONS

Linear classifiers, while inherently simpler, are favored in certain contexts due to their reduced tendency to overfit and their interpretability in decision-making. However, achieving both high scalability and robust predictive performance simultaneously remains a significant challenge.

In this study, the introduction of the Geometric Discriminant Analysis (GDA) framework marks a notable step forward in addressing this challenge. By leveraging geometric properties of centroids – a fundamental concept across disciplines – GDA provides a unifying framework for certain linear classifiers. The core innovation lies in a special type of Centroid Discriminant Basis (CDB0), which serves as the foundation for deriving discriminants. These discriminants, augmented with geometric corrections under varying constraints, extend the theoretical flexibility of GDA. Notably, Minimum Distance Classifier (MDC) and Linear Discriminant Analysis (LDA) are shown to be a subset of this broader framework, demonstrating how they converge to CDB0 under specific conditions. This generalization not only validates GDA but also sets the stage for novel classifier designs.

A key practical contribution of this work is the Centroid Discriminant Analysis (CDA), a specialized implementation of GDA. CDA employs geometric rotations of the CDB within planes defined by centroid-vectors with shifted sample weights. These rotations, combined with Bayesian optimization techniques, enhance the method's scalability, achieving quadratic time complexity $O(NM + N\log N)$. Across diverse datasets—including standard images, medical images, and chemical property classifications—CDA demonstrated superior performance in scalability, predictive performance, and stability. We emphasize that CDA not merely robustly outperforms most of existing linear classification methods, but crucially, it achieves this with superior computational efficiency. Specifically, CDA exhibits a quadratic time complexity in the worst-case scenario, compared to the cubic complexity typical of established methods such as LDA and SVM, with significantly shorter runtimes that highlight its practical advantage in real-world applications. These findings hold particular relevance for the broader machine learning community, as linear classifiers remain widely used across numerous scientific domains where interpretability, scalability, and computational efficiency are critical. In such contexts, practitioners often prefer models that are not only robust and fast but also transparent and easy to deploy in real-world decision-making scenarios.

For data exhibiting complex nonlinear structures that linear classifiers cannot adequately capture, linear CDA can be extended to nonlinear form via kernel methods. This generalization significantly broadens the applicability of CDA by enabling it to handle nonlinear patterns in the data. Together, linear and kernel-based CDA form a complementary toolkit for classifications. A practical strategy is to begin with linear CDA to see whether the predictions are satisfying, which offers fast and interpretable results. Our preliminary results also suggest that extending CDA with nonlinear kernels is a promising direction for future research, since the variety of available kernels broadens the opportunity to tailor the mapping of original data onto higher-dimensional spaces according to the specific structure of the data - where it may become more easily separable. Another future work is to explore an inherent multiclass extension of CDA (see Section T), especially given that other classifiers such as LDA and SVM have their inherent multiclass approach. In the cases where $C$ is large and not negligible, inherent multiclass approaches offer attractive lower computational complexity with respect to the number of classes ($C$ versus $C(C-1)/2$ discriminants in pairwise schemes).

One limitation of this study is that the current CDA does not have a natively multiclass approach, hence it can be interesting to investigate how to formulate CDA in a natively multiclass way. However, in Section T we showed that using inherently multiclass LDA, SVM, and PLS-DA the performance does not improve and even drops significantly for SVM, a reason for which in this study we decided to focus on binary CDA classification and leveraging external multiclass framework. In addition, in this study, the GDA framework is used to show that LDA is a special centroid-based method case, and then is adopted to design CDA. CDA itself is presented as a single, highly effective derivative, but the GDA framework's utility in systematically generating other novel classifiers is not investigated enough. Since GDA in the current formulation is a framework that can include only centroid based classifiers, methods such as SVM that are based on large margin classification are not included in the GDA framework to explain and design classifiers based on linear centroid-based separability. The mission to investigate the extent to which GDA framework may inspire the design of new classifiers will be a not trivial point to expand in future studies.

In conclusion, the GDA framework and its CDA implementation represent a paradigm shift in classifier design, combining the interpretability of geometric principles with state-of-the-art computational efficiency. This work not only advances the field of classification but also lays groundwork for innovative approaches to supervised learning.

ACKNOWLEDGMENTS

This work was supported by the Zhou Yahui Chair Professorship award of Tsinghua University (to CVC), the National High-Level Talent Program of the Ministry of Science and Technology of China (grant number 20241710001, to CVC), and National Natural Science Fund of China - Research Fund for International Senior Scientists (RFIS-III, grant number W2531064, to CVC). The authors thank Yuchi Liu and Mo Yang for their administrative support.

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

# A SUPPLEMENTARY: GDA THEORETICAL FRAMEWORK

## A.1 DIFFERENCE BETWEEN THEORY AND THEORETICAL FRAMEWORK

We emphasize that GDA does not constitute a new theory, but rather a theoretical framework. A theory in data science and machine learning provides a foundational explanation often accompanied by rigorous mathematical derivations, such as performance bounds or convergence guarantees. In contrast, a theoretical framework offers a structured set of concepts and assumptions, typically formalized through equations, that guide the design of methods and the interpretation of results - without necessarily proving theoretical performance guarantees. Within this clarified scope, GDA serves as the framework under which we define and motivate an efficient geometric, centroid-based linear classification method: Centroid Discriminant Analysis (CDA), which is discussed in Section 3.

## A.2 SCALING INVARIANCE OF THE DISCRIMINANTS IN THE GDA FRAMEWORK

GDA is a geometrical projection-based theoretical framework. In this framework, the discriminant (the high-dimensional vector orthogonal to the classification boundary) of the involved classifiers (including CDA), is normalized to a unit vector before projecting data. The final classification depends solely on these projections; scaling the discriminant length by any factor (including terms like $\sigma_{yy}^2/|\Sigma|$ in the derivation of GDA by LDA) uniformly scales all projections without changing their relative positions. Consequently, the optimal threshold (which can be found by search) and classification results remain unchanged. This normalization is explicitly implemented in CDA (see, for example, Appendix C.3, Algorithm 1, line 4: "Normalize $\boldsymbol{w}_{\mathrm{CDB1}}\ldots$"), ensuring that only the direction is refined during iterations.

This property distinguishes GDA and CDA from approaches like SVM, where the magnitude of the discriminant vector is tied to the margin width during training. However, even for SVM, once the model is trained, scaling $\boldsymbol{w}$ and $b$ together leaves predictions unchanged because the decision function $y = H(\boldsymbol{w}^{\mathrm{T}}\boldsymbol{x} + b)$ is homogeneous in $\boldsymbol{w}$ and $b$. ($y$: predicted class; $H(\cdot)$: Heaviside function; $\boldsymbol{x}$: sample; $b$: bias). In CDA training, magnitude carries no interpretive meaning - only direction matters. These also reflect that GDA reinterprets certain existing classifiers from the geometric projection perspective, which is an innovation that inherently differs from their own interpretations.

## A.3 EXPLANATION OF THE DISCRIMINANT BIAS IN THE GDA FRAMEWORK

In practice, GDA-based methods—including CDA—explicitly normalize the discriminant to unit length before projection (see Algorithm 1, line 4). This normalization ensures that the training process refines only the direction of the discriminant, while the bias can always be computed afterwards by placing a threshold between the projected class distributions. For binary classification, the bias may be chosen using a variety of monotonic criteria—e.g., the midpoint between class means, the empirical risk minimizer on the projected line, or a direct search for the threshold optimizing AUROC, accuracy, or any other performance metric. Because projection preserves the ordering of samples, all of these methods yield a valid bias once the discriminant direction is fixed.

This perspective also clarifies that the discriminant and the bias play fundamentally different roles: the discriminant determines the orientation of the separating hyperplane, while the bias determines its position. The former is geometrically meaningful in GDA; the latter can always be adjusted post hoc without altering the interpretation of the discriminant.

## A.4 GDA Demonstration with 2D Artificial Data

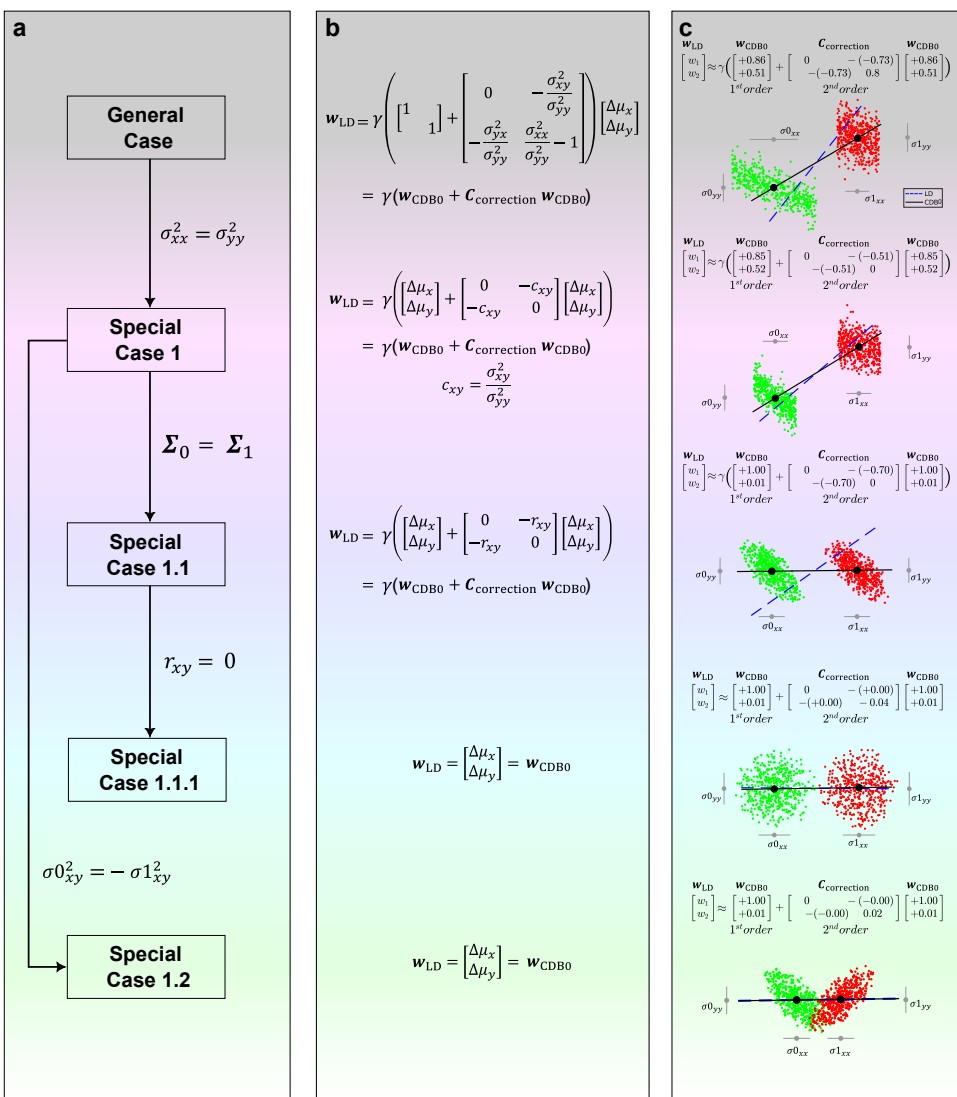

Figure 4: The GDA theoretical framework in 2 dimensions showing how the relation between LD and CDB0 evolves under specific conditions. (a) The relations between LD and CDB0 proceeding from the general case to different special cases under the conditions shown along each arrow. (b) The specific expressions of LD in terms of CDB0 and corrections corresponding to each case in column (a). (c) Binary classifications models of LD (blue dashed) and CDB0 (black solid) on different 2D data corresponding to each case in column (a). The lines show the direction instead of the unit vector of the discriminants. LD: Linear Discriminant; CDB0: Centroid Discriminant Basis 0. $\gamma$ denotes a generic normalizing factor.

# B  CDA Schematic Diagram

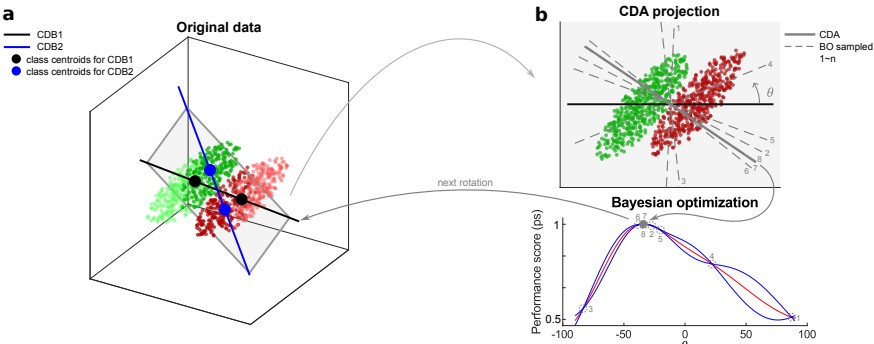

Figure 5: Diagram for the first CDA rotation. (a) CDB1 is obtained from specific sample weights. CDB2 is obtained from centroids with a shifted sample weights toward the decision boundary. Darker points represent larger sample weights. CDB1 and CDB2 form a 2d plane on which BO is performed to estimate the best classification discriminant. (b) The process of BO to estimate the best discriminant. The rotation angle $\theta$ from CDB1 is the only independent variable to search in BO. The estimated optimal line CDA serves as the new CDB1 in the next CDA rotation. CDB: Centroid Discriminant Basis; CDA: Centroid Discriminant Analysis.

# C  The CDA Algorithm

## C.1  Bayesian Optimization

One factor related to the effectiveness of CDA is the estimation with varying precision in BO. We set the number of BO sampling times to $\min(3 + rot, 10)$ as default parameter during the CDA rotation, where $rot$ is the $rot$-th rotation. During the first few CDA rotations, the BO estimation has relatively less precision. This gives CDA enough randomness to first enter a large region in which there are more discriminants with high performance, under the hypothesis that regions close to global optimum or suboptimum have more high-performance solutions that are more likely to be randomly selected. Starting from this large region, BO precision is gradually enhanced to refine the search in or close to this region in terms of Euclidean distance, and force CDA to converge. The upper limit of 10 CDA rotations still creates a small level of imprecision, in order to make CDA escape from small-size local minimum, to acquire higher performance, and improve training efficiency.

## C.2  Refining CDA with statistical examination on 2D plane

In CDA, the plane associated with the best performance is selected, but it is uncertain whether there are higher-performance discriminants on this plane, since the precision might not be high enough with 10 BO samplings and even lower for less samplings. To determine whether the BO is precise enough, a statistical examination using p-value with respect to null-model is performed. On this best plane, we generate 100 random CDBs, run BO in turn for 10, 20 and to the maximum 30 BO samplings, until the p-value of training performance-score of the BO-estimated CDB with regard to the null model is 0, otherwise using the best random CDB. In this way, we are able to give a confidence level of the BO estimated discriminant in the best CDA rotation plane. Alg. 6 shows the pseudocode for the refining finalization step.

## C.3  Linear CDA Pseudocode

This study deals with a supervised binary classification problem on labeled data $\{\boldsymbol{X}, \boldsymbol{y}\} = \{\boldsymbol{x}_i, y_i\}_{i=1}^N$ with $N$ samples and $M$ features. The samples consist of positive and negative class data $\chi_1$ and $\chi_2$ with sizes $N_1$ and $N_2$, respectively, and the corresponding labels are tokenized as 1 and 0, respectively. The sample features are flattened to 1D if they have higher dimensions, such as

image data. The following section introduces CDA in pseudo-code. CDA rotations, BO sampling and p-value computing involve constant numbers, thus, they do not contribute to the time complexity.

---

**Algorithm 1** CDA Main Algorithm (CDA)    $O(NM + N\log N)$

---

**Input:** $N \times M$ data matrix $\boldsymbol{X}$, $N \times 1$ labels $\boldsymbol{y}$
Initialize $\boldsymbol{\alpha} = [1; 1; \ldots; 1]^{N \times 1}/\sqrt{N}$    $\{O(N)\}$
Compute $\boldsymbol{w}_{\text{CDB1}} = \sum\limits_{c=1}^{2} (-1)^{c+1} \frac{1}{N_c} \sum\limits_{\boldsymbol{x}_i \in \chi_c} \alpha_i \boldsymbol{x}_i$    $\{O(NM)\}$
Normalize $\boldsymbol{w}_{\text{CDB1}} = \boldsymbol{w}_{\text{CDB1}}/\|\boldsymbol{w}_{\text{CDB1}}\|_2$    $\{O(M)\}$
Initialize $ps^* = 0$    $\{O(1)\}$
**for** $i = 1$ **to** $50$ **do**
  $\boldsymbol{\alpha} = \text{updateSampleWeights}(\boldsymbol{X}, \boldsymbol{y}, \boldsymbol{\alpha}, \boldsymbol{w}_{\text{CDB1}})$    $\{O(NM + N\log N)\}$
  Compute $\boldsymbol{w}_{\text{CDB2}} = \sum\limits_{c=1}^{2} (-1)^{c+1} \frac{1}{N_c} \sum\limits_{\boldsymbol{x}_i \in \chi_c} \alpha_i \boldsymbol{x}_i$    $\{O(NM)\}$
  Normalize $\boldsymbol{w}_{\text{CDB2}} = \boldsymbol{w}_{\text{CDB2}}/\|\boldsymbol{w}_{\text{CDB2}}\|_2$    $\{O(M)\}$
  Set $N_{\text{BO}} = \min(i + 3, 10)$    $\{O(1) \text{ Number of BO sampling}\}$
  $(\boldsymbol{w}_{\text{CDA}}, ps) = \text{CdaRotation}(\boldsymbol{w}_{\text{CDB1}}, \boldsymbol{w}_{\text{CDB2}}, \boldsymbol{X}, \boldsymbol{y}, N_{\text{BO}})$    $\{O(NM + N\log N)\}$
  **if** $ps > ps^*$ **then**
    $(\boldsymbol{w}_{\text{CDB1}}^*, \boldsymbol{w}_{\text{CDB2}}^*, ps^*) = (\boldsymbol{w}_{\text{CDB1}}, \boldsymbol{w}_{\text{CDB2}}, ps)$    $\{O(M)\}$
  **end if**
  **if** coefficient of variation of the last 10 $ps < 0.001$ **then**
    **break**    $\{\text{Early stop convergence}\}$
  **end if**
  Update $\boldsymbol{w}_{\text{CDB1}} = \boldsymbol{w}_{\text{CDA}}$    $\{O(M)\}$
**end for**
Compute $(\boldsymbol{w}_{\text{CDA}}, oop) = \text{refineOnBestPlane}(\boldsymbol{w}_{\text{CDB1}}^*, \boldsymbol{w}_{\text{CDB2}}^*, \boldsymbol{X}, \boldsymbol{y})$    $\{O(NM + N\log N)\}$
**Output:** $\boldsymbol{w}_{\text{CDA}}, oop$

---

**Algorithm 2** Update Sample Weights (updateSampleWeights)    $O(NM + N\log N)$

---

**Input:** $N \times M$ data matrix $\boldsymbol{X}$, $N \times 1$ labels $\boldsymbol{y}$, $N \times 1$ sample weights $\boldsymbol{\alpha}$, $1 \times M$ vector $\boldsymbol{w}_{\text{CDB1}}$
Initialize $oop = \text{searchOOP}(\boldsymbol{X}, \boldsymbol{w})$    $\{O(N\log N)\}$
Compute $\boldsymbol{q} = \boldsymbol{X}\boldsymbol{w}_{\text{CDB1}}^{\text{T}}$    $\{O(NM)\}$
Compute $d_i = |q_i - oop|, \forall i \in \{1, 2, \ldots, N\}$    $\{O(N)\}$
Compute $\boldsymbol{d}_r = |\boldsymbol{d} - \min(\boldsymbol{d}) - \max(\boldsymbol{d})|$    $\{O(N)\}$
Update $\boldsymbol{\alpha} = \boldsymbol{\alpha} \odot \boldsymbol{d}_r/\|\boldsymbol{\alpha} \odot \boldsymbol{d}_r\|_2$    $\{O(N), \odot \text{ is Hadamard product}\}$
**Output:** $\boldsymbol{\alpha}$

---

**Algorithm 3** Search Optimal Operating Point (searchOOP)    $O(N\log N)$

---

**Input:** $N \times 1$ projected points $\boldsymbol{q}$, $N \times 1$ labels $\boldsymbol{y}$
Sort $(\boldsymbol{q}, \boldsymbol{y})$    $\{O(N\log N), \text{ sort according to } \boldsymbol{q}\}$
Initialize $\boldsymbol{cm} = \text{evaluateMetrics}(\boldsymbol{y}, [0; 0; \ldots; 0]^{N \times 1})$    $\{O(N) \text{ initial confusion matrix}\}$
**for** $i = 1$ **to** $N - 1$ **do**
  Update $\boldsymbol{cm} = \text{updateCM}(y_i, \boldsymbol{cm})$    $\{O(1) \text{ scan each label and adjust cm}\}$
  Compute $ps_i = \text{evaluateMetrics}(\boldsymbol{cm})$    $\{O(1)\}$
**end for**
Compute $idx = \arg\max_i ps_i$    $\{O(N)\}$
Compute $ps = \max(\boldsymbol{ps})$    $\{O(N)\}$
Compute $oop = \frac{q_{idx} + q_{(idx+1)}}{2}$    $\{O(1)\}$
**Output:** $oop, ps$

---

---

**Algorithm 4** Approximate Optimal Line by BO (CdaRotation)          $O(NM + N\log N)$

---

**Input:** $1 \times M$ lines $\boldsymbol{w}_{\text{CDB1}}, \boldsymbol{w}_{\text{CDB2}}$, $N \times M$ data matrix $\boldsymbol{X}$, $N \times 1$ labels $\boldsymbol{y}$, total BO iteration $N_{\text{BO}}$

Compute $\boldsymbol{w}_{\text{orth}} = \boldsymbol{w}_{\text{CDB2}} - \boldsymbol{w}_{\text{CDB1}}^{\text{T}} \boldsymbol{w}_{\text{CDB2}} \boldsymbol{w}_{\text{CDB1}}$          $\{O(M)\}$

Normalize $\boldsymbol{w}_{\text{orth}} = \boldsymbol{w}_{\text{orth}} / \|\boldsymbol{w}_{\text{orth}}\|_2$          $\{O(M)\}$

Estimate $\hat{\theta} = \text{BayesianOptimization}(\text{evaluateRotation}(\theta, \boldsymbol{w}_{\text{CDB1}}, \boldsymbol{w}_{\text{orth}}, \boldsymbol{X}, \boldsymbol{y}), N_{\text{BO}})$, $\theta \in [-\frac{\pi}{2}, \frac{\pi}{2}]$ $\{O(NM + N\log N)\}$

Compute $\boldsymbol{w}_{\text{CDA}} = \boldsymbol{w}_{\text{CDB1}} \cos\hat{\theta} + \boldsymbol{w}_{\text{orth}} \sin\hat{\theta}$          $\{O(M)\}$

Compute $\boldsymbol{q} = \boldsymbol{X} \boldsymbol{w}_{\text{CDA}}^{T}$          $\{O(NM)\}$

Get the mean of $(oop, ps) = \text{searchOOP}(\boldsymbol{q}, \boldsymbol{y})$ with 5-fold cross-validation          $\{O(N\log N)\}$

**Output:** $\boldsymbol{w}_{\text{CDA}}, oop, ps$

---

**Algorithm 5** Evaluate the Line with Rotation Angle (evaluateRotation)          $O(NM + N\log N)$

---

**Input:** Rotation angle $\theta$, $1 \times M$ lines $\boldsymbol{w}_{\text{CDB1}}, \boldsymbol{w}_{\text{orth}}$, $N \times M$ data matrix $\boldsymbol{X}$, $N \times 1$ labels $\boldsymbol{y}$

Compute $\boldsymbol{w}_{\text{CDA}} = \boldsymbol{w}_{\text{CDB1}} \cos\theta + \boldsymbol{w}_{\text{orth}} \sin\theta$          $\{O(M)\}$

Compute $\boldsymbol{q} = \boldsymbol{X} \boldsymbol{w}_{\text{CDA}}^{T}$          $\{O(NM)\}$

Get mean $ps = \text{searchOOP}(\boldsymbol{q}, \boldsymbol{y})$ with 5-fold cross-validation scheme          $\{O(N\log N)\}$

**Output:** $ps$

---

**Algorithm 6** Refine on the Best Model (refineOnBestPlane)          $O(NM + N\log N)$

---

**Input:** $1 \times M$ $\boldsymbol{w}_{\text{CDB1}}$ and its orthogonal vector $\boldsymbol{w}_{\text{orth}}$, $N \times M$ data matrix $\boldsymbol{X}$, $N \times 1$ labels $\boldsymbol{y}$

**for** $i = 1$ **to** 100 **do**

    Randomly choose $\theta \in \left[-\frac{\pi}{2}, \frac{\pi}{2}\right]$          $\{O(1)\}$

    Compute $\boldsymbol{w}_{\text{r}} = \boldsymbol{w}_{\text{CDB1}} \cos\theta + \boldsymbol{w}_{\text{orth}} \sin\theta$          $\{O(M)\}$

    Compute $\boldsymbol{q} = \boldsymbol{X} \boldsymbol{w}_{\text{r}}^{\text{T}}$          $\{O(NM)\}$

    Get mean $(oop_{\text{r}}, ps_{\text{r}})_i = \text{searchOOP}(\boldsymbol{q}, \boldsymbol{y})$ with 5-fold cross-validation          $\{O(N\log N)\}$

**end for**

Sort $idx = \text{sortIdx}([ps, ps_{\text{r}}])$          $\{O(1)$ sort returning position indices$\}$

Compute $p = 1 - \frac{(idx(1)-1)}{N_r}$          $\{O(1)\}$

Initialize $N_{\text{BO}} = 0$

**while** $p \neq 0$ and $N_{\text{BO}} < 30$ **do**

    Increment $N_{\text{BO}} = N_{\text{BO}} + 10$

    Compute $(\boldsymbol{w}_{\text{CDA}}, oop, ps) = \text{CdaRotation}(\boldsymbol{w}_{\text{CDB1}}, \boldsymbol{w}_{\text{CDB2}}, \boldsymbol{X}, \boldsymbol{y}, N_{\text{BO}})$          $\{O(NM + N\log N)\}$

    Sort $idx = \text{sortIdx}([ps, ps_r])$          $\{O(1)$ sort returning position indices$\}$

    Compute $p = 1 - \frac{(idx(1)-1)}{N_r}$          $\{O(1)\}$

**end while**

**if** $p \neq 0$ **then**

    Compute $i = \arg\min_i ps_{\text{r},i}$          $\{O(1)\}$

    Set $(\boldsymbol{w}_{\text{CDA}}, oop, ps) = (\boldsymbol{w}_r, oop_{\text{r}}, ps_{\text{r}})_i$          $\{O(1)\}$

**end if**

**Output:** $\boldsymbol{w}_{\text{CDA}}, oop, ps$

---

## C.4 NONLINEAR KERNEL CDA

The CDA algorithm involves computing the CDBs using the equation provided in Appendix. C.3, Algorithm 1:

$$\boldsymbol{w}_{\text{CDB}} = \sum_{c=1}^{2} (-1)^{c+1} \cdot \frac{1}{N_c} \sum_{\boldsymbol{x}_i \in \chi_c} \alpha_i \boldsymbol{x}_i \tag{11}$$

Combining this with the normalization step for each CDB $\boldsymbol{w}_{\text{CDB}} \leftarrow \boldsymbol{w}_{\text{CDB}} / \|\boldsymbol{w}_{\text{CDB}}\|_2$, the equation can be rewritten as:

$$\boldsymbol{w}_{\text{CDB}} = (\boldsymbol{\alpha} \odot \boldsymbol{n} \odot \boldsymbol{y}_t \cdot k)^\top \boldsymbol{X} = \boldsymbol{\beta}^\top \boldsymbol{X}, \tag{12}$$

where $\boldsymbol{\alpha}$ is the vector of sample weights; $\boldsymbol{n} = [1/N_{\text{map}(i)}]$ for all $i \in \{1, 2, \ldots, N\}$ is the weight sum division for each class, and $\text{map}(i)$ maps sample index $i$ to class index $c$; $\boldsymbol{y}_t = (-1)^{c+1}$ are the tokenized labels ($+1$ or $-1$); $k$ is the factor to normalize each CDB as a unit vector.

For the rotated CDBs used in Bayesian optimization (Appendix B, Algorithms 4–5), they can be expressed as linear combinations of CDB1 and orthogonalized CDB2, mixing their corresponding $\boldsymbol{\beta}$ vectors. Thus, the sample coefficient $\boldsymbol{\beta}$ is tracked throughout the training process for each CDB.

The data projection can be expressed as:

$$\boldsymbol{q} = \boldsymbol{X} \boldsymbol{w}_{\text{CDB}}^\top = \boldsymbol{X}(\boldsymbol{\beta}^\top \boldsymbol{X})^\top = \boldsymbol{X} \boldsymbol{X}^\top \boldsymbol{\beta} \tag{13}$$

With this expression, the data projection can incorporate kernel methods to realize nonlinear classification, shown by the following equation.

$$\boldsymbol{q} = Ker(\boldsymbol{X}, \boldsymbol{X}^\top)\boldsymbol{\beta} \tag{14}$$

In fact, kernel methods implicitly projects data into a high-dimensional space, then applying linear classifiers in the transformed feature space.

# D CDA CONVERGENCE PROOF

In this section we provide a mathematical proof with a proposition showing that, under the defined fitness function, CDA is guaranteed to converge. This convergence relies solely on the monotonicity and boundedness of the objective sequence and therefore holds independently of the specific 1D optimizer employed (e.g., BO, Brent's method).

**Proposition.** Let $w_{\mathrm{CDB1}}^{(k)} \in \mathbb{R}^M$ be the CDB1 vector at iteration $k$. In each CDA iteration, $w_{\mathrm{CDB1}}^{(k)}$ determines the CDB2 vector $w_{\mathrm{CDB2}}^{(k)}$ via the CDA algorithm with sample weights update strategy $w_{\mathrm{CDB2}}^{(k)} = g(X, w_{\mathrm{CDB1}}^{(k)})$. Each discriminant has an associated performance metric $ps = f(w_{\mathrm{CDB}})$, bounded above by 1. At the current iteration, the optimal vector is found in the 2D plane spanned by CDB1 and CDB2 as $w_{\mathrm{CDB1}}^{(k+1)} = \lambda_1^* w_{\mathrm{CDB1}}^{(k)} + \lambda_2^* w_{\mathrm{CDB2}}^{(k)}$, where $(\lambda_1^*, \lambda_2^*) = \arg\max_{\lambda_1, \lambda_2 \in \mathbb{R}} f(\lambda_1 w_{\mathrm{CDB1}}^{(k)} + \lambda_2 w_{\mathrm{CDB2}}^{(k)})$, which is then used as the new CDB1 at iteration $k+1$. Since $w_{\mathrm{CDB1}}^{(k)} \in \{\lambda_1 w_{\mathrm{CDB1}}^{(k)} + \lambda_2 w_{\mathrm{CDB2}}^{(k)}\}$ as a special case with $\lambda_1 = 1$ and $\lambda_2 = 0$, it follows that $ps_{\mathrm{CDB1}}^{(k+1)} \geq ps_{\mathrm{CDB1}}^{(k)}$. Therefore, the sequence $ps_{\mathrm{CDB1}}^{(k)} = f(w_{\mathrm{CDB1}}^{(k)})$ is non-decreasing and bounded above, and hence converges to a finite limit $ps^*$.

# E   DERIVING LDA IN THE GDA THEORETICAL FRAMEWORK FOR M-DIMENSIONS (M>2)

In the GDA theoretical framework, we have demonstrated that linear discriminant (LD) can be expressed by the basis CDB0 with different geometric corrections on CDB0 under different conditions, as shown in Fig. 4. The conclusions are not confined only to the 2-dimensional case but also applicable to the $m$-dimensional case ($m > 2$). Here we derive the generalization of the LD coefficient to the $m$-dimensional case. The discriminant of LDA is given by $\boldsymbol{w}_{\mathrm{LD}} = \gamma\boldsymbol{\Sigma}^{-1}(\boldsymbol{\mu}_1 - \boldsymbol{\mu}_0) = \gamma\boldsymbol{\Sigma}^{-1}\boldsymbol{w}_{\mathrm{CDB0}}$, where $\gamma$ is a constant, and it is assumed that the within-class covariance matrix $\boldsymbol{\Sigma}$ is invertible, and $\boldsymbol{\Sigma} = \boldsymbol{\Sigma_0} + \boldsymbol{\Sigma_1}$ is the sum of covariance matrices of individual classes. Denote the elements in $\boldsymbol{\Sigma}$, $\boldsymbol{\Sigma_0}$, and $\boldsymbol{\Sigma_1}$ by $\sigma_{ij}$, $\sigma0_{ij}$, and $\sigma1_{ij}$ for all $i, j \in \{1, 2, \ldots, m\}$, respectively.

**Special Case 1.** Assume that pairwise features have the same covariance $\alpha$, and all pairwise features have the same covariance $\alpha + \beta$ (In experiments, we relax the conditions to have similar variance and similar covariance). Then, $\boldsymbol{\Sigma} = \beta\mathbf{I} + \alpha\mathbf{1}$, where $\mathbf{I}$ is the identity matrix, and $\mathbf{1}$ is the all-one matrix. According to the Woodbury matrix identity, we have $\boldsymbol{\Sigma}^{-1} = \frac{1}{\beta}\mathbf{I} - \frac{\alpha}{\beta(m\alpha+\beta)}\mathbf{1} = \gamma(\mathbf{I} - \frac{\alpha}{(m-1)\alpha+\beta}\mathbf{1}_N)$, where $\mathbf{1}_N$ is the matrix with all ones except on the diagonal. Thus, $\boldsymbol{w}_{\mathrm{LD}} = \gamma(\boldsymbol{w}_{\mathrm{CDB0}} - \frac{\alpha}{(m-1)\alpha+\beta}\mathbf{1}_N\boldsymbol{w}_{\mathrm{CDB0}})$, which corresponds to the row for special case 1 in Fig. 4.

**Special Case 1.1.** Based on the assumptions in Special Case 1, we further assume that the two covariance matrices are similar, i.e., $\boldsymbol{\Sigma_0} \approx \boldsymbol{\Sigma_1}$, so that $\sigma_{0,ij} = \sigma_{1,ij}$ for all $i, j$. We further have $\frac{\sigma_{ij}}{\sigma_{ii}} = \frac{2\sigma0_{ij}}{2\sigma0_{ii}} = \frac{\sigma0_{ij}}{\sqrt{\sigma0_{ii}}\sqrt{\sigma0_{jj}}} = r_{ij} \quad \forall i \neq j$, where $r_{ij}$ is the Pearson correlation coefficient (PCC) between features $i$ and $j$. Based on the assumptions made in this special case, all pairwise PCCs are the same, i.e., $r_{ij} = r$ for all $i \neq j$. Thus, $\boldsymbol{\Sigma}^{-1} = \gamma(\mathbf{I} - \frac{r}{(m-2)r+1}\mathbf{1}_N)$, which shows that the geometric correction part is related to feature correlations. Accordingly, $\boldsymbol{w}_{\mathrm{LD}} = \gamma(\boldsymbol{w}_{\mathrm{CDB0}} - \frac{r}{(m-2)r+1}\mathbf{1}_N\boldsymbol{w}_{\mathrm{CDB0}})$, which corresponds to the row for special case 1.1 in Fig. 4.

**Special Case 1.1.1.** Based on Special Case 1.1, assume further that all pairwise features have no correlations (i.e., $r = 0$), then according to the equation, $\boldsymbol{\Sigma}^{-1} = \gamma\mathbf{I}$. Thus, $\boldsymbol{w}_{\mathrm{LD}} = \gamma\boldsymbol{w}_{\mathrm{CDB0}} = \boldsymbol{w}_{\mathrm{CDB0}}$, where $\gamma = 1$ since $\boldsymbol{w}_{\mathrm{CDB0}}$ is already a unit vector, which corresponds to the row for special case 1.1.1 in Fig. 4. By these derivations, we show how LDA converges to CDB0 under specific conditions of the data.

# F   DERIVING CDA IN THE GDA THEORETICAL FRAMEWORK

In this section, we give the mathematical demonstration of how CDA is formulated in the GDA theoretical framework. Since the correction matrix for LDA in Section 2 is a special case of a linear operator, we extend to a more general correction term deriving CDA in the GDA theoretical framework. The construction of the geometric discriminant $\boldsymbol{w}_{\mathrm{GD}}$ can be realized by continuously rotating CDB0 in planes that satisfy several constraints, shown by:

$$
\begin{aligned}
\boldsymbol{w}_{\mathrm{CDA}}^{(n)} &= \boldsymbol{A}_{\mathrm{cda}}(\boldsymbol{w}_{\mathrm{CDB1}}^{(n)}, \boldsymbol{\alpha}^{(n)}, \boldsymbol{X}, \boldsymbol{y}) \\
&= \boldsymbol{A}_{\mathrm{cda}}(\boldsymbol{w}_{\mathrm{CDA}}^{(n-1)}, \boldsymbol{\alpha}^{(n)}, \boldsymbol{X}, \boldsymbol{y}) \\
&= \boldsymbol{A}_{\mathrm{cda}}(\boldsymbol{A}_{\mathrm{cda}}(\boldsymbol{w}_{\mathrm{CDB1}}^{(n-1)}, \boldsymbol{\alpha}^{(n-1)}, \boldsymbol{X}, \boldsymbol{y}), \boldsymbol{X}, \boldsymbol{y}) \\
&= \cdots = \left(\prod^n \boldsymbol{A}_{\mathrm{cda}}\right)(\boldsymbol{w}_{\mathrm{CDB1}}^{(1)}, \boldsymbol{\alpha}^{(1)}, \boldsymbol{X}, \boldsymbol{y}) \\
&= \left(\prod^n \boldsymbol{A}_{\mathrm{cda}}\right)(\boldsymbol{w}_{\mathrm{CDB0}}) \\
&= \boldsymbol{w}_{\mathrm{CDB0}} + (\prod^n \boldsymbol{A}_{\mathrm{cda}} - \boldsymbol{I})(\boldsymbol{w}_{\mathrm{CDB0}}) \\
&= \boldsymbol{w}_{\mathrm{CDB0}} + \boldsymbol{C}_1\boldsymbol{w}_{\mathrm{CDB0}} = \boldsymbol{w}_{\mathrm{GD}}
\end{aligned}
\tag{15}
$$

where $\boldsymbol{A}_{\mathrm{CDA}}$ is the CDA rotation operator, $\boldsymbol{C}_1 = (\prod^n \boldsymbol{A}_{\mathrm{cda}} - \boldsymbol{I})$, in which $\boldsymbol{I}$ is the identity operator. This equation matches with Eq. 10 and involves only one correction term, where $\gamma = 1$ because both the CDB0 and CDA are unit vectors and $\boldsymbol{C_1}$ is an overall matrix of high-dimensional rotation. For a given dataset $\boldsymbol{X}, \boldsymbol{y}$ are fixed, it is simplified by: $(\boldsymbol{w}_{\mathrm{CDA}}, \boldsymbol{\alpha}) = \boldsymbol{A}_{\mathrm{cda}}(\boldsymbol{w}_{\mathrm{CDB1}}, \boldsymbol{\alpha}, \boldsymbol{X}, \boldsymbol{y}) = \boldsymbol{A}_{\mathrm{cda}}(\boldsymbol{w}_{\mathrm{CDB1}}, \boldsymbol{\alpha}) : \mathbb{R}^m \times \mathbb{R}^n \to \mathbb{R}^m \times \mathbb{R}^n$, which defines the linear operator that maps the line CDB1 to CDA in each CDA iteration and maps the sample weights $\boldsymbol{\alpha}$ to the updated sample weights.

In Eq. 15, the second equality holds because $\boldsymbol{w}_{\mathrm{CDA}}$ obtained at the end of each rotation is used as $\boldsymbol{w}_{\mathrm{CDB1}}$ in the next rotation. The fifth equality shows that the final CDA line can be written in a form that only depends on variables of the first iteration. This is because, in the first CDA rotation, the initial sample weights $\boldsymbol{\alpha}^{(1)}$ are uniform, and CDB1 is the same as CDB0 of the GDA theoretical framework. Thus, from the sixth equality, $\boldsymbol{X}, \boldsymbol{y}, \boldsymbol{\alpha}^{(1)}$ are omitted. The last three equalities show that the final CDA discriminant is a subcase of the generalized GDA theoretical framework.

The operator $\boldsymbol{A}_{\mathrm{cda}}$ can be decomposed as two sequential operators:

$$\boldsymbol{A}_{\mathrm{cda}} = \boldsymbol{A}_{\mathrm{BO}}\Big(\boldsymbol{w}_{\mathrm{CDB1}}, \boldsymbol{A}_{\mathrm{cdb2}}(\boldsymbol{w}_{\mathrm{CDB1}}, \boldsymbol{A}_{\mathrm{swu}}(\boldsymbol{w}_{\mathrm{CDB1}}, \boldsymbol{\alpha}, \boldsymbol{X}, \boldsymbol{y}), \boldsymbol{X}, \boldsymbol{y}), \boldsymbol{X}, \boldsymbol{y}\Big)$$

$$= \boldsymbol{A}_{\mathrm{BO}}\Big(\boldsymbol{w}_{\mathrm{CDB1}}, \boldsymbol{\alpha}\Big) \tag{16}$$

where $\boldsymbol{A}_{\mathrm{cdb2}}(\boldsymbol{w}_{\mathrm{CDB1}}, \boldsymbol{\alpha}) : \mathbb{R}^m \times \mathbb{R}^n \to \mathbb{R}^m \times \mathbb{R}^n$ is the linear operator that maps CDB1 to CDB2 and updates sample weights by $\boldsymbol{A}_{\mathrm{swu}}(\boldsymbol{w}_{\mathrm{CDB1}}, \boldsymbol{\alpha}) : \mathbb{R}^m \times \mathbb{R}^n \to \mathbb{R}^n$ which corresponds to Appendix C.3 Algorithm 2. $\boldsymbol{A}_{\mathrm{BO}}(\boldsymbol{w}_{\mathrm{CDB1}}, \boldsymbol{\alpha}) : \mathbb{R}^m \times \mathbb{R}^n \to \mathbb{R}^m \times \mathbb{R}^n$ is the linear operator that maps two vectors CDB1 and CDB2 to the BO-estimated discriminant on the plane spanned by CDB1 and CDB2, and updates samples weights.

$\boldsymbol{A}_{\mathrm{cdb2}}$ performs the mapping to CDB2 through the following equation:

$$\tilde{\boldsymbol{w}}_{\mathrm{CDB2}} = \sum_{c=0}^{1} (-1)^{c+1} \frac{\sum_{i=1}^{N} \alpha_i \boldsymbol{x}_i \delta(y_i - c)}{\sum_{i=1}^{N} \alpha_i \delta(y_i - c)} \tag{17}$$

$$\boldsymbol{w}_{\mathrm{CDB2}} = \tilde{\boldsymbol{w}}_{\mathrm{CDB2}}/\|\tilde{\boldsymbol{w}}_{\mathrm{CDB2}}\|_2 \tag{18}$$

The operator $\boldsymbol{A}_{\mathrm{BO}}$ does the mapping to CDA through the following equation:

$$\tilde{\boldsymbol{w}}_{\mathrm{orth}} = \boldsymbol{w}_{\mathrm{CDB2}} - \boldsymbol{w}_{\mathrm{CDB1}} \boldsymbol{w}_{\mathrm{CDB2}}^{\mathrm{T}} \boldsymbol{w}_{\mathrm{CDB1}} \tag{19}$$

$$\boldsymbol{w}_{\mathrm{orth}} = \tilde{\boldsymbol{w}}_{\mathrm{orth}}/\|\tilde{\boldsymbol{w}}_{\mathrm{orth}}\|_2 \tag{20}$$

$$\hat{\theta}^* \approx \arg\max_{\theta} \boldsymbol{A}_{\mathrm{ps}}(\cos\theta \boldsymbol{w}_{\mathrm{CDB1}} + \sin\theta \boldsymbol{w}_{\mathrm{CDB2}}) \tag{21}$$

$$\boldsymbol{w} = (\cos\hat{\theta}^* \boldsymbol{w}_{\mathrm{CDB1}} + \sin\hat{\theta}^* \boldsymbol{w}_{\mathrm{CDB2}}) \tag{22}$$

Where the first two equations find the unit orthogonal vector to CDB1 by the Gram-Schmidt process. The orthogonalization process ensures an efficient search in the space of CDBs during the rotation. The last two equations estimate the optimal rotation angle using Bayesian optimization and the corresponding discriminant.

$ps = \boldsymbol{A}_{\mathrm{ps}}(\boldsymbol{w}_{\mathrm{CDB}}) : \mathbb{R}^m \to \mathbb{R}$ is the operator that outputs the performance score given a classification discriminant, which projects all data onto the discriminant and performs the OOP search of the score.

## G  CLASSIFICATION PERFORMANCE EVALUATION

The datasets used in this study span standard image classification, medical image classification, and chemical property classification, most of which involve multiclass data. To comprehensively assess classification performance across these diverse tasks, we employed four key metrics: AUROC, AUPR, F-score, and AC-score. Accuracy was excluded due to its limitations in reflecting true performance, especially on imbalanced data. Though data augmentation can make data balanced, it introduces uncertainty due to the augmentation strategy and the data quality.

Since we do not prioritize any specific class (as designing a dataset-specific metric is beyond this study's scope), we perform two evaluations—one assuming each class as positive—and take the average. This approach applies to AUPR and F-score, whereas AUROC and AC-score, being symmetric about class labels, require only a single evaluation.

For multiclass prediction, a $C$-dimensional confusion matrix is obtained by comparing predicted labels with true labels. This is converted to $C$ binary confusion matrices by each time taking one class as positive and all the others as negative, and the final evaluation is the average of evaluations on these individual confusion matrices.

To interpret this evaluation scheme, consider the MNIST dataset and the AUPR metric as an example. In binary classification, for the pair of digits "1" and "2", it calculates the sensitivity-recall for detecting "1" and the sensitivity-recall for detecting "2", then takes the average. For other pairs, the calculation follows the same pattern. In the multiclass scenario, it considers the sensitivity-recall for detecting "1" and the sensitivity-recall for detecting "not 1", then takes the average. The same approach applies to other pairs.

INDIVIDUAL METRICS

(A) AUROC

Area Under the Receiver Operating Characteristic curve involves the true positive rate (TPR) and false positive rate (FPR) as follows:

$$AUROC = \int_{x=0}^{1} TPR \times FPR^{-1}(x)dx \tag{23}$$

This measure indicates how well the model distinguishes between classes. A high score means that the model can identify most positive cases with few false positives.

(B) AUPR

Area Under the Precision-Recall curve considers two complementary indices: precision ($prec$) and recall ($rec$), defined as:

$$AUPR = \int_{x=0}^{1} prec \times rec^{-1}(x)dx \tag{24}$$

AUPR measures whether the classifier retrieves most positive cases.

(C) F-SCORE

F-score (using F1-score in this study) is the harmonic mean of precision and recall which contribute equally:

$$F_{score} = 2 \times \frac{prec \times rec}{prec + rec} \tag{25}$$

(D) AC-SCORE

An accuracy-related metric AC-score (Wu & Cannistraci (2025)) addresses the limitations of traditional accuracy for imbalanced data. It imposes a stronger penalty for deviations from optimal performance on imbalanced data and is defined as:

$$AC_{score} = \frac{2 \times TPR \times TNR}{TPR + TNR} \tag{26}$$

This metric assigns equal importance to both classes and penalizes imbalanced performance by ensuring that if one class is poorly classified, the overall score remains low. Thus, AC-score provides a more conservative but reliable evaluation of classifier performance.

### (E) PERFORMANCE-SCORE (PS)

The proposed CDA classifier employs performance-dependent learning, guided by a performance score (ps):

$$ps = (F_{score}^{\mathrm{pos}} + F_{score}^{\mathrm{neg}} + AC_{score})/3 \tag{27}$$

This formulation ensures a balanced consideration of both F-score and AC-score.

Though CDA as a generic classifier utilizes a relatively balanced metric, it can be customized to specific applications as needed.

## H EXPERIMENT ON LINEAR CLASSIFICATION OF LARGE-SCALE DATA

There are real-world problems where the samples are at million-level in addition to plenty of features, posing a significant challenge to train classifiers in reasonable time. One of these problems is omics data classification such as single-cell sequencing data. We collected the 1.3-million-cell mouse brain dataset (10x Genomics (2017)) and took the largest two classes to create a binary classification task (757,526 samples and 27,998 features), with varying number of samples by random subsampling till all samples were included. We compared CDA and fast SVM (dual optimizer), which is one of the most efficient and scalable linear SVM, with regard to performance and speed. We applied CDA with Fibonacci optimizer as the fast version for large data. Original data of sequencing counts were used, and the training and test sets were created using a 4:1 split. The results show that with growing sample sizes, linear CDA outperforms fast SVM not only on classification performance AUROC (Figure 2a and Appendix Figure 6), but also on the single-core training speed (Figure 2b). These results indicate that linear CDA is even more efficient and scalable than the flagship SVM method in efficiency. Hence, CDA as a fast approach has the potential to drive large-scale real-world applications in fields such as biomedicine and autonomous driving.

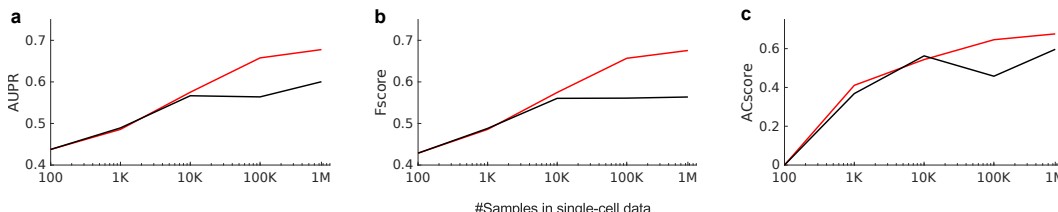

Figure 6: Test set performance with varying sizes of training samples in single-cell mouse brain data, evaluated by (a) AUPR, (b) Fscore and (c) ACscore. CDA: Centroid Discriminant Analysis; SVM: Support Vector Machine; AUPR: Area under precision-recall curve; ACscore: Accuracy score.

## I SUPPLEMENTAL TRAINING SPEED RESULTS

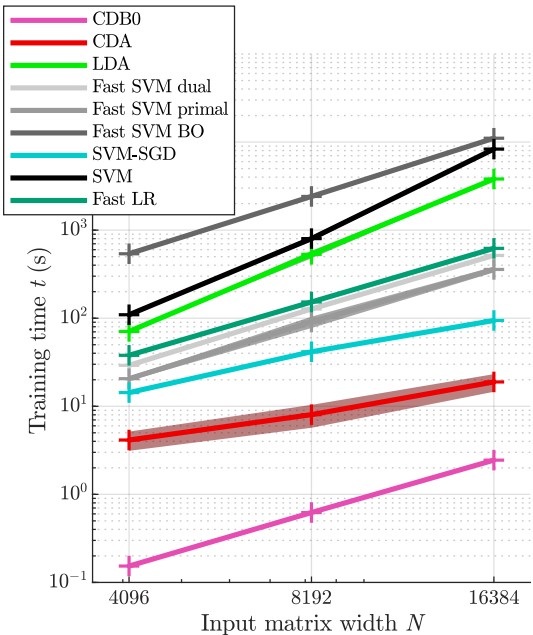

Figure 7: The training time with constant overhead kept. The training time of linear classifiers with increasing input matrix sizes. The number of features were set the same as number of samples $N$. Shaded area represents standard error. Log-scale is used for both axes to reveal the scalability of algorithms. CDB0: Centroid Discriminant Basis 0; CDA: Centroid Discriminant Analysis; LDA: Linear Discriminant Analysis; SVM: Support Vector Machine; BO: Bayesian Optimization; SGD: Stochastic Gradient Descent; LR: Logistic Regression.

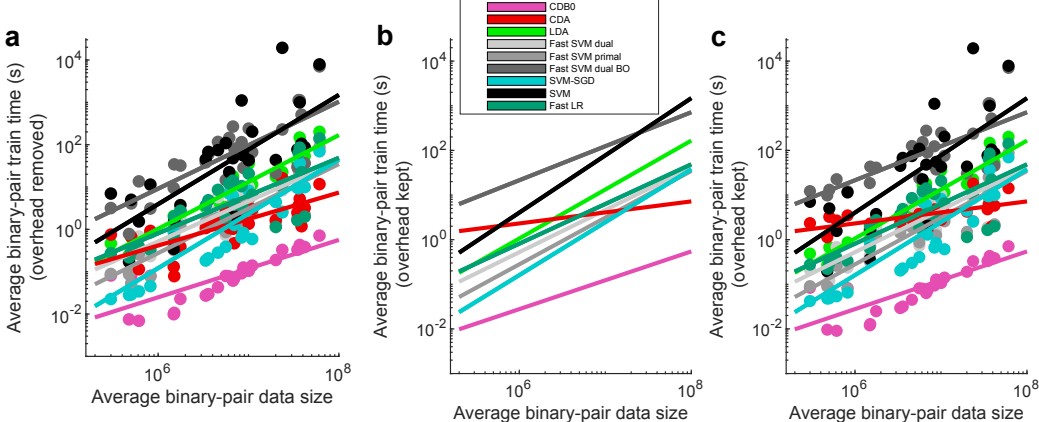

Figure 8: Running time results of linear classifiers on 27 real datasets. (a) Linear regression fit of the running time against data sizes, together with original running time data shown by the circles. Overhead is removed. (b-c) Running time and speed results with overhead kept. (b) Linear regression fit of the running time against data sizes. (c) The same as (b) with original running time data shown by the circles. The log-log scale is applied then linear regression is performed in (a-c) to reflect scaling behavior. CDB0: Centroid Discriminant Basis 0; CDA: Centroid Discriminant Analysis; LDA: Linear Discriminant Analysis; SVM: Support Vector Machine; BO: Bayesian Optimization; SGD: Stochastic Gradient Descent; LR: Logistic Regression.

## J    IMPLEMENTATION DETAILS

All data samples were converted into 1d representations without feature extraction or standardization throughout the paper, as the focus of this study is to compare the general performance of various classifiers on common tasks, rather than optimizing feature extraction for specific applications such as image classification.

### J.1    LINEAR CLASSIFIERS

Linear classifiers compared in this study include LDA, SVM, and fast SVM. Data were input to classifiers in their original values without standardization.

For CDA, the threshold for the coefficient of variation (CV) of performance-score was set to 0.001. 5-fold cross-validation is applied on the OOP search.

For LDA, the MATLAB toolbox function `fitcdiscr` was applied, with `pseudolinear` as the discriminant type, which solves the pseudo-inverse with SVD to adapt to poorly conditioned covariance matrices.

For SVM, the MATLAB toolbox function `fitcsvm` was applied, with sequential minimal optimization (SMO) as the optimizer. The cost function used the L2-regularized (margin) L1-loss (misclassification) form. The misclassification cost coefficient was set to $C = 1$ by default. The maximum number of iterations was linked to the number of training samples in the binary pair as $100 \times N_{\text{train}}$. The convergence criteria were set to the default values.

For fast SVM, the LIBLINEAR MATLAB `mex` function was applied. For the dual form SVM using dual coordinate descent (DCD), the cost function used the L2-regularized (margin) L1-loss (misclassification) form; For the primal form SVM, both regularization and loss are in L2 form. The misclassification cost coefficient was set to $C = 1$ by default. A bias term was added, and correspondingly, the data had an additional dimension with a value of 1. The convergence criteria were set to the default values. Training epochs were set to 300 in general tasks and 100 in large-scale single-cell test.

For SVM-SGD, the passes of all training data are 10; batchsize is 10; learning rate is $1/\sqrt{1 + \max_i \|\boldsymbol{x}_i\|_2}$.

Hyperparameter tuning was performed for fast SVM to achieve the its best performance. CDA, as well as CDB0, do not require parameter tuning.

Speed tests and performance test were from different computational conditions. To obtain classification performance in a reasonable time span, GPU was enabled for SVM, and multicore was enabled for CDA and LDA. To compare speed fairly, all linear classifiers used single-core computing mode.

### J.2    NONLINEAR CLASSIFIERS

For the experiments on the nonlinear kernel-based approaches, on SVHN subset and chemical Clintox, the data were first divided into train+validation and test set with 5:1 and 4:1 ratio respectively, then the train+validation set was further divided into train set and validation set with a 4:1 ratio.

The gaussian kernel parameter sigma was tuned on the validation set by Bayesian optimization with 30 sampling. For the tuned gaussian parameter, on SVHN subset gaussian CDA has $coeff = 0.2$ and gaussian SVM has $coeff = 0.134$; on ClinTox gaussian CDA has $coeff = 1.34$ and gaussian SVM has $coeff = 68.9$, where in gaussian kernel $\sigma^2 = coeff * \text{median}(\text{pairwiseSquareEuclideanDistance}(X_{\text{train}}))$.

For CDA, the threshold for the coefficient of variation (CV) of performance-score was set to 0.001. For SVM, libSVM was implemented with L2-regularized (margin) L1-loss (misclassification) cost function and with default misclassification cost coefficient $C = 1$.

# K  DATASET DESCRIPTION

The datasets tested in this study encompass standard image classification, medical image classification, and chemical property prediction, described in details in Table. 1.

Table 2: Dataset description

| Dataset | #Samples | #Features | #Classes | Balancedness | Modality/source | Classification task |
|---|---|---|---|---|---|---|
| **Standard images** | | | | | | |
| MNIST | 70000 | 400 | 10 | imbalanced | image | digits |
| USPS | 9298 | 256 | 10 | imbalanced | image | digits |
| EMNIST | 145600 | 784 | 26 | balanced | image | letters |
| CIFAR10 | 60000 | 3072 | 10 | balanced | image | objects |
| SVHN | 99289 | 3072 | 10 | imbalanced | image | house numbers |
| flower | 3670 | 1200 | 5 | imbalanced | image | flowers |
| GTSRB | 26635 | 1200 | 43 | imbalanced | image | traffic signs |
| STL10 | 13000 | 2352 | 10 | balanced | image | objects |
| FMNIST | 70000 | 784 | 10 | balanced | image | fashion objects |
| **Medical images** | | | | | | |
| dermamnist | 10015 | 2352 | 7 | imbalanced | dermatoscope | dermal diseases |
| pneumoniamnist | 5856 | 784 | 2 | imbalanced | chest X-Ray | pneumonia |
| retinamnist | 1600 | 2352 | 5 | imbalanced | fundus camera | diabetic retinopathy |
| breastmnist | 780 | 784 | 2 | imbalanced | breast ultrasound | breast diseases |
| bloodmnist | 17092 | 2352 | 8 | imbalanced | blood cell microscope | blood diseases |
| organamnist | 58830 | 784 | 11 | imbalanced | abdominal CT | human organs |
| organcmnist | 23583 | 784 | 11 | imbalanced | abdominal CT | human organs |
| organsmnist | 25211 | 784 | 11 | imbalanced | abdominal CT | human organs |
| organmnist3d | 1472 | 21952 | 11 | imbalanced | abdominal CT | human organs |
| nodulemnist3d | 1633 | 21952 | 2 | imbalanced | chest CT | nodule malignancy |
| fracturemnist3d | 1370 | 21952 | 3 | imbalanced | chest CT | fracture types |
| adrenalmnist3d | 1584 | 21952 | 2 | imbalanced | shape from abdominal CT | adrenal gland mass |
| vesselmnist3d | 1908 | 21952 | 2 | imbalanced | shape from brain MRA | aneurysm |
| synapsemnist3d | 1759 | 21952 | 2 | imbalanced | electron microscope | excitatory/inhibitory |
| **Chemical formula** | | | | | | |
| bace | 1513 | 198 | 2 | imbalanced | chemical formula | BACE1 enzyme |
| BBBP | 2050 | 400 | 2 | imbalanced | chemical formula | blood-brain barrier permeability |
| clintox | 1484 | 339 | 2 | imbalanced | chemical formula | clinical toxicity |
| HIV | 41127 | 575 | 2 | imbalanced | chemical formula | HIV drug activity |
| **Large-scale single-cell sequencing data** | | | | | | |
| Mouse brain | 1306127 | 27998 | 10 | imbalanced | single-cell sequencing | cell type |

**Data processing:** Image data were used in their original value flattened to 1d vectors; chemical formula were processed by simplified molecular input line entry system (SMILES) tokenized encoding. SMILES is a line notation for describing the structure of chemical entities using short ASCII strings. Chemical formula were stacked and aligned from the left since they have different SMILES lengths. The strings were mapped to natural numbers using a predefined dictionary, and those missing string positions were filled with 0. On these converted data we performed different classifiers.

## L    OPTIMIZER COMPARISON FOR CDA ROTATION

In this section we provide an expanded justification and empirical assessment of the one-dimensional (1D) optimizer used within the CDA iterative update. Although Bayesian Optimization (BO) was originally adopted, a 1D optimization problem does not inherently require BO; several classical black-box optimizers can also be applied with low implementation overhead. To rigorously assess the optimizer choice, we conducted a meticulous study comparing BO with five alternative 1D optimizers: (1) random search, (2) grid search, (3) Fibonacci search, (4) Brent's method, and (5) ternary search. All methods were evaluated under identical settings.

Figure L summarizes the comparative performance of CDA under six different 1D optimization strategies across 27 real datasets, reported using AUPR, Fscore, and ACscore. CDA with Fibonacci and BO achieves the most frequent top-ranked positions, further confirming the empirical robustness of the proposed recommendations.

Based on these findings, several practical recommendations emerge. For maximum computational efficiency, Fibonacci search is the preferred choice and provides the most favorable balance between speed and stability. BO remains a competitive secondary option, particularly when log-transformed data are used, as BO tends to exploit smoothness more effectively in this domain. For large-scale datasets, Fibonacci search consistently offers the most robust and efficient solution.

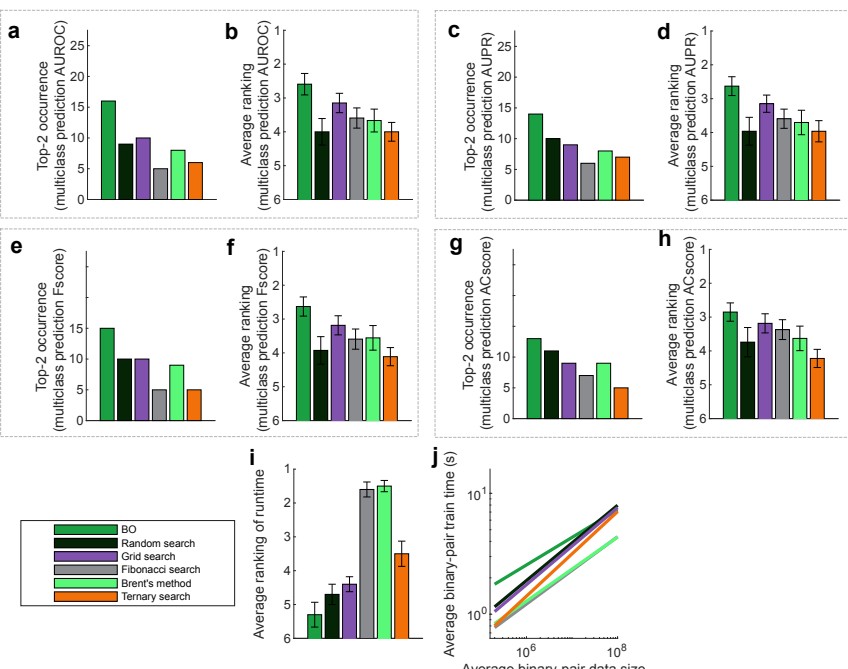

Figure 9: Performance comparison of CDA using six different one-dimensional optimizers: Bayesian Optimization (BO), random search, grid search, Fibonacci search, Brent's method, and ternary search. Panels (a–b) show AUROC, (c–d) AUPR, (e–f) F-score, and (g–h) AC-score, reporting both top-2 occurrences and average ranking of multiclass performance across 27 real datasets. Panels (i–j) show the average ranking of running speed and the corresponding runtime plot. Error bars represent standard errors. CDA: Centroid Discriminant Analysis.

Table 3 reports the results on the same experiments with (i) final CDA performance, (ii) average running time, (iii) average number of CDA iterations, and (iv) average runtime per iteration, as summarized in the table below. Notably, across all optimizers, CDA consistently converges to a similar performance level, demonstrating that CDA algorithm is well-posed and stable, regardless of the choice of 1D optimizer. For runtime per iteration (iv), random search is the fastest due to minimal structural overhead, whereas BO is the slowest because of the cost associated with

maintaining its surrogate model. CDA employs an early-stop strategy, monitoring the stability of training performance across iterations. Among the tested optimizers, Fibonacci search provides the most stable training, yielding the lowest average running time and the fewest CDA iterations.

To further validate this conclusion, CDA with Fibonacci search was applied to a large-scale mouse-brain single-cell dataset containing up to one million samples for multiclass cell-type classification (see Section H). The resulting performance and running time demonstrate that the Fibonacci-based CDA significantly improves computational efficiency while maintaining high predictive accuracy, whereas even optimized SVM implementations require substantially longer runtimes.

The CDA convergence analysis with Fibonacci optimizer is provided in Section O.

Table 3: CDA performance and efficiency across six 1D optimizers for original and log-transformed data.

|  | BO | Random search | Grid search | Fibonacci search | Brent's method | Ternary search |
|---|---|---|---|---|---|---|
| **Original data** | | | | | | |
| Avg. performance (AUROC) | 0.758 | 0.755 | 0.751 | 0.757 | 0.753 | **0.760** |
| Avg. runtime (s) | 4.681 | 4.339 | 4.137 | **2.473** | 2.838 | 3.230 |
| Avg. number of rotations | 36.278 | 46.330 | 43.811 | **29.689** | 29.252 | 35.441 |
| Time per iteration (s) | 0.136 | **0.096** | 0.099 | 0.099 | 0.107 | 0.114 |
| **Log-transformed data** | | | | | | |
| Avg. performance (AUROC) | **0.764** | 0.745 | 0.761 | 0.755 | 0.754 | 0.758 |
| Avg. runtime (s) | 4.693 | 4.313 | 4.003 | **2.503** | 2.546 | 3.355 |
| Avg. number of rotations | 35.796 | 46.330 | 43.507 | 30.244 | **26.178** | 35.759 |
| Time per iteration (s) | 0.134 | **0.095** | 0.098 | 0.098 | 0.113 | 0.108 |

## M Stability of BO in CDA with Initial Few Sampling

To assess the stability of the BO-driven rotation process when only a few initial samples are available, we compared two BO schemes that differ only in the number of initial samples (all other BO hyperparameters and total BO budget were kept the same). In the first BO sampling scheme, the number of sampling is set to min(10, max(4, Iter)) that gradually increase the number of sampling from 4 to 10 according to the current CDA iteration. In the second BO sampling scheme, the number of sampling is set to always 10. The goal is to test whether the early-stage sampling scarcity destabilizes convergence or substantially degrades multiclass performance.

The results in Table 4 show that, across a wide set of image, medical and chemical datasets, performance is essentially unchanged between the two schemes (average AUROC 0.758 vs 0.754). The absolute average difference is 0.004, which is negligible for our tasks and indicates that BO-driven rotations are stable even when the initial sampling budget is small. It is even beneficial to use fewer sampling at initial CDA iterations.

Table 4: Multiclass AUROC for CDA with different BO sampling schemes.

|  | BO with few initial sampling | BO control with high initial sampling |
|---|---|---|
| Average AUROC | **0.758** | 0.754 |

## N    SUPPLEMENTAL PERFORMANCES ON REAL DATASETS OF LINEAR CDA

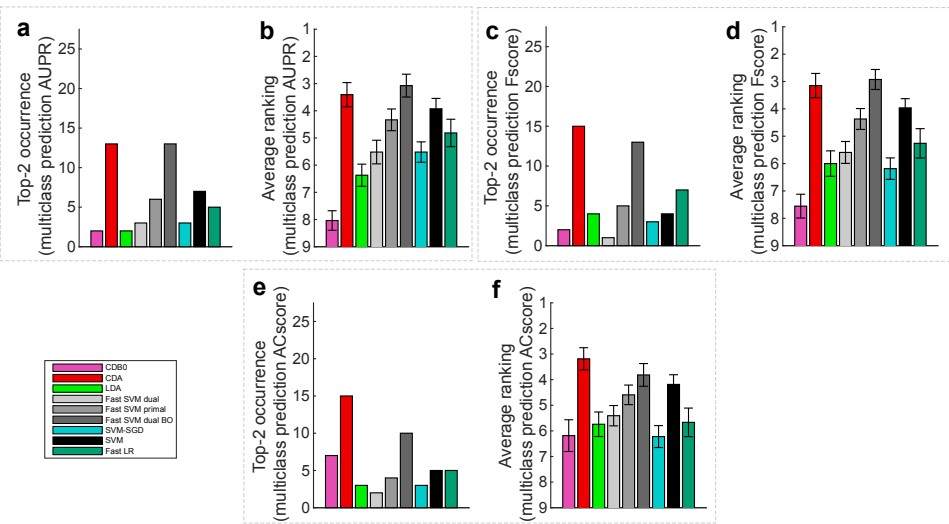

Figure 10: Multiclass prediction performance on 27 real datasets of CDA-BO. Panels (a–b) show AUPR, (c–d) Fscore, (e–f) ACscore, reporting both top-2 occurrences and average ranking of multiclass performance across 27 real datasets. Error bars represent standard errors. CDB0: Centroid Discriminant Basis 0; CDA: Centroid Discriminant Analysis; LDA: Linear Discriminant Analysis; SVM: Support Vector Machine; LR: Logistic Regression; BO: Bayesian Optimization

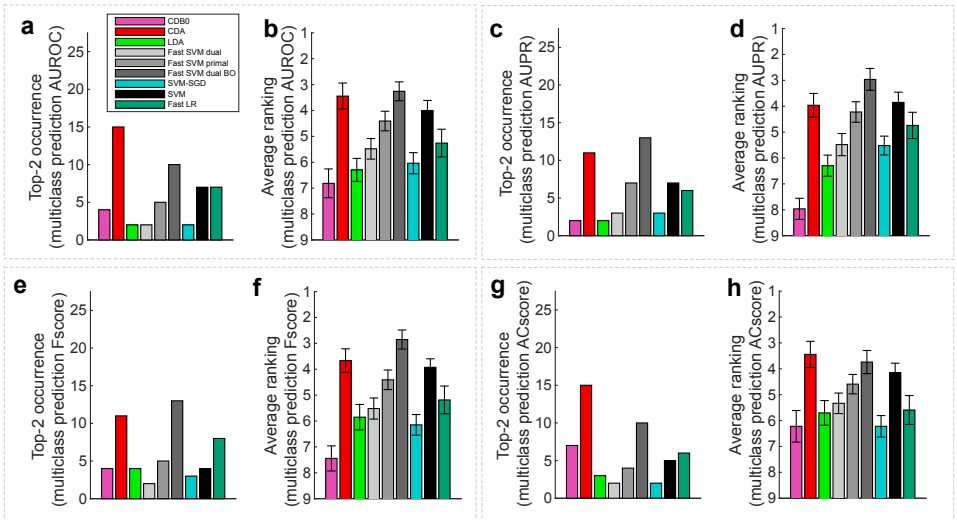

Figure 11: Multiclass prediction performance on 27 real datasets of CDA-Fibonacci. Panels (a–b) show AUROC, (c–d) AUPR, (e–f) F-score, and (g–h) AC-score, reporting both top-2 occurrences and average ranking of multiclass performance across 27 real datasets. Error bars represent standard errors. CDB0: Centroid Discriminant Basis 0; CDA: Centroid Discriminant Analysis; LDA: Linear Discriminant Analysis; SVM: Support Vector Machine; LR: Logistic Regression; BO: Bayesian Optimization

## O    CDA-FIBONACCI CONVERGENCE

We analyzed the relationship between the actual stopping iteration of CDA and the average binary classification performance (ps-score) across different datasets and iteration limits. The results revealed two distinct regimes. In the right half of Fig. 12, for tasks converging before 50 iterations, we observed a significant negative correlation (Pearson's R = –0.099), indicating that datasets with lower performance required more iterations to converge. This highlights the importance of allowing at least 50 iterations, as early stopping before this point may prevent convergence for more challenging tasks. In contrast, for tasks exceeding 50 iterations, the correlation was weak and positive (Pearson's R = 0.089), suggesting that beyond this threshold, the number of iterations no longer plays a significant role in ensuring convergence. This indicates that once 50 iterations are reached, CDA can stabilize regardless of the underlying task performance.

To further validate this observation, in the left half of Fig. 12, we compared the distributions of ps-scores obtained under maximum iteration limits of 50 and 150. The estimated probability densities were identical, and the Wilcoxon signed-rank test confirmed no difference between the two conditions (p=1). This result supports the conclusion that extending the maximum number of iterations beyond 50 does not provide systematic benefits in terms of classification performance. Note that the correlation before 50 in CDA-Fibonacci is weaker than CDA-BO in Figure 3, indicating that the Fibonacci optimizer helps CDA converge faster.

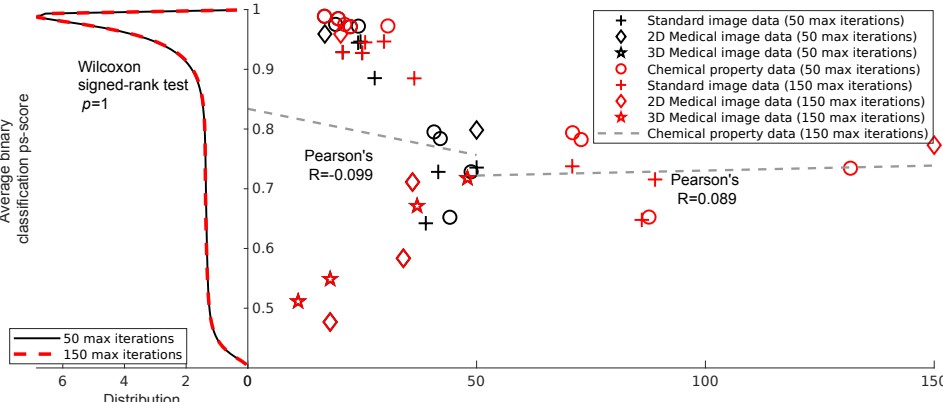

Figure 12: Convergence characterization of CDA-Fibonacci. In the right half, relationship between average binary classification ps-score and the actual stopping iteration of CDA across four dataset categories (standard images, 2D medical images, 3D medical images, and chemical property data) under two maximum iteration limits (50 and 150). A negative correlation was observed for tasks converging before 50 iterations (Pearson's R = –0.099), while a weak positive correlation was found for tasks converging after 50 iterations (Pearson's R = 0.089). The gray dash lines show linear regression of points till 50 and points between 51 and 150. The black vertical line indicates the 50-iteration threshold. In the left half, probability density distributions of ps-scores under 50 and 150 maximum iterations show substantial overlap. The Wilcoxon signed-rank test confirmed no difference (p=1), demonstrating that increasing the maximum iteration limit beyond 50 does not improve classification performance.

## P  LOG-TRANSFORMING THE DATA FOR CDA TO HANDLE OUTLIERS

We performed controlled simulations under three conditions: Gaussian data, elliptic data, and log-transformed elliptic data. Each simulation consisted of 10,000 samples with 10 features, and the covariance structure was specified using a Toeplitz matrix. The distance between group centers was gradually increased to generate different levels of class separability. We included SVM in the comparison, since its margin-based formulation is known to provide a degree of robustness to outliers.

The results shown in Figure 13 indicate that for Gaussian data, CDA performs comparably to LDA and SVM, confirming that it does not lose predictive power under ideal Gaussian conditions. For elliptic distributions, CDA is affected by outliers, as expected. However, applying a log-transform to the features effectively mitigates the influence of extreme values. After log-transforming the elliptic data, the performance of CDA improves substantially, and this preprocessing step benefits CDA more than it benefits LDA or SVM.

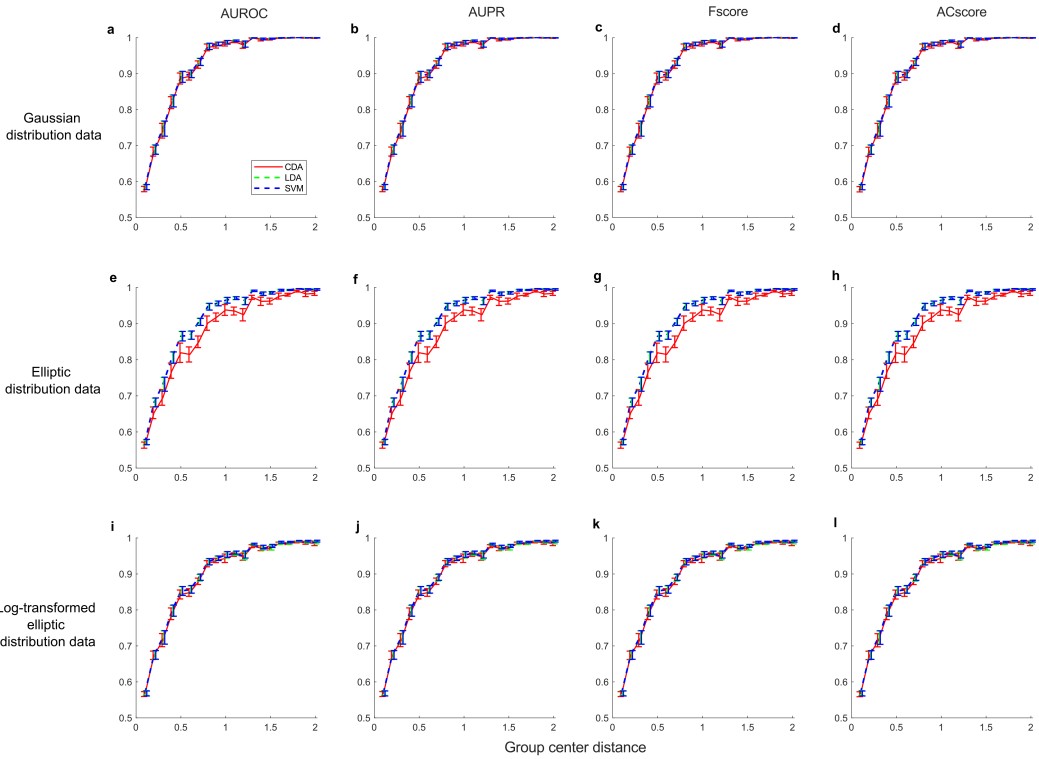

Figure 13: Controlled simulation results comparing CDA, LDA, and SVM under different data distributions. Panels (a–d) correspond to Gaussian data, (e–h) to elliptic data, and (i–l) to log-transformed elliptic data. Within each group, the metrics shown are AUROC (a, e, i), AUPR (b, f, j), F-score (c, g, k), and AC-score (d, h, l). The simulations consist of 10,000 samples with 10 features, and the covariance structure is specified using a Toeplitz matrix. Group center distances were gradually increased to evaluate performance under varying class separability. These results illustrate that CDA maintains comparable performance to LDA and SVM for Gaussian data, is affected by outliers under elliptic distributions, and benefits substantially from log-transforming features.

We further applied the log-transform to 27 real datasets, with results summarized in Table 5. For CDA, both the winning rate (from 0.519 to 0.741) and the average performance increased. For LDA, although the winning rate improved, the average performance decreased, while for SVM, the log-transform led to an overall decline in performance. These results suggest that the log-transform is well suited for variance stabilization and outlier control, and that CDA uniquely benefits from this preprocessing step, yielding enhanced overall performance. Based on these findings, we recommend incorporating a log-transform preprocessing step when applying CDA to data with heavy-tailed or elliptic distributions.

Table 5: Binary classification performance comparison under original and log-transformed data.

|  | CDA | | LDA | | SVM fast | |
|---|---|---|---|---|---|---|
|  | Original | Log | Original | Log | Original | Log |
| Average performance | 0.800 | **0.805** | **0.758** | 0.742 | **0.765** | 0.757 |
| Winning rate | 0.519 | **0.741** | 0.481 | **0.688** | **0.704** | 0.407 |

## Q   PERFORMANCE OF CENTROID-BASED METHODS IN DATASETS WITH MULTIPLE SUBCLASSES

We analyzed a dataset in which each class naturally decomposes into several subclasses, for example, due to illumination changes or rotations. To illustrate this, we selected two classes from the GTSRB dataset and applied Isomap dimensionality reduction (k-nearest neighbors, k=3), marking the ground-truth subclasses in the same color. The resulting embeddings are shown in Figure 14 for the two groups.

We first applied a log-transform to the data to assess the potential impact of outliers. As indicated by the results in Table 6 using CDA with Bayesian Optimization, the log-transform did not produce a significant difference, ruling out outliers as the primary source of variability. Therefore, the challenges in modeling this dataset can be attributed to the presence of multiple subclusters within each class.

We then compared CDA, CDB0, LDA, and SVM on the GTSRB dataset. The results in Table 7 show that the purely centroid-based baseline CDB0 performs poorly (AUROC = 0.589, close to random), confirming that a single centroid is insufficient for classes with multiple subclasses. In contrast, CDA—while still centroid-driven—incorporates adaptive discriminant corrections and achieves AUROC = 0.878, outperforming LDA (AUROC = 0.821). This demonstrates that, among centroid-based methods, CDA is substantially more robust to intra-class multimodality. While all SVM variants achieve AUROC > 0.90, CDA remains competitive, offering a strong balance between robustness and computational efficiency in datasets with multiple subclusters per class.

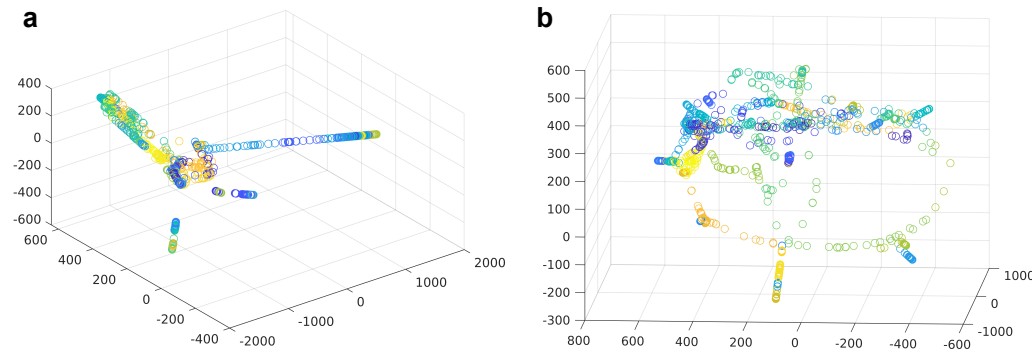

Figure 14: Isomap dimension reduction results of two classes in GTSRB dataset (Pictures of different traffic signs). The same subclasses are marked with the same color according to the ground truth, which shows there are obvious subclusters.

Table 6: Multiclass prediction performance (AUROC) of CDA with Bayesian Optimization on the GTSRB dataset under original and log-transformed data.

| Data | CDA BO |
|---|---|
| Original | 0.878±0.01 |
| Log-transformed | **0.887±0.01** |

Table 7: Multiclass prediction performance (AUROC) on the GTSRB dataset.

| Method | CDB0 | CDA | LDA | SVM |
|---|---|---|---|---|
| AUROC | 0.589±0.01 | 0.878±0.01 | 0.821±0.02 | **0.983±0.003** |

# R   MORE ABOUT LINEAR CDA CLASSIFICATION

While the core CDA algorithm is efficient, the kernelized version inherits the standard computational bottleneck of kernel matrix construction. This bottleneck is not specific to CDA, but is an inherent limitation of all kernel-based linear classifiers. When any linear method (e.g., LDA, SVM) is kernelized, the dominant cost comes from constructing and storing the $N \times N$ kernel matrix, which scales quadratically with the sample size. Thus, the kernelized CDA inherits the standard kernel-matrix bottleneck shared universally by kernel methods, rather than introducing a new one. Although, we provide preliminary results that CDA can be formulated in the nonlinear versions via the kernel trick, we stress that in this paper we focused on linear CDA classification, because linear methods remain widely used in several scientific domains - particularly neuroscience, biomedicine, and social sciences - where explainability, stability, interpretability, and transparent decision boundaries are essential. In many neuroimaging studies, for example, linear classifiers are preferred Shen et al. (2017); Misaki et al. (2010) because they provide direct insight into which brain regions contribute to a behavioral or cognitive prediction, making them a standard analytical tool despite the inherent nonlinearity of the data. To further assess the capability of CDA under nonlinear data distributions, we conducted an additional experiment on a real fMRI-behavior prediction task, where the goal is to classify human subjects into four behavioral score categories (fluid intelligence (gF); 4 groups are low gF, low-middle gF, middle-high gF and high gF). Although the underlying neurobiological patterns are highly nonlinear, this setting reflects a realistic domain where linear models are still favored for interpretability and reproducibility Shen et al. (2017); Misaki et al. (2010). The results (Table 8) show that CDA consistently outperforms all other linear classifiers, including LDA, logistic regression, and several SVM variants. This demonstrates that CDA can better capture discriminative structures even when the data distribution is nonlinear, while maintaining the transparency advantages of linear models.

Table 8: Test set multiclass prediction performance on fMRI-behavior dataset (N=1076, M=64620, 4 classes).

| Method | AUROC | AUPR | Fscore | ACscore |
|---|---|---|---|---|
| CDA | **0.577±0.02** | **0.574±0.02** | **0.571±0.02** | **0.495±0.04** |
| LDA | 0.538±0.02 | 0.558±0.03 | 0.536±0.03 | 0.384±0.07 |
| SVM fast | 0.534±0.005 | 0.536±0.005 | 0.532±0.007 | 0.4±0.04 |
| SVM SGD | 0.544±0.005 | 0.547±0.007 | 0.545±0.007 | 0.424±0.04 |
| SVM | 0.538±0.006 | 0.539±0.007 | 0.538±0.007 | 0.419±0.03 |
| LR fast | 0.537±0.004 | 0.539±0.005 | 0.536±0.006 | 0.4±0.05 |

# S  FULL CLASSIFICATION PERFORMANCE ON REAL DATASETS OF LINEAR CDA

In the below tables we give full classification performance on 27 real datasets based on 4 metrics with binary classification (Table. 9-12) and multiclass prediction (Table. 13-16). In multiclass prediction, the performances on datasets with only two classes were filled with the one of binary classification.

## S.1  BINARY CLASSIFICATION

Table 9: AUROC (Binary Classification Performance)

| Dataset | CDB0 | CDA | LDA | Fast SVM dual | Fast SVM primal | Fast SVM BO | SVM SGD | SVM | Fast LR |
|---|---|---|---|---|---|---|---|---|---|
| **Standard images** | | | | | | | | | |
| MNIST | 0.957±0.004 | 0.985±0.002 | 0.981±0.002 | 0.985±0.002 | **0.986±0.002** | **0.986±0.002** | **0.986±0.002** | **0.986±0.002** | **0.986±0.002** |
| USPS | 0.966±0.005 | 0.989±0.001 | 0.982±0.002 | 0.99±0.002 | 0.99±0.002 | 0.99±0.001 | **0.991±0.001** | 0.99±0.002 | **0.991±0.001** |
| EMNIST | 0.928±0.003 | 0.972±0.001 | 0.964±0.001 | 0.97±0.001 | 0.97±0.001 | **0.973±0.001** | 0.97±0.002 | 0.97±0.001 | 0.97±0.002 |
| CIFAR10 | 0.696±0.01 | 0.797±0.01 | 0.741±0.01 | 0.754±0.01 | 0.784±0.01 | **0.807±0.01** | 0.762±0.01 | 0.787±0.01 | 0.757±0.02 |
| SVHN | 0.528±0.003 | **0.667±0.005** | 0.555±0.003 | 0.55±0.004 | 0.578±0.004 | 0.592±0.005 | 0.537±0.003 | 0.591±0.004 | 0.57±0.008 |
| flower | 0.703±0.02 | 0.739±0.02 | 0.571±0.01 | 0.71±0.03 | 0.705±0.03 | **0.754±0.03** | 0.722±0.03 | 0.71±0.03 | 0.734±0.02 |
| GTSRB | 0.767±0.003 | 0.972±0.001 | 0.942±0.002 | **0.995±0.0004** | 0.995±0.0003 | **0.995±0.0004** | 0.99±0.0006 | **0.995±0.0004** | 0.994±0.0004 |
| STL10 | 0.723±0.02 | 0.781±0.02 | 0.667±0.01 | 0.758±0.02 | 0.757±0.02 | **0.791±0.02** | 0.766±0.02 | 0.761±0.02 | 0.712±0.03 |
| FMNIST | 0.937±0.01 | 0.975±0.006 | 0.973±0.006 | 0.976±0.006 | 0.976±0.006 | **0.978±0.005** | 0.975±0.006 | 0.976±0.006 | 0.976±0.006 |
| **Medical images** | | | | | | | | | |
| dermamnist | 0.682±0.01 | **0.753±0.02** | 0.684±0.02 | 0.676±0.02 | 0.708±0.02 | 0.698±0.02 | 0.608±0.02 | 0.712±0.02 | 0.663±0.03 |
| pneumoniamnist | 0.837±0 | 0.933±0 | 0.912±0 | 0.941±0 | 0.943±0 | 0.941±0 | 0.942±0 | 0.941±0 | **0.944±0** |
| retinamnist | 0.63±0.03 | **0.662±0.04** | 0.616±0.02 | 0.631±0.03 | 0.626±0.02 | 0.632±0.03 | 0.615±0.03 | 0.622±0.03 | 0.619±0.03 |
| breastmnist | 0.66±0 | **0.763±0** | 0.703±0 | 0.726±0 | 0.757±0 | 0.705±0 | 0.734±0 | 0.709±0 | 0.688±0 |
| bloodmnist | 0.89±0.02 | 0.947±0.01 | 0.898±0.02 | 0.951±0.01 | 0.955±0.01 | 0.955±0.01 | 0.946±0.01 | **0.957±0.01** | 0.955±0.01 |
| organamnist | 0.897±0.009 | 0.948±0.008 | 0.95±0.008 | 0.928±0.01 | 0.953±0.008 | **0.958±0.008** | 0.939±0.01 | 0.957±0.008 | 0.954±0.008 |
| organcmnist | 0.89±0.01 | 0.925±0.01 | 0.908±0.01 | 0.895±0.01 | 0.913±0.01 | **0.928±0.01** | 0.911±0.01 | 0.919±0.01 | 0.902±0.02 |
| organsmnist | 0.831±0.01 | 0.886±0.01 | 0.866±0.01 | 0.842±0.02 | 0.871±0.01 | **0.888±0.01** | 0.853±0.02 | 0.88±0.01 | 0.844±0.03 |
| organmnist3d | 0.924±0.01 | 0.957±0.008 | 0.953±0.008 | 0.965±0.007 | 0.966±0.007 | **0.966±0.007** | 0.962±0.007 | **0.966±0.007** | 0.937±0.02 |
| nodulemnist3d | 0.715±0 | **0.781±0** | 0.732±0 | 0.687±0 | 0.702±0 | 0.735±0 | 0.691±0 | 0.724±0 | 0.743±0 |
| fracturemnist3d | **0.671±0.06** | 0.556±0.03 | 0.525±0.04 | 0.576±0.007 | 0.578±0.008 | 0.6±0.004 | 0.592±0.004 | 0.612±0.007 | 0.592±0.01 |
| adrenalmnist3d | 0.653±0 | **0.756±0** | 0.692±0 | 0.697±0 | 0.619±0 | 0.647±0 | 0.637±0 | 0.665±0 | 0.641±0 |
| vesselmnist3d | 0.605±0 | **0.685±0** | 0.681±0 | 0.61±0 | 0.648±0 | 0.628±0 | 0.604±0 | 0.6±0 | 0.584±0 |
| synapsemnist3d | 0.539±0 | **0.544±0** | 0.508±0 | 0.518±0 | 0.527±0 | 0.518±0 | 0.525±0 | 0.539±0 | 0.517±0 |
| **Chemical formula** | | | | | | | | | |
| bace | 0.621±0 | **0.705±0** | 0.684±0 | 0.618±0 | 0.677±0 | 0.697±0 | 0.639±0 | 0.637±0 | 0.693±0 |
| BBBP | 0.711±0 | **0.743±0** | 0.693±0 | 0.667±0 | 0.707±0 | 0.71±0 | 0.646±0 | 0.697±0 | 0.712±0 |
| clintox | **0.65±0** | 0.575±0 | 0.543±0 | 0.517±0 | 0.515±0 | 0.519±0 | 0.508±0 | 0.514±0 | 0.515±0 |
| HIV | 0.6±0 | **0.616±0** | 0.537±0 | 0.51±0 | 0.506±0 | 0.51±0 | 0.505±0 | 0.506±0 | 0.51±0 |

Table 10: AUPR (Binary Classification Performance)

| Dataset | CDB0 | CDA | LDA | Fast SVM dual | Fast SVM primal | Fast SVM BO | SVM SGD | SVM | Fast LR |
|---|---|---|---|---|---|---|---|---|---|
| **Standard images** | | | | | | | | | |
| MNIST | 0.957±0.004 | 0.985±0.002 | 0.981±0.002 | 0.985±0.002 | **0.986±0.002** | **0.986±0.002** | **0.986±0.002** | **0.986±0.002** | **0.986±0.002** |
| USPS | 0.966±0.005 | 0.989±0.001 | 0.983±0.002 | 0.99±0.002 | 0.99±0.002 | 0.99±0.001 | **0.991±0.001** | 0.99±0.002 | **0.991±0.001** |
| EMNIST | 0.928±0.003 | 0.972±0.001 | 0.964±0.001 | 0.97±0.001 | 0.97±0.001 | **0.973±0.001** | 0.97±0.002 | 0.97±0.001 | 0.97±0.002 |
| CIFAR10 | 0.697±0.01 | 0.797±0.01 | 0.741±0.01 | 0.762±0.01 | 0.784±0.01 | **0.807±0.01** | 0.774±0.01 | 0.787±0.01 | 0.757±0.02 |
| SVHN | 0.528±0.003 | **0.682±0.006** | 0.559±0.003 | 0.577±0.005 | 0.604±0.004 | 0.638±0.006 | 0.566±0.005 | 0.634±0.006 | 0.587±0.009 |
| flower | 0.704±0.02 | 0.741±0.02 | 0.571±0.01 | 0.712±0.03 | 0.707±0.03 | **0.759±0.03** | 0.73±0.03 | 0.712±0.03 | 0.736±0.02 |
| GTSRB | 0.757±0.003 | 0.973±0.001 | 0.934±0.002 | 0.995±0.0003 | **0.996±0.0003** | **0.996±0.0003** | 0.991±0.0005 | 0.995±0.0003 | 0.995±0.0003 |
| STL10 | 0.724±0.02 | 0.782±0.02 | 0.667±0.01 | 0.759±0.02 | 0.757±0.02 | **0.791±0.02** | 0.772±0.02 | 0.762±0.02 | 0.713±0.03 |
| FMNIST | 0.937±0.01 | 0.975±0.006 | 0.973±0.006 | 0.976±0.006 | 0.976±0.006 | **0.978±0.005** | 0.975±0.006 | 0.976±0.006 | 0.977±0.006 |
| **Medical images** | | | | | | | | | |
| dermamnist | 0.653±0.01 | 0.743±0.02 | 0.681±0.02 | 0.729±0.02 | 0.746±0.02 | 0.744±0.02 | 0.646±0.03 | **0.752±0.02** | 0.717±0.03 |
| pneumoniamnist | 0.817±0 | 0.931±0 | 0.922±0 | 0.937±0 | 0.944±0 | **0.946±0** | 0.927±0 | 0.945±0 | **0.946±0** |
| retinamnist | 0.614±0.03 | **0.649±0.03** | 0.612±0.02 | 0.641±0.02 | 0.634±0.02 | 0.637±0.03 | 0.622±0.03 | 0.632±0.03 | 0.631±0.03 |
| breastmnist | 0.653±0 | 0.759±0 | 0.69±0 | 0.743±0 | **0.766±0** | 0.718±0 | 0.726±0 | 0.725±0 | 0.736±0 |
| bloodmnist | 0.889±0.02 | 0.947±0.01 | 0.895±0.02 | 0.953±0.01 | 0.955±0.01 | **0.957±0.01** | 0.945±0.01 | **0.957±0.01** | 0.956±0.01 |
| organamnist | 0.902±0.009 | 0.95±0.007 | 0.951±0.008 | 0.929±0.01 | 0.953±0.008 | **0.959±0.008** | 0.941±0.01 | 0.957±0.008 | 0.955±0.008 |
| organcmnist | 0.901±0.009 | 0.931±0.009 | 0.908±0.01 | 0.893±0.01 | 0.911±0.01 | **0.932±0.009** | 0.913±0.01 | 0.918±0.01 | 0.902±0.02 |
| organsmnist | 0.839±0.01 | 0.892±0.01 | 0.867±0.01 | 0.84±0.02 | 0.87±0.01 | **0.894±0.01** | 0.86±0.02 | 0.88±0.01 | 0.845±0.03 |
| organmnist3d | 0.924±0.009 | 0.958±0.008 | 0.954±0.008 | 0.965±0.007 | 0.966±0.007 | **0.967±0.007** | 0.963±0.007 | **0.967±0.007** | 0.937±0.02 |
| nodulemnist3d | 0.7±0 | **0.771±0** | 0.745±0 | 0.695±0 | 0.709±0 | 0.749±0 | 0.711±0 | 0.732±0 | 0.752±0 |
| fracturemnist3d | **0.663±0.05** | 0.566±0.02 | 0.531±0.05 | 0.583±0.003 | 0.586±0.008 | 0.606±0.004 | 0.6±0.006 | 0.618±0.01 | 0.608±0.01 |
| adrenalmnist3d | 0.65±0 | **0.774±0** | 0.705±0 | 0.708±0 | 0.627±0 | 0.663±0 | 0.657±0 | 0.689±0 | 0.665±0 |
| vesselmnist3d | 0.582±0 | 0.671±0 | 0.694±0 | 0.627±0 | **0.7±0** | 0.692±0 | 0.668±0 | 0.646±0 | 0.661±0 |
| synapsemnist3d | 0.537±0 | 0.542±0 | 0.544±0 | 0.533±0 | 0.558±0 | 0.533±0 | 0.538±0 | **0.572±0** | 0.546±0 |
| **Chemical formula** | | | | | | | | | |
| bace | 0.62±0 | **0.704±0** | 0.685±0 | 0.643±0 | 0.679±0 | 0.699±0 | 0.652±0 | 0.64±0 | 0.695±0 |
| BBBP | 0.701±0 | 0.747±0 | 0.734±0 | 0.712±0 | 0.743±0 | **0.751±0** | 0.701±0 | 0.723±0 | 0.742±0 |
| clintox | **0.602±0** | 0.57±0 | 0.548±0 | 0.553±0 | 0.54±0 | 0.575±0 | 0.514±0 | 0.53±0 | 0.54±0 |
| HIV | 0.565±0 | 0.583±0 | 0.558±0 | **0.612±0** | 0.578±0 | 0.584±0 | 0.601±0 | 0.585±0 | 0.596±0 |

Table 11: F-score (Binary Classification Performance)

| Dataset | CDB0 | CDA | LDA | Fast SVM dual | Fast SVM primal | Fast SVM BO | SVM SGD | SVM | Fast LR |
|---|---|---|---|---|---|---|---|---|---|
| **Standard images** | | | | | | | | | |
| MNIST | 0.957±0.004 | 0.985±0.002 | 0.981±0.002 | 0.985±0.002 | **0.986±0.002** | **0.986±0.002** | **0.986±0.002** | **0.986±0.002** | **0.986±0.002** |
| USPS | 0.966±0.005 | 0.989±0.001 | 0.983±0.002 | 0.99±0.002 | 0.99±0.002 | 0.99±0.001 | **0.991±0.001** | 0.99±0.002 | **0.991±0.001** |
| EMNIST | 0.928±0.003 | 0.972±0.001 | 0.964±0.001 | 0.97±0.001 | 0.97±0.001 | **0.973±0.001** | 0.97±0.002 | 0.97±0.001 | 0.97±0.002 |
| CIFAR10 | 0.696±0.01 | 0.797±0.01 | 0.741±0.01 | 0.747±0.01 | 0.784±0.01 | **0.807±0.01** | 0.751±0.02 | 0.787±0.01 | 0.757±0.02 |
| SVHN | 0.523±0.003 | **0.664±0.005** | 0.555±0.003 | 0.51±0.01 | 0.568±0.005 | 0.574±0.007 | 0.471±0.01 | 0.577±0.006 | 0.563±0.009 |
| flower | 0.701±0.02 | 0.738±0.02 | 0.57±0.01 | 0.709±0.03 | 0.705±0.03 | **0.754±0.03** | 0.715±0.03 | 0.709±0.03 | 0.734±0.03 |
| GTSRB | 0.743±0.003 | 0.972±0.001 | 0.931±0.002 | 0.995±0.0003 | **0.996±0.0003** | **0.996±0.0003** | 0.99±0.0005 | 0.995±0.0003 | 0.995±0.0003 |
| STL10 | 0.722±0.02 | 0.781±0.02 | 0.666±0.01 | 0.756±0.02 | 0.756±0.02 | **0.791±0.02** | 0.76±0.02 | 0.761±0.02 | 0.712±0.03 |
| FMNIST | 0.937±0.01 | 0.975±0.006 | 0.973±0.006 | 0.976±0.006 | 0.976±0.006 | **0.978±0.005** | 0.975±0.006 | 0.976±0.006 | 0.976±0.006 |
| **Medical images** | | | | | | | | | |
| dermamnist | 0.621±0.02 | **0.736±0.02** | 0.677±0.02 | 0.682±0.03 | 0.726±0.02 | 0.722±0.02 | 0.595±0.03 | 0.731±0.02 | 0.68±0.04 |
| pneumoniamnist | 0.812±0 | 0.931±0 | 0.921±0 | 0.937±0 | 0.944±0 | 0.945±0 | 0.925±0 | 0.944±0 | **0.946±0** |
| retinamnist | 0.594±0.02 | **0.639±0.03** | 0.61±0.02 | 0.611±0.04 | 0.611±0.04 | 0.631±0.03 | 0.586±0.04 | 0.623±0.03 | 0.619±0.03 |
| breastmnist | 0.651±0 | 0.759±0 | 0.684±0 | 0.739±0 | **0.765±0** | 0.716±0 | 0.725±0 | 0.722±0 | 0.712±0 |
| bloodmnist | 0.888±0.02 | 0.947±0.01 | 0.894±0.02 | 0.952±0.01 | 0.955±0.01 | **0.957±0.01** | 0.943±0.01 | **0.957±0.01** | 0.955±0.01 |
| organamnist | 0.899±0.01 | 0.949±0.008 | 0.951±0.008 | 0.923±0.02 | 0.953±0.008 | **0.959±0.008** | 0.939±0.01 | 0.957±0.008 | 0.955±0.008 |
| organcmnist | 0.896±0.01 | 0.929±0.009 | 0.908±0.01 | 0.892±0.02 | 0.911±0.01 | **0.932±0.009** | 0.911±0.01 | 0.918±0.01 | 0.901±0.02 |
| organsmnist | 0.834±0.01 | 0.89±0.01 | 0.866±0.01 | 0.834±0.02 | 0.869±0.01 | **0.893±0.01** | 0.85±0.02 | 0.88±0.01 | 0.844±0.03 |
| organmnist3d | 0.92±0.01 | 0.957±0.008 | 0.953±0.008 | 0.964±0.007 | **0.966±0.007** | **0.966±0.007** | 0.962±0.007 | **0.966±0.007** | 0.937±0.02 |
| nodulemnist3d | 0.695±0 | **0.769±0** | 0.743±0 | 0.694±0 | 0.708±0 | 0.747±0 | 0.706±0 | 0.731±0 | 0.751±0 |
| fracturemnist3d | **0.651±0.05** | 0.523±0.03 | 0.514±0.05 | 0.577±0.007 | 0.579±0.007 | 0.602±0.003 | 0.594±0.004 | 0.615±0.009 | 0.593±0.01 |
| adrenalmnist3d | 0.65±0 | **0.771±0** | 0.703±0 | 0.707±0 | 0.625±0 | 0.659±0 | 0.651±0 | 0.681±0 | 0.656±0 |
| vesselmnist3d | 0.56±0 | 0.669±0 | **0.693±0** | 0.623±0 | 0.681±0 | 0.663±0 | 0.635±0 | 0.625±0 | 0.611±0 |
| synapsemnist3d | 0.534±0 | **0.539±0** | 0.45±0 | 0.493±0 | 0.501±0 | 0.493±0 | 0.511±0 | 0.522±0 | 0.48±0 |
| **Chemical formula** | | | | | | | | | |
| bace | 0.619±0 | **0.704±0** | 0.684±0 | 0.569±0 | 0.678±0 | 0.698±0 | 0.634±0 | 0.637±0 | 0.694±0 |
| BBBP | 0.699±0 | **0.747±0** | 0.718±0 | 0.691±0 | 0.731±0 | 0.736±0 | 0.669±0 | 0.716±0 | 0.733±0 |
| clintox | 0.54±0 | **0.569±0** | 0.547±0 | 0.517±0 | 0.515±0 | 0.52±0 | 0.506±0 | 0.513±0 | 0.515±0 |
| HIV | 0.531±0 | **0.562±0** | 0.548±0 | 0.511±0 | 0.504±0 | 0.511±0 | 0.501±0 | 0.504±0 | 0.511±0 |

Table 12: AC-score (Binary Classification Performance)

| Dataset | CDB0 | CDA | LDA | Fast SVM dual | Fast SVM primal | Fast SVM BO | SVM SGD | SVM | Fast LR |
|---|---|---|---|---|---|---|---|---|---|
| **Standard images** | | | | | | | | | |
| MNIST | 0.957±0.004 | 0.985±0.002 | 0.981±0.002 | 0.985±0.002 | **0.986±0.002** | **0.986±0.002** | **0.986±0.002** | **0.986±0.002** | **0.986±0.002** |
| USPS | 0.966±0.005 | 0.989±0.001 | 0.982±0.002 | 0.99±0.002 | 0.99±0.002 | 0.99±0.001 | **0.991±0.001** | 0.99±0.002 | **0.991±0.001** |
| EMNIST | 0.927±0.003 | 0.972±0.001 | 0.964±0.001 | 0.97±0.001 | 0.969±0.001 | **0.973±0.001** | 0.969±0.002 | 0.97±0.001 | 0.97±0.002 |
| CIFAR10 | 0.694±0.01 | 0.796±0.01 | 0.74±0.01 | 0.725±0.02 | 0.784±0.01 | **0.807±0.01** | 0.72±0.02 | 0.787±0.01 | 0.756±0.02 |
| SVHN | 0.524±0.002 | **0.614±0.004** | 0.499±0.008 | 0.352±0.03 | 0.444±0.02 | 0.419±0.02 | 0.302±0.03 | 0.438±0.02 | 0.461±0.02 |
| flower | 0.698±0.02 | 0.733±0.02 | 0.567±0.01 | 0.701±0.03 | 0.696±0.03 | **0.74±0.03** | 0.678±0.05 | 0.7±0.03 | 0.725±0.03 |
| GTSRB | 0.753±0.003 | 0.97±0.002 | 0.941±0.002 | **0.995±0.0004** | **0.995±0.0003** | **0.995±0.0004** | 0.99±0.0006 | **0.995±0.0004** | 0.994±0.0004 |
| STL10 | 0.72±0.01 | 0.779±0.02 | 0.664±0.01 | 0.75±0.02 | 0.755±0.02 | **0.79±0.02** | 0.743±0.02 | 0.76±0.02 | 0.711±0.03 |
| FMNIST | 0.936±0.01 | 0.975±0.006 | 0.973±0.006 | 0.975±0.006 | 0.976±0.006 | **0.978±0.005** | 0.975±0.006 | 0.976±0.006 | 0.976±0.006 |
| **Medical images** | | | | | | | | | |
| dermamnist | 0.658±0.02 | **0.72±0.02** | 0.608±0.04 | 0.535±0.05 | 0.602±0.05 | 0.62±0.05 | 0.323±0.07 | 0.606±0.05 | 0.518±0.05 |
| pneumoniamnist | 0.837±0 | 0.932±0 | 0.908±0 | 0.94±0 | 0.942±0 | 0.94±0 | 0.942±0 | 0.94±0 | **0.943±0** |
| retinamnist | 0.62±0.03 | **0.639±0.05** | 0.567±0.03 | 0.513±0.07 | 0.543±0.04 | 0.544±0.05 | 0.448±0.08 | 0.518±0.06 | 0.501±0.06 |
| breastmnist | 0.641±0 | **0.751±0** | 0.698±0 | 0.682±0 | 0.732±0 | 0.658±0 | 0.722±0 | 0.661±0 | 0.59±0 |
| bloodmnist | 0.889±0.02 | 0.946±0.01 | 0.897±0.02 | 0.949±0.01 | 0.954±0.01 | 0.954±0.01 | 0.944±0.01 | **0.956±0.01** | 0.953±0.01 |
| organamnist | 0.892±0.01 | 0.946±0.008 | 0.949±0.008 | 0.919±0.02 | 0.952±0.009 | **0.958±0.008** | 0.935±0.01 | 0.956±0.008 | 0.954±0.008 |
| organcmnist | 0.881±0.01 | 0.92±0.01 | 0.907±0.01 | 0.893±0.02 | 0.913±0.01 | **0.927±0.01** | 0.905±0.01 | 0.919±0.01 | 0.9±0.02 |
| organsmnist | 0.822±0.01 | 0.88±0.01 | 0.862±0.01 | 0.833±0.02 | 0.868±0.01 | **0.884±0.01** | 0.828±0.02 | 0.877±0.01 | 0.839±0.03 |
| organmnist3d | 0.918±0.01 | 0.956±0.008 | 0.952±0.008 | 0.964±0.007 | **0.965±0.007** | **0.965±0.007** | 0.962±0.007 | **0.965±0.007** | 0.937±0.02 |
| nodulemnist3d | 0.707±0 | **0.773±0** | 0.693±0 | 0.636±0 | 0.657±0 | 0.694±0 | 0.623±0 | 0.688±0 | 0.71±0 |
| fracturemnist3d | **0.668±0.06** | 0.38±0.1 | 0.351±0.1 | 0.491±0.06 | 0.491±0.04 | 0.535±0.03 | 0.514±0.03 | 0.562±0.01 | 0.485±0.05 |
| adrenalmnist3d | 0.602±0 | **0.718±0** | 0.631±0 | 0.642±0 | 0.52±0 | 0.553±0 | 0.526±0 | 0.57±0 | 0.528±0 |
| vesselmnist3d | 0.547±0 | **0.615±0** | 0.58±0 | 0.43±0 | 0.488±0 | 0.434±0 | 0.376±0 | 0.375±0 | 0.313±0 |
| synapsemnist3d | 0.498±0 | **0.506±0** | 0.0612±0 | 0.189±0 | 0.19±0 | 0.189±0 | 0.253±0 | 0.239±0 | 0.137±0 |
| **Chemical formula** | | | | | | | | | |
| bace | 0.62±0 | **0.704±0** | 0.678±0 | 0.483±0 | 0.673±0 | 0.694±0 | 0.583±0 | 0.621±0 | 0.688±0 |
| BBBP | 0.693±0 | **0.715±0** | 0.603±0 | 0.553±0 | 0.63±0 | 0.632±0 | 0.501±0 | 0.626±0 | 0.646±0 |
| clintox | **0.634±0** | 0.365±0 | 0.238±0 | 0.0869±0 | 0.0869±0 | 0.0869±0 | 0.0868±0 | 0.0869±0 | 0.0869±0 |
| HIV | **0.471±0** | 0.465±0 | 0.159±0 | 0.0407±0 | 0.0273±0 | 0.0407±0 | 0.0205±0 | 0.0273±0 | 0.0407±0 |

## S.2 MULTICLASS PREDICTION

Table 13: AUROC (Multiclass Prediction Performance)

| Dataset | CDB0 | CDA | LDA | Fast SVM dual | Fast SVM primal | Fast SVM BO | SVM SGD | SVM | Fast LR |
|---|---|---|---|---|---|---|---|---|---|
| **Standard images** | | | | | | | | | |
| MNIST | 0.897±0.01 | 0.963±0.005 | 0.958±0.006 | 0.965±0.005 | 0.965±0.005 | **0.967±0.004** | 0.965±0.005 | 0.966±0.005 | **0.967±0.004** |
| USPS | 0.914±0.01 | 0.971±0.004 | 0.969±0.006 | 0.974±0.005 | 0.974±0.005 | 0.973±0.004 | 0.974±0.004 | 0.973±0.005 | **0.976±0.004** |
| EMNIST | 0.773±0.01 | 0.896±0.008 | 0.879±0.009 | 0.891±0.008 | 0.888±0.008 | **0.898±0.008** | 0.883±0.01 | 0.892±0.008 | 0.878±0.01 |
| CIFAR10 | 0.599±0.02 | 0.671±0.02 | 0.627±0.01 | 0.641±0.02 | 0.661±0.02 | **0.681±0.01** | 0.645±0.02 | 0.663±0.02 | 0.627±0.03 |
| SVHN | 0.522±0.006 | **0.638±0.01** | 0.531±0.007 | 0.536±0.01 | 0.55±0.01 | 0.551±0.01 | 0.534±0.008 | 0.558±0.01 | 0.543±0.01 |
| flower | 0.61±0.03 | 0.666±0.03 | 0.554±0.02 | 0.632±0.03 | 0.629±0.02 | **0.669±0.03** | 0.634±0.03 | 0.632±0.03 | 0.652±0.03 |
| GTSRB | 0.589±0.01 | 0.878±0.01 | 0.821±0.02 | 0.982±0.003 | 0.982±0.003 | 0.982±0.003 | 0.959±0.005 | **0.983±0.003** | 0.972±0.004 |
| STL10 | 0.607±0.02 | 0.655±0.02 | 0.596±0.02 | 0.648±0.02 | 0.648±0.02 | **0.663±0.02** | 0.648±0.03 | 0.653±0.02 | 0.584±0.02 |
| FMNIST | 0.836±0.03 | 0.917±0.02 | 0.92±0.02 | 0.924±0.02 | 0.923±0.02 | **0.927±0.02** | 0.918±0.02 | 0.924±0.02 | 0.924±0.02 |
| **Medical images** | | | | | | | | | |
| dermamnist | 0.614±0.03 | **0.658±0.03** | 0.588±0.02 | 0.595±0.03 | 0.596±0.02 | 0.601±0.03 | 0.531±0.01 | 0.606±0.03 | 0.563±0.02 |
| pneumoniamnist | 0.837±0 | 0.933±0 | 0.912±0 | 0.941±0 | 0.943±0 | 0.941±0 | 0.942±0 | 0.941±0 | **0.944±0** |
| retinamnist | 0.575±0.04 | **0.622±0.04** | 0.592±0.03 | 0.596±0.04 | 0.596±0.04 | 0.608±0.04 | 0.61±0.04 | 0.603±0.04 | 0.596±0.04 |
| breastmnist | 0.66±0 | **0.763±0** | 0.703±0 | 0.726±0 | 0.757±0 | 0.705±0 | 0.734±0 | 0.709±0 | 0.688±0 |
| bloodmnist | 0.789±0.04 | 0.88±0.03 | 0.817±0.03 | 0.882±0.03 | 0.893±0.03 | 0.891±0.03 | 0.877±0.03 | **0.895±0.03** | 0.891±0.03 |
| organamnist | 0.815±0.03 | 0.888±0.02 | 0.885±0.03 | 0.85±0.04 | 0.891±0.03 | **0.9±0.02** | 0.866±0.03 | 0.897±0.02 | 0.893±0.03 |
| organcmnist | 0.809±0.03 | **0.869±0.03** | 0.833±0.03 | 0.811±0.04 | 0.839±0.03 | 0.862±0.03 | 0.833±0.03 | 0.847±0.03 | 0.798±0.05 |
| organsmnist | 0.694±0.03 | **0.761±0.03** | 0.735±0.03 | 0.7±0.03 | 0.733±0.03 | 0.754±0.03 | 0.732±0.04 | 0.746±0.03 | 0.705±0.05 |
| organmnist3d | 0.867±0.03 | 0.913±0.02 | 0.903±0.03 | 0.924±0.02 | **0.93±0.02** | 0.925±0.02 | 0.925±0.02 | 0.925±0.02 | 0.834±0.06 |
| nodulemnist3d | 0.715±0 | **0.781±0** | 0.732±0 | 0.687±0 | 0.702±0 | 0.735±0 | 0.691±0 | 0.724±0 | 0.743±0 |
| fracturemnist3d | **0.622±0.04** | 0.518±0.01 | 0.554±0.04 | 0.574±0.004 | 0.579±0.005 | 0.578±0.01 | 0.575±0.009 | 0.576±0.009 | 0.578±0.01 |
| adrenalmnist3d | 0.653±0 | **0.756±0** | 0.692±0 | 0.697±0 | 0.619±0 | 0.647±0 | 0.637±0 | 0.665±0 | 0.641±0 |
| vesselmnist3d | 0.605±0 | **0.685±0** | 0.681±0 | 0.61±0 | 0.648±0 | 0.628±0 | 0.604±0 | 0.6±0 | 0.584±0 |
| synapsemnist3d | 0.539±0 | **0.544±0** | 0.508±0 | 0.518±0 | 0.527±0 | 0.518±0 | 0.525±0 | 0.539±0 | 0.517±0 |
| **Chemical formula** | | | | | | | | | |
| bace | 0.621±0 | **0.705±0** | 0.684±0 | 0.618±0 | 0.677±0 | 0.697±0 | 0.639±0 | 0.637±0 | 0.693±0 |
| BBBP | 0.711±0 | **0.743±0** | 0.693±0 | 0.667±0 | 0.707±0 | 0.71±0 | 0.646±0 | 0.697±0 | 0.712±0 |
| clintox | **0.65±0** | 0.575±0 | 0.543±0 | 0.517±0 | 0.515±0 | 0.519±0 | 0.508±0 | 0.514±0 | 0.515±0 |
| HIV | 0.6±0 | **0.616±0** | 0.537±0 | 0.51±0 | 0.506±0 | 0.51±0 | 0.505±0 | 0.506±0 | 0.51±0 |

Table 14: AUPR (Multiclass Prediction Performance)

| Dataset | CDB0 | CDA | LDA | Fast SVM dual | Fast SVM primal | Fast SVM BO | SVM SGD | SVM | Fast LR |
|---|---|---|---|---|---|---|---|---|---|
| **Standard images** | | | | | | | | | |
| MNIST | 0.897±0.01 | 0.963±0.004 | 0.958±0.005 | 0.965±0.004 | 0.965±0.004 | **0.967±0.004** | 0.965±0.004 | 0.966±0.004 | **0.967±0.004** |
| USPS | 0.918±0.01 | 0.971±0.005 | 0.969±0.005 | 0.974±0.004 | 0.974±0.005 | 0.974±0.004 | 0.974±0.004 | 0.973±0.005 | **0.976±0.003** |
| EMNIST | 0.774±0.01 | 0.896±0.008 | 0.88±0.009 | 0.891±0.008 | 0.888±0.008 | **0.898±0.008** | 0.884±0.009 | 0.892±0.008 | 0.878±0.01 |
| CIFAR10 | 0.6±0.01 | 0.67±0.01 | 0.627±0.01 | 0.64±0.02 | 0.66±0.01 | **0.68±0.01** | 0.651±0.02 | 0.663±0.02 | 0.615±0.02 |
| SVHN | 0.528±0.009 | **0.666±0.01** | 0.533±0.006 | 0.546±0.005 | 0.572±0.006 | 0.589±0.007 | 0.565±0.01 | 0.597±0.005 | 0.552±0.01 |
| flower | 0.611±0.02 | 0.665±0.02 | 0.554±0.02 | 0.631±0.02 | 0.628±0.02 | **0.671±0.02** | 0.641±0.03 | 0.632±0.02 | 0.652±0.02 |
| GTSRB | 0.619±0.01 | 0.891±0.01 | 0.805±0.01 | 0.981±0.003 | 0.981±0.003 | **0.982±0.002** | 0.959±0.005 | 0.981±0.003 | 0.975±0.003 |
| STL10 | 0.604±0.02 | 0.653±0.02 | 0.598±0.02 | 0.648±0.02 | 0.647±0.02 | **0.663±0.02** | 0.65±0.02 | 0.651±0.02 | 0.585±0.01 |
| FMNIST | 0.836±0.03 | 0.916±0.02 | 0.92±0.02 | 0.923±0.02 | 0.923±0.02 | **0.926±0.02** | 0.918±0.02 | 0.924±0.02 | 0.924±0.02 |
| **Medical images** | | | | | | | | | |
| dermamnist | 0.592±0.02 | 0.645±0.02 | 0.603±0.02 | 0.608±0.03 | 0.633±0.03 | 0.629±0.03 | 0.615±0.03 | **0.649±0.03** | 0.596±0.02 |
| pneumoniamnist | 0.817±0 | 0.931±0 | 0.922±0 | 0.937±0 | 0.944±0 | **0.946±0** | 0.927±0 | 0.945±0 | **0.946±0** |
| retinamnist | 0.568±0.03 | 0.621±0.04 | 0.594±0.03 | 0.589±0.04 | 0.593±0.04 | 0.608±0.04 | **0.632±0.03** | 0.608±0.04 | 0.598±0.04 |
| breastmnist | 0.653±0 | 0.759±0 | 0.69±0 | 0.743±0 | **0.766±0** | 0.718±0 | 0.726±0 | 0.725±0 | 0.736±0 |
| bloodmnist | 0.785±0.04 | 0.878±0.03 | 0.818±0.03 | 0.89±0.03 | 0.895±0.03 | 0.894±0.03 | 0.872±0.03 | **0.897±0.02** | 0.895±0.02 |
| organamnist | 0.82±0.03 | 0.892±0.02 | 0.889±0.02 | 0.851±0.03 | 0.892±0.02 | **0.903±0.02** | 0.871±0.03 | 0.898±0.02 | 0.895±0.02 |
| organcmnist | 0.818±0.03 | **0.877±0.02** | 0.838±0.03 | 0.811±0.04 | 0.839±0.03 | 0.87±0.02 | 0.836±0.03 | 0.847±0.03 | 0.793±0.05 |
| organsmnist | 0.697±0.02 | **0.764±0.03** | 0.741±0.03 | 0.703±0.03 | 0.736±0.03 | 0.763±0.03 | 0.734±0.03 | 0.748±0.03 | 0.698±0.04 |
| organmnist3d | 0.867±0.02 | 0.913±0.02 | 0.904±0.03 | 0.924±0.02 | **0.93±0.02** | 0.926±0.02 | 0.925±0.02 | 0.926±0.02 | 0.82±0.05 |
| nodulemnist3d | 0.7±0 | **0.771±0** | 0.745±0 | 0.695±0 | 0.709±0 | 0.749±0 | 0.711±0 | 0.732±0 | 0.752±0 |
| fracturemnist3d | **0.617±0.04** | 0.526±0.02 | 0.553±0.04 | 0.578±0.004 | 0.583±0.003 | 0.582±0.008 | 0.579±0.007 | 0.579±0.008 | 0.585±0.009 |
| adrenalmnist3d | 0.65±0 | **0.774±0** | 0.705±0 | 0.708±0 | 0.627±0 | 0.663±0 | 0.657±0 | 0.689±0 | 0.665±0 |
| vesselmnist3d | 0.582±0 | 0.671±0 | 0.694±0 | 0.627±0 | **0.7±0** | 0.692±0 | 0.668±0 | 0.646±0 | 0.661±0 |
| synapsemnist3d | 0.537±0 | 0.542±0 | 0.544±0 | 0.533±0 | 0.558±0 | 0.533±0 | 0.538±0 | **0.572±0** | 0.546±0 |
| **Chemical formula** | | | | | | | | | |
| bace | 0.62±0 | **0.704±0** | 0.685±0 | 0.643±0 | 0.679±0 | 0.699±0 | 0.652±0 | 0.64±0 | 0.695±0 |
| BBBP | 0.701±0 | 0.747±0 | 0.734±0 | 0.712±0 | 0.743±0 | **0.751±0** | 0.701±0 | 0.723±0 | 0.742±0 |
| clintox | **0.602±0** | 0.57±0 | 0.548±0 | 0.553±0 | 0.54±0 | 0.575±0 | 0.514±0 | 0.53±0 | 0.54±0 |
| HIV | 0.565±0 | 0.583±0 | 0.558±0 | **0.612±0** | 0.578±0 | 0.584±0 | 0.601±0 | 0.585±0 | 0.596±0 |

Table 15: F-score (Multiclass Prediction Performance)

| Dataset | CDB0 | CDA | LDA | Fast SVM dual | Fast SVM primal | Fast SVM BO | SVM SGD | SVM | Fast LR |
|---|---|---|---|---|---|---|---|---|---|
| **Standard images** | | | | | | | | | |
| MNIST | 0.896±0.01 | 0.963±0.004 | 0.958±0.005 | 0.965±0.004 | 0.965±0.004 | **0.967±0.004** | 0.965±0.004 | 0.966±0.004 | **0.967±0.004** |
| USPS | 0.917±0.01 | 0.971±0.005 | 0.969±0.005 | 0.974±0.004 | 0.973±0.005 | 0.973±0.004 | 0.974±0.004 | 0.973±0.005 | **0.976±0.003** |
| EMNIST | 0.773±0.01 | 0.896±0.008 | 0.879±0.009 | 0.891±0.008 | 0.888±0.008 | **0.898±0.008** | 0.883±0.009 | 0.892±0.008 | 0.878±0.01 |
| CIFAR10 | 0.59±0.01 | 0.67±0.01 | 0.627±0.01 | 0.634±0.02 | 0.66±0.01 | **0.679±0.01** | 0.634±0.02 | 0.663±0.02 | 0.603±0.02 |
| SVHN | 0.509±0.006 | **0.649±0.01** | 0.529±0.005 | 0.526±0.006 | 0.542±0.007 | 0.54±0.01 | 0.521±0.008 | 0.55±0.01 | 0.53±0.009 |
| flower | 0.599±0.02 | 0.662±0.02 | 0.553±0.02 | 0.629±0.02 | 0.626±0.02 | **0.665±0.02** | 0.628±0.03 | 0.63±0.02 | 0.65±0.02 |
| GTSRB | 0.586±0.01 | 0.886±0.01 | 0.782±0.01 | 0.98±0.003 | 0.981±0.003 | **0.982±0.003** | 0.957±0.005 | 0.98±0.003 | 0.974±0.003 |
| STL10 | 0.601±0.02 | 0.653±0.02 | 0.597±0.02 | 0.646±0.02 | 0.646±0.02 | **0.662±0.02** | 0.641±0.02 | 0.651±0.02 | 0.57±0.02 |
| FMNIST | 0.835±0.03 | 0.916±0.02 | 0.919±0.02 | 0.923±0.02 | 0.923±0.02 | **0.926±0.02** | 0.918±0.02 | 0.924±0.02 | 0.924±0.02 |
| **Medical images** | | | | | | | | | |
| dermamnist | 0.571±0.02 | **0.639±0.02** | 0.599±0.02 | 0.602±0.03 | 0.611±0.03 | 0.615±0.03 | 0.533±0.01 | 0.624±0.03 | 0.572±0.02 |
| pneumoniamnist | 0.812±0 | 0.931±0 | 0.921±0 | 0.937±0 | 0.944±0 | 0.945±0 | 0.925±0 | 0.944±0 | **0.946±0** |
| retinamnist | 0.547±0.04 | **0.615±0.04** | 0.592±0.03 | 0.579±0.04 | 0.584±0.04 | 0.604±0.04 | 0.605±0.03 | 0.6±0.04 | 0.591±0.04 |
| breastmnist | 0.651±0 | 0.759±0 | 0.684±0 | 0.739±0 | **0.765±0** | 0.716±0 | 0.725±0 | 0.722±0 | 0.712±0 |
| bloodmnist | 0.779±0.04 | 0.877±0.03 | 0.817±0.03 | 0.887±0.03 | 0.894±0.03 | 0.894±0.03 | 0.869±0.03 | **0.897±0.02** | 0.894±0.03 |
| organamnist | 0.812±0.03 | 0.889±0.02 | 0.889±0.02 | 0.847±0.04 | 0.892±0.02 | **0.903±0.02** | 0.869±0.03 | 0.898±0.02 | 0.894±0.02 |
| organcmnist | 0.811±0.03 | **0.874±0.02** | 0.837±0.03 | 0.809±0.04 | 0.839±0.03 | 0.867±0.02 | 0.834±0.03 | 0.847±0.03 | 0.789±0.05 |
| organsmnist | 0.685±0.02 | **0.76±0.03** | 0.74±0.03 | 0.698±0.03 | 0.736±0.03 | **0.76±0.03** | 0.729±0.04 | 0.748±0.03 | 0.684±0.04 |
| organmnist3d | 0.862±0.02 | 0.913±0.02 | 0.903±0.03 | 0.922±0.02 | **0.929±0.02** | 0.923±0.02 | 0.924±0.02 | 0.923±0.02 | 0.809±0.06 |
| nodulemnist3d | 0.695±0 | **0.769±0** | 0.743±0 | 0.694±0 | 0.708±0 | 0.747±0 | 0.706±0 | 0.731±0 | 0.751±0 |
| fracturemnist3d | **0.601±0.04** | 0.486±0.02 | 0.547±0.04 | 0.575±0.002 | 0.581±0.003 | 0.579±0.008 | 0.575±0.006 | 0.577±0.008 | 0.577±0.01 |
| adrenalmnist3d | 0.65±0 | **0.771±0** | 0.703±0 | 0.707±0 | 0.625±0 | 0.659±0 | 0.651±0 | 0.681±0 | 0.656±0 |
| vesselmnist3d | 0.56±0 | 0.669±0 | **0.693±0** | 0.623±0 | 0.681±0 | 0.663±0 | 0.635±0 | 0.625±0 | 0.611±0 |
| synapsemnist3d | 0.534±0 | **0.539±0** | 0.45±0 | 0.493±0 | 0.501±0 | 0.493±0 | 0.511±0 | 0.522±0 | 0.48±0 |
| **Chemical formula** | | | | | | | | | |
| bace | 0.619±0 | **0.704±0** | 0.684±0 | 0.569±0 | 0.678±0 | 0.698±0 | 0.634±0 | 0.637±0 | 0.694±0 |
| BBBP | 0.699±0 | **0.747±0** | 0.718±0 | 0.691±0 | 0.731±0 | 0.736±0 | 0.669±0 | 0.716±0 | 0.733±0 |
| clintox | 0.54±0 | **0.569±0** | 0.547±0 | 0.517±0 | 0.515±0 | 0.52±0 | 0.506±0 | 0.513±0 | 0.515±0 |
| HIV | 0.531±0 | **0.562±0** | 0.548±0 | 0.511±0 | 0.504±0 | 0.511±0 | 0.501±0 | 0.504±0 | 0.511±0 |

Table 16: AC-score (Multiclass Prediction Performance)

| Dataset | CDB0 | CDA | LDA | Fast SVM dual | Fast SVM primal | Fast SVM BO | SVM SGD | SVM | Fast LR |
|---|---|---|---|---|---|---|---|---|---|
| **Standard images** | | | | | | | | | |
| MNIST | 0.888±0.01 | 0.962±0.005 | 0.956±0.006 | 0.964±0.005 | 0.964±0.005 | **0.966±0.005** | 0.964±0.005 | 0.965±0.005 | **0.966±0.005** |
| USPS | 0.907±0.01 | 0.97±0.005 | 0.968±0.007 | 0.974±0.005 | 0.973±0.005 | 0.973±0.005 | 0.974±0.004 | 0.973±0.005 | **0.975±0.004** |
| EMNIST | 0.708±0.02 | 0.884±0.01 | 0.863±0.01 | 0.878±0.01 | 0.874±0.01 | **0.886±0.01** | 0.867±0.01 | 0.879±0.01 | 0.855±0.02 |
| CIFAR10 | 0.404±0.05 | 0.562±0.03 | 0.48±0.03 | 0.491±0.05 | 0.542±0.03 | **0.579±0.03** | 0.487±0.06 | 0.548±0.03 | 0.431±0.08 |
| SVHN | 0.223±0.05 | **0.478±0.03** | 0.241±0.04 | 0.223±0.06 | 0.242±0.05 | 0.216±0.04 | 0.234±0.06 | 0.247±0.05 | 0.224±0.06 |
| flower | 0.492±0.07 | **0.597±0.05** | 0.417±0.05 | 0.545±0.05 | 0.542±0.04 | 0.591±0.05 | 0.518±0.08 | 0.546±0.05 | 0.576±0.04 |
| GTSRB | 0.296±0.03 | 0.853±0.02 | 0.762±0.03 | 0.981±0.004 | 0.981±0.004 | 0.981±0.004 | 0.956±0.006 | **0.982±0.004** | 0.97±0.004 |
| STL10 | 0.425±0.05 | 0.523±0.05 | 0.413±0.04 | 0.512±0.05 | 0.51±0.05 | **0.541±0.04** | 0.501±0.05 | 0.521±0.05 | 0.364±0.06 |
| FMNIST | 0.801±0.05 | 0.909±0.02 | 0.912±0.02 | 0.916±0.02 | 0.915±0.02 | **0.92±0.02** | 0.91±0.02 | 0.917±0.02 | 0.917±0.02 |
| **Medical images** | | | | | | | | | |
| dermamnist | 0.439±0.09 | **0.527±0.07** | 0.352±0.07 | 0.348±0.1 | 0.324±0.07 | 0.337±0.09 | 0.131±0.04 | 0.346±0.08 | 0.239±0.07 |
| pneumoniamnist | 0.837±0 | 0.932±0 | 0.908±0 | 0.94±0 | 0.942±0 | 0.94±0 | 0.942±0 | 0.94±0 | **0.943±0** |
| retinamnist | 0.377±0.1 | **0.506±0.07** | 0.444±0.08 | 0.401±0.1 | 0.404±0.1 | 0.433±0.1 | 0.422±0.1 | 0.426±0.1 | 0.405±0.1 |
| breastmnist | 0.641±0 | **0.751±0** | 0.698±0 | 0.682±0 | 0.732±0 | 0.658±0 | 0.722±0 | 0.661±0 | 0.59±0 |
| bloodmnist | 0.742±0.05 | 0.864±0.04 | 0.784±0.04 | 0.865±0.03 | 0.88±0.03 | 0.878±0.03 | 0.86±0.04 | **0.883±0.03** | 0.878±0.03 |
| organamnist | 0.771±0.05 | 0.872±0.03 | 0.868±0.03 | 0.814±0.05 | 0.874±0.03 | **0.887±0.03** | 0.84±0.04 | 0.883±0.03 | 0.877±0.03 |
| organcmnist | 0.766±0.04 | **0.848±0.03** | 0.797±0.04 | 0.76±0.05 | 0.804±0.04 | 0.836±0.03 | 0.788±0.05 | 0.816±0.04 | 0.705±0.09 |
| organsmnist | 0.58±0.06 | **0.69±0.05** | 0.653±0.04 | 0.589±0.06 | 0.649±0.05 | 0.677±0.05 | 0.624±0.07 | 0.669±0.05 | 0.54±0.1 |
| organmnist3d | 0.845±0.04 | 0.902±0.03 | 0.89±0.03 | 0.914±0.03 | **0.923±0.02** | 0.916±0.02 | 0.917±0.02 | 0.916±0.02 | 0.736±0.1 |
| nodulemnist3d | 0.707±0 | **0.773±0** | 0.693±0 | 0.636±0 | 0.657±0 | 0.694±0 | 0.623±0 | 0.688±0 | 0.71±0 |
| fracturemnist3d | **0.569±0.09** | 0.279±0.05 | 0.425±0.2 | 0.5±0.07 | 0.508±0.06 | 0.503±0.06 | 0.495±0.06 | 0.505±0.05 | 0.482±0.08 |
| adrenalmnist3d | 0.602±0 | **0.718±0** | 0.631±0 | 0.642±0 | 0.52±0 | 0.553±0 | 0.526±0 | 0.57±0 | 0.528±0 |
| vesselmnist3d | 0.547±0 | **0.615±0** | 0.58±0 | 0.43±0 | 0.488±0 | 0.434±0 | 0.376±0 | 0.375±0 | 0.313±0 |
| synapsemnist3d | 0.498±0 | **0.506±0** | 0.0612±0 | 0.189±0 | 0.19±0 | 0.189±0 | 0.253±0 | 0.239±0 | 0.137±0 |
| **Chemical formula** | | | | | | | | | |
| bace | 0.62±0 | **0.704±0** | 0.678±0 | 0.483±0 | 0.673±0 | 0.694±0 | 0.583±0 | 0.621±0 | 0.688±0 |
| BBBP | 0.693±0 | **0.715±0** | 0.603±0 | 0.553±0 | 0.63±0 | 0.632±0 | 0.501±0 | 0.626±0 | 0.646±0 |
| clintox | **0.634±0** | 0.365±0 | 0.238±0 | 0.0869±0 | 0.0869±0 | 0.0869±0 | 0.0868±0 | 0.0869±0 | 0.0869±0 |
| HIV | **0.471±0** | 0.465±0 | 0.159±0 | 0.0407±0 | 0.0273±0 | 0.0407±0 | 0.0205±0 | 0.0273±0 | 0.0407±0 |

# T    COMPARISON WITH NATIVE MULTICLASS APPROACHES

The training time complexity of CDA for a multiclass problem is on the order of $C^2 * O(NM + N \log N)$, where $C$ is the number of classes, $N$ is the number of samples per class, and $M$ is the feature dimension. However, Since $C$ is typically much smaller than NM, it is negligible, and the overall complexity effectively becomes $C^2 * O(NM + N \log N)$, similar to the binary case.

We think that exploring an inherent multiclass extension of CDA is an important direction, especially given that other classifiers such as LDA and SVM have their inherent multiclass approach. In the cases where $C$ is large and not negligible, inherent multiclass approaches offer attractive lower computational complexity with respect to the number of classes ($C$ versus $C(C-1)/2$ discriminants in pairwise schemes).

In this section, we evaluated inherent multiclass LDA, SVM, and PLS-DA. The results in Table 17 show that although these inherent multiclass methods indeed achieve faster training times, their predictive performance is consistently lower than that obtained by their binary counterparts combined with an external multiclass strategy. This makes sense since these methods are natively designed for binary classification such as the margin in SVM, and their optimization objective becomes less effective when extended to an inherent multiclass setting.

Since we have evidence that the inherent multiclass approaches for LDA and SVM performed worse than their one-versus-one scheme, we decided to leave the investigation the multiclass CDA to future study.

Table 17: AUROC (Multiclass Prediction Performance)

| Dataset | CDB0 | CDA | LDA | Inherent multiclass LDA | SVM | Inherent multiclass SVM | PLSDA | Inherent multiclass PLSDA |
|---|---|---|---|---|---|---|---|---|
| **Standard images** | | | | | | | | |
| MNIST | 0.897±0.01 | 0.963±0.005 | 0.958±0.006 | 0.926±0.009 | **0.966±0.005** | 0.953±0.005 | 0.956±0.006 | 0.913±0.01 |
| USPS | 0.914±0.01 | 0.971±0.004 | 0.969±0.006 | 0.95±0.008 | **0.973±0.005** | 0.967±0.007 | 0.968±0.006 | 0.939±0.01 |
| EMNIST | 0.773±0.01 | 0.896±0.008 | 0.879±0.009 | 0.796±0.01 | 0.892±0.008 | 0.855±0.01 | 0.82±0.02 | 0.544±0.009 |
| CIFAR10 | 0.599±0.02 | 0.671±0.02 | 0.627±0.01 | 0.651±0.02 | 0.663±0.02 | 0.651±0.02 | 0.624±0.01 | 0.647±0.02 |
| SVHN | 0.522±0.006 | **0.638±0.01** | 0.531±0.007 | 0.541±0.007 | 0.558±0.01 | 0.52±0.003 | 0.528±0.007 | 0.532±0.008 |
| flower | 0.61±0.03 | 0.666±0.03 | 0.554±0.02 | 0.618±0.02 | **0.632±0.03** | 0.623±0.02 | 0.507±0.003 | 0.616±0.02 |
| GTSRB | 0.589±0.01 | 0.878±0.01 | 0.821±0.02 | 0.886±0.007 | **0.983±0.003** | 0.974±0.003 | 0.551±0.01 | 0.902±0.01 |
| STL10 | 0.607±0.02 | **0.655±0.02** | 0.596±0.02 | 0.623±0.02 | 0.653±0.02 | 0.625±0.02 | 0.5±0.02 | 0.619±0.02 |
| FMNIST | 0.836±0.03 | 0.917±0.02 | 0.92±0.02 | 0.902±0.02 | **0.924±0.02** | 0.919±0.02 | 0.904±0.02 | 0.9±0.02 |
| **Medical images** | | | | | | | | |
| dermamnist | 0.614±0.03 | **0.658±0.03** | 0.588±0.02 | 0.588±0.02 | 0.606±0.03 | 0.5±0 | 0.518±0.009 | 0.561±0.02 |
| pneumoniamnist | 0.837±0 | 0.933±0 | 0.912±0 | 0.912±0 | **0.941±0** | 0.5±0 | 0.913±0 | 0.91±0 |
| retinamnist | 0.575±0.04 | **0.622±0.04** | 0.592±0.03 | 0.57±0.02 | 0.603±0.04 | 0.5±0 | 0.517±0.009 | 0.5±0.0009 |
| breastmnist | 0.66±0 | **0.763±0** | 0.703±0 | 0.656±0 | 0.709±0 | 0.5±0 | 0.467±0 | 0.523±0 |
| bloodmnist | 0.789±0.04 | 0.88±0.03 | 0.817±0.03 | 0.859±0.03 | **0.895±0.03** | 0.5±0 | 0.716±0.05 | 0.826±0.04 |
| organamnist | 0.815±0.03 | 0.888±0.02 | 0.885±0.03 | 0.851±0.03 | **0.897±0.03** | 0.5±0 | 0.878±0.03 | 0.803±0.04 |
| organcmnist | 0.809±0.03 | **0.869±0.03** | 0.833±0.03 | 0.84±0.03 | 0.847±0.03 | 0.5±0 | 0.827±0.03 | 0.812±0.03 |
| organsmnist | 0.694±0.03 | **0.761±0.03** | 0.735±0.03 | 0.725±0.02 | 0.746±0.03 | 0.5±0 | 0.73±0.03 | 0.684±0.03 |
| organmnist3d | 0.867±0.03 | 0.913±0.02 | 0.903±0.03 | 0.844±0.03 | **0.925±0.02** | 0.5±0 | 0.505±0.008 | 0.498±0.005 |
| nodulemnist3d | 0.715±0 | **0.781±0** | 0.732±0 | 0.722±0 | 0.724±0 | 0.5±0 | 0.508±0 | 0.448±0 |
| fracturemnist3d | **0.622±0.04** | 0.518±0.01 | 0.554±0.04 | 0.52±0.03 | 0.576±0.009 | 0.5±0 | 0.478±0.03 | 0.506±0.002 |
| adrenalmnist3d | 0.653±0 | **0.756±0** | 0.692±0 | 0.671±0 | 0.665±0 | 0.5±0 | 0.51±0 | 0.487±0 |
| vesselmnist3d | 0.605±0 | **0.685±0** | 0.681±0 | 0.712±0 | 0.6±0 | 0.5±0 | 0.521±0 | 0.555±0 |
| synapsemnist3d | 0.539±0 | **0.544±0** | 0.508±0 | 0.53±0 | 0.539±0 | 0.5±0 | 0.515±0 | 0.488±0 |
| **Chemical formula** | | | | | | | | |
| bace | 0.621±0 | **0.705±0** | 0.684±0 | 0.696±0 | 0.637±0 | 0.699±0 | 0.497±0 | 0.508±0 |
| BBBP | 0.711±0 | **0.743±0** | 0.693±0 | 0.691±0 | 0.697±0 | 0.678±0 | 0.506±0 | 0.547±0 |
| clintox | **0.65±0** | 0.575±0 | 0.543±0 | 0.562±0 | 0.514±0 | 0.549±0 | 0.621±0 | 0.584±0 |
| HIV | 0.6±0 | **0.616±0** | 0.537±0 | 0.536±0 | 0.506±0 | 0.511±0 | 0.51±0 | 0.492±0 |

## U    RELATION BETWEEN CDA AND NEURAL NETWORK

### U.1    COMPARISON BETWEEN CDA WITH NEURAL NETWORK

Given that neural networks are prevalent, especially deep learning architectures such as ResNet, we made a comparison between linear CDA and them. Specifically, we compared CDA with ResNet-18, and included a single-hidden-layer MLP as the baseline for the simplest neural network. We tested on 6 3D medical image datasets of MedMNIST. The results are shown in Appendix Table. 18. The parameters of MLP and ResNet-18 are: 100 training epochs; SGDM optimizer; initial learning rate 0.01 with decay of 50% every 30 epochs. For MLP, we used ReLU activation function; the number of hidden-layer neurons was set to approximately $\sqrt{\#\text{samples} \times \#\text{features}}$. We can see that even compared with the more complex architecture ResNet, CDA still outperforms on the adrenalmnist3d dataset. In addition, CDA demonstrates a better performance than MLP, outperforming on 4 out of 6 datasets.

Table 18: Test set multiclass AUROC on MedMNIST3D datasets.

| Method | CDA | MLP | ResNet-18 |
|---|---|---|---|
| organmnist3d | 0.913 | 0.936 | **0.977** |
| nodulemnist3d | 0.781 | 0.766 | **0.850** |
| fracturemnist3d | 0.518 | 0.564 | **0.611** |
| adrenalmnist3d | **0.756** | 0.635 | 0.737 |
| vesselmnist3d | 0.685 | 0.674 | **0.769** |
| synapsemnist3d | 0.544 | 0.507 | **0.656** |

### U.2    COMBINING CDA WITH NEURAL NETWORK

We conducted two experiments using more sophisticated, learned features. First, we used CDA to initialize the final linear layer of a ResNet-18 by taking an ImageNet-pretrained ResNet-18 and performing transfer learning on SVHN. Instead of randomly initializing the final linear layer, we initialized it using class-wise averaged CDA weights. This led to clear improvements over the baseline, yielding higher test accuracy and faster training, as measured by area-across-epochs (AAE). Second, we evaluated Gaussian kernel CDA as a nonlinear model and compared it to a single-hidden-layer MLP (512 input features $\rightarrow$ 45 hidden units $\rightarrow$ 10 output neurons). Because kernel CDA requires kernel matrix computations that scale quadratically with the number of samples, we used the same reduced dataset (24k/99k) employed in our first manuscript. Kernel CDA outperformed the MLP baseline, and a CDA-initialized MLP achieved similar test performance to the randomly initialized MLP while training much faster, as shown by both AAE and the training curve comparison in Figure 15.

These experiments demonstrate that CDA is well-performing and flexible: it can act as a strong initializer in neural networks—whether for a simple linear layer or an MLP—improving both convergence speed and final performance, and it can be extended to kernelized nonlinear variants that outperform standard nonlinear baselines.

When the data is complex with strong nonlinear relations, it is also feasible to combine linear CDA with prevalent deep learning architectures by leveraging deep features. We validated this combined approach on SVHN by extracting features from the penultimate layer of an ImageNet-pretrained ResNet-18 and training linear CDA on these features. For prediction, we used the pretrained ResNet weights together with the trained CDA weights. As shown in Appendix Table 19, this combination significantly outperforms applying linear CDA directly on the raw data, illustrating the strong potential of CDA for handling complex real-world datasets with substantial nonlinear structure.

Table 19: Test set multiclass prediction performance on the SVHN dataset (ResNet-18 pretrained feature extractor + classifier).

| Method | AUROC | AUPR | Fscore | ACscore | Acc | AAE |
|---|---|---|---|---|---|---|
| CDA initialized linear layer | **0.79** | **0.797** | **0.795** | **0.748** | **0.664** | **0.637** |
| Randomly initialized linear layer | 0.78 | 0.795 | 0.789 | 0.73 | 0.653 | 0.621 |
| Gaussian CDA | **0.815** | **0.816** | **0.815** | **0.783** | **0.683** | - |
| CDA initialized MLP | 0.796 | 0.799 | 0.798 | 0.757 | 0.674 | 0.824 |
| Randomly initialized MLP | 0.795 | 0.798 | 0.798 | 0.755 | 0.666 | 0.797 |

Hyperparameters: 150 epochs; Batch size = 128; Initial LR = 0.005 with 50% decay every 40 epochs; L2 regularization = $10^{-4}$. For nonlinear method tests (the last 3), we take a subset=24000/99289 as total data due to time limit to perform kernel method on full dataset

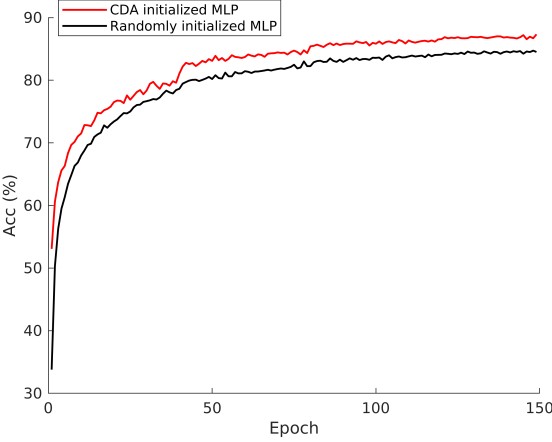

Figure 15: Training curve comparison between CDA initialized MLP and randomly initialized MLP.

## V  COMPUTATION CONDITIONS

The simulations were performed on a Lenovo P620 workstation (a AMD Ryzen Threadripper PRO 3995WX, 64 CPU cores at maximum 4.308GHz, 256GB memory) with single-thread computing. The large-scale tests on single-cell data were performed on a server (two AMD EPYC 7H12 CPUs, each with 64 CPU cores at maximum 2.6GHz, 4096 GB memory) with single-thread computing. MATLAB 2023b was used as the platform to run the main algorithms. The "tic" and "toc" functions in Matlab were used to measure the computation time.

## W  LICENSING

The code packages libSVM and liblinear are with BSD 3-Clause license. Matlab is with a paid proprietary license.

## X  CODE AVAILABILITY

The CDA algorithm has been implemented in both MATLAB and Python, and the code is publicly available at `https://github.com/biomedical-cybernetics/Centroid_discriminant_Analysis`

## Y  CLAIM OF THE LLM USAGE

We used LLM-based tools to improve the language and flow; the principles, core logic, and innovations are entirely the authors'.

