# OpenReview forum: "A Generalized Geometric Theoretical Framework of Centroid Discriminant Analysis for Linear Classification of Multi-dimensional Data"
_ICLR.cc/2026/Conference — ICLR 2026 Poster_

### Official Review · Reviewer_PuwL · 2025-10-26

**Soundness:** 2
**Presentation:** 3
**Contribution:** 2
**Rating:** 4
**Confidence:** 4

**Summary:**

The authors propose a novel theoretical framework named Geometric Discriminant Analysis (GDA), which unifies a family of linear classifiers expressible as functions of a Centroid Discriminant Basis (CDB0)—the line connecting two class centroids—adjusted by geometric corrections under varying constraints. The authors formally demonstrate that Linear Discriminant Analysis (LDA) is a special case of GDA.

**Strengths:**

GDA provides a unified geometric foundation for linear classifiers, explicitly connecting centroid-based discriminants (CDB0) to broader geometric constraints.

 GDA achieves higher accuracy, stability, and efficiency than LDA, SVM, and logistic regression (LR) on 27 real-world datasets.

Kernelized CDa extends linear CDA to nonlinear classification via kernel methods.

**Weaknesses:**

CDA initializes training from CDB0 (the line between centroids), which may be suboptimal if centroids are poorly estimated.

It is crucial to explore the model's extension to multi-class scenarios since traditional LDA is based on multi-class problems.

While kernelized CDA extends to nonlinear problems, its performance depends heavily on kernel choice and bandwidth.

**Questions:**

In equation 10, how to determine various correction terms since  LDA is based on the first-order and second-order information of data.

What are the theoretical explanations for obtaining the bias?

If each class of data has multiple cluster centers, using a single center may not be appropriate.

---

> ### Author Response · Authors · 2025-11-28
>
> **1. Reviewer**: CDA initializes training from CDB0 (the line between centroids), which may be suboptimal if centroids are poorly estimated.
>
> **Reply**: We thank the reviewer for the valuable comment. In many scenarios, the centroids are poorly estimated due to outliers. Therefore, we address the reviewer’s comment by showing how CDA can handle outliers in both synthetic and realistic data.
>
> Using synthetic data, we performed controlled simulations under three conditions: (1) Gaussian data, (2) elliptic data, which is the distribution to describe data with outliers and (3) log-transformed elliptic data. Each simulation consisted of 10,000 samples with 10 features, and the covariance structure was specified using a Toeplitz matrix. The group center distance was gradually increased, and performance was evaluated. We included SVM in the comparison, since its margin-based principle can provide some robustness to outliers.
>
> The results in the figure link below show that for Gaussian data, CDA performs comparably to LDA and SVM, demonstrating that it does not lose significant predictive power under ideal Gaussian conditions. For elliptic distributions, CDA is affected by the presence of outliers, as expected. However, applying a log-transform to the features effectively mitigates the bias introduced by extreme values. After log-transforming the elliptic data, CDA’s performance improves substantially. Interestingly, this preprocessing benefits CDA more than LDA or SVM.
>
> The results for synthetic data are shown in the google drive weblink https://drive.google.com/file/d/1eDVqt0XifI_pNCJd7wsLRu14SXe7z_8o/view?usp=sharing
>
> We further tested log-transform on 27 real datasets with the results shown in Table 1. For CDA we can see that both the winning rate increased from 0.519 to 0.741 and an increase in average performance; while LDA increases winning rate, the average performance drops; for SVM, log-transform leads to underperformance. These results suggest log-transform is adequate for estimating CDA centroids by promoting variance stabilization and outlier control, and CDA can indeed benefit more than other methods for the log-transform to enhance its performance.
>
> Complete table please see weblink: https://docs.google.com/spreadsheets/d/1mR1Yk7_tWpaFRHtKKNLbI8rmRTvgsP6E/edit?usp=sharing
>
> **Table 1. AUROC (Binary prediction performance)**
> | Metric              | CDA Original | CDA Log-transformed | \| | LDA Original | LDA Log-transformed | \| | SVM fast Original | SVM fast Log-transformed |
> |---------------------|--------------|-------------------|---|--------------|-------------------|---|-----------------|-------------------------|
> | Average performance | 0.800        | **0.805**         | \| | **0.758**    | 0.742             | \| | **0.765**       | 0.757                   |
> | Winning rate        | 0.519        | **0.741**         | \| | 0.481        | **0.688**         | \| | **0.704**       | 0.407                   |                   |                   |                   |
>
> Based on these findings, we recommend incorporating a log-transform preprocessing step when applying CDA to data with heavy-tailed or elliptic distributions. We have updated the corresponding section in the revised manuscript to include this recommendation, see Appendix section “Log-transforming the data for CDA to Handle Outliers”.

---

> ### Author Response · Authors · 2025-11-28
>
> **2. Reviewer**: It is crucial to explore the model's extension to multi-class scenarios since traditional LDA is based on multi-class problems.
>
> **Reply**: We thank the Reviewer for this insightful comment. The training time complexity of CDA for a multiclass problem is on the order of $C^2*O(NM+N\log N)$, where $C$ is the number of classes, $N$ is the number of samples per class, and M is the feature dimension. However, Since $C$ is typically much smaller than $NM$, it is neglible, and the overall complexity effectively becomes $O(NM+N\log N)$, similar to the binary case.
>
> Considering the reviewer’s question, we think that exploring an inherent multiclass extension of CDA is an important direction, especially given that other classifiers such as LDA and SVM have their inherent multiclass approach. In the cases where $C$ is large and not negligible, inherent multiclass approaches offer attractive lower computational complexity with respect to the number of classes ($C$ versus $C(C−1)/2$ discriminants in pairwise schemes).
>
> In the current study, we evaluated only binary versions of LDA, SVM, and LR, and relied on the external ECOC framework to handle multiclass prediction. To address the reviewer’s point, we additionally tested inherent multiclass LDA, SVM, and PLS-DA. Our results (Table 2) show that although these inherent multiclass methods indeed achieve faster training times, their predictive performance is consistently lower than that obtained by their binary counterparts combined with an external multiclass strategy. This makes sense since these methods are natively designed for binary classification such as the margin in SVM, and their optimization objective becomes less effective when extended to an inherent multiclass setting.
>
> Since we have evidence that the inherent multiclass approaches for LDA and SVM performed worse than their one-versus-one scheme, we decided to leave the investigation the multiclass CDA to future study. We have updated the native multiclass comparison in the revised manuscript with an Appendix section titled “Comparison with Native Multiclass Approaches”.
>
> For full table please see link https://docs.google.com/spreadsheets/d/1CfwdEf0RJRXb4cunZ4g0pk0l9s-zP_jG/edit?usp=sharing
>
> **Table 2. Average multiclass prediction performance (AUROC) across different classifiers**
> | Metric              | CDB0 | CDA | LDA | Inherent multiclass LDA | Fast SVM | SVM primal | SVM BO | SVM SGD | SVM | Inherent multiclass SVM | PLSDA | Inherent multiclass PLSDA | LR   |
> |---------------------|------|-----|-----|---------------------------|----------|------------|--------|---------|------|---------------------------|--------|------------------------------|------|
> | Average performance | 0.693 | **0.758** | 0.720 | 0.718 | 0.727 | 0.736 | 0.742 | 0.724 | 0.736 | 0.612 | 0.634 | 0.642 | 0.723 |

---

> ### Author Response · Authors · 2025-11-28
>
> **3. Reviewer**: While kernelized CDA extends to nonlinear problems, its performance depends heavily on kernel choice and bandwidth.
>
> **Reply**: We thank the reviewer for raising this point.
>
> Since the dependency on kernel choice and bandwidth indicates more trials are needed to search appropriate ones that lead to longer computation time, we addressed the Reviewer’s concern by adding this part of experiments and results in the revised manuscript with a section in Appendix titled “More about Linear CDA Classification”:
>
> “While the core CDA algorithm is efficient, the kernelized version inherits the standard computational bottleneck of kernel matrix construction. This bottleneck is not specific to CDA, but is an inherent limitation of all kernel-based linear classifiers. When any linear method (e.g., LDA, SVM) is kernelized, the dominant cost comes from constructing and storing the N×N kernel matrix, which scales quadratically with the sample size. Thus, the kernelized CDA inherits the standard kernel-matrix bottleneck shared universally by kernel methods, rather than introducing a new one. Although, we provide preliminary results that CDA can be formulated in the nonlinear versions via the kernel trick, we stress that in this paper we focused on linear CDA classification, because linear methods remain widely used in several scientific domains - particularly neuroscience, biomedicine, and social sciences - where explainability, stability, interpretability, and transparent decision boundaries are essential. In many neuroimaging studies, for example, linear classifiers are preferred [1-2] because they provide direct insight into which brain regions contribute to a behavioral or cognitive prediction, making them a standard analytical tool despite the inherent nonlinearity of the data. To further assess the capability of CDA under nonlinear data distributions, we conducted an additional experiment on a real fMRI-behavior prediction task, where the goal is to classify human subjects into four behavioral score categories (fluid intelligence (gF); 4 groups are low gF, low-middle gF, middle-high gF and high gF). Although the underlying neurobiological patterns are highly nonlinear, this setting reflects a realistic domain where linear models are still favored for interpretability and reproducibility [1-2]. The results (Table 3) show that CDA consistently outperforms all other linear classifiers, including LDA, logistic regression, and several SVM variants. This demonstrates that CDA can better capture discriminative structures even when the data distribution is nonlinear, while maintaining the transparency advantages of linear models.”
>
> **Table 3. Test set multiclass prediction performance on fMRI-behavior dataset (N=1076, M=64620, 4 classes)**
> | Method    | AUROC        | AUPR         | Fscore       | ACscore      |
> |-----------|--------------|--------------|--------------|--------------|
> | CDA       | **0.577±0.02** | **0.574±0.02** | **0.571±0.02** | **0.495±0.04** |
> | LDA       | 0.538±0.02   | 0.558±0.03   | 0.536±0.03   | 0.384±0.07   |
> | SVM fast  | 0.534±0.005  | 0.536±0.005  | 0.532±0.007  | 0.4±0.04     |
> | SVM SGD   | 0.544±0.005  | 0.547±0.007  | 0.545±0.007  | 0.424±0.04   |
> | SVM       | 0.538±0.006  | 0.539±0.007  | 0.538±0.007  | 0.419±0.03   |
> | LR fast   | 0.537±0.004  | 0.539±0.005  | 0.536±0.006  | 0.4±0.05     |
>
> [1] Shen, X., Finn, E., Scheinost, D. et al. Using connectome-based predictive modeling to predict individual behavior from brain connectivity. Nat Protoc 12, 506–518 (2017).
>
> [2] Misaki M., Kim Y., Bandettini P.A., Kriegeskorte N. “Comparison of multivariate classifiers and response normalizations for pattern‑information fMRI.” NeuroImage, 2010; 53(1): 103–118.

---

> ### Author Response · Authors · 2025-11-28
>
> **4. Reviewer**: In equation 10, how to determine various correction terms since LDA is based on the first-order and second-order information of data.
>
> **Reply**: We thank the reviewer for giving us the opportunity to clarify better how LDA is obtained in the geometric discriminant (GD) equation in Eq. 10. In Eq. 10, the generalized geometric framework expresses a linear discriminant as the base centroid vector (CDB0) plus geometric correction terms. In the GDA demonstration we show mathematically that LDA is connected to CDA by only the first and second correction term. To clarify this better, we add a sentence that right after Eq. 10 “in which LDA only involves the first order base term $w_{\rm CDB0}$ and the second order correction $C_1$, while higher order terms $C_2$, $C_3$, … become 0.”, in the Section of “Geometric Discriminant Analysis (GDA)”
>
> Then, we invite the reviewer to Figure 1, which may explain this better. In Figure 1, we show that the there is only the first order base $w_{\rm CDB0}$ and the second order correction $C_1$. All higher order terms $C_2$, $C_3$, etc. go to 0. The first-order term is the centroid difference which defines CDB0. The second-order term contains the variance information of each class and encodes the second-order statistics.
>
> Although LDA only involves two terms, the GDA in Eq. 10 generalizes the discriminants with more possible correction terms. This shows that LDA naturally emerges as a first-order base term and second-order correction of the centroid vector in our GDA framework.
>
> **5. Reviewer**: What are the theoretical explanations for obtaining the bias?
>
> **Reply**: We thank the reviewer for giving us the opportunity to clarify this better. In practice, GDA-based methods—including CDA, LDA, etc. —explicitly normalize the discriminant to unit length before projection. This normalization ensures that the training process refines only the direction of the discriminant, while the bias can always be computed afterwards by placing a threshold between the projected class distributions. For binary classification, the bias may be chosen using a variety of monotonic criteria—e.g., the midpoint between class means, the empirical risk minimizer on the projected line, or a direct search for the threshold optimizing AUROC, accuracy, or any other performance metric. Because projection preserves the ordering of samples, all of these methods yield a valid bias once the discriminant direction is fixed.
>
> This perspective also clarifies that the discriminant and the bias play fundamentally different roles: the discriminant determines the orientation of the separating hyperplane, while the bias determines its position. The former is geometrically meaningful in GDA; the latter can always be adjusted post hoc without altering the interpretation of the discriminant.
>
> We have updated this part of theoretical explanation of discriminant bias in the revised manuscript with Appendix Subsection titled “Explanation of the Discriminant Bias in the GDA Framework”.

---

> ### Author Response · Authors · 2025-11-28
>
> **6. Reviewer**: If each class of data has multiple cluster centers, using a single center may not be appropriate.
>
> **Reply**: We thank the reviewer for highlighting that centroid-based methods may be inappropriate when each class contains multiple cluster centers. To validate this concern, we analyzed a dataset in which each class naturally decomposes into several subclasses (e.g., due to illumination changes and rotations). In the figure link below, we took 2 classes in the GTSRB dataset and applied an Isomap dimension reduction (knn, k=3), in which the ground truths of different subclasses are marked as the same color.
>
> See figure from this link for Isomap dimension reduction for 2 individual groups: https://drive.google.com/file/d/14rj8gle2Vi7G7Q2edNGeRPR057Hapfh1/view?usp=sharing
>
> We first apply log-transform to see if there are outliers in this dataset. From the results shown in Table 4 using CDA BO, we found log-transform does not make significant difference ruling out outliers, therefore, it is not the problem of outliers, and the problem attributes to multiple subclusters.
>
> **Table 4. AUROC (Multiclass prediction performance) on GTSRB**
> | Data            | CDA BO         |
> |-----------------|----------------|
> | Original        | 0.878±0.01     |
> | Log-transformed | **0.887±0.01**     |
>
> Then we compare CDA, CDB0, LDA, and SVM. The GTSRB results are shown in Table 5. As expected, the purely centroid-based baseline CDB0 performs poorly (AUROC = 0.589, close to random), confirming that a single centroid cannot model classes with multiple subclasses. In contrast, CDA - while still centroid-driven - incorporates adaptive discriminant corrections and achieves AUROC = 0.878, outperforming LDA (AUROC = 0.821). This demonstrates that among centroid-based methods, CDA is substantially more robust to intra-class multimodality. As expected, all SVM variants achieve AUROC > 0.90, but importantly, CDA remains competitive while still addressing the challenges posed by multiple subclusters within each class.
>
> **Table 5. AUROC (Multiclass prediction performance) on GTSRB**
> | Method | CDB0 | CDA | LDA | SVM |
> |--------|------|------|------|-------|
> | AUROC | 0.589±0.01 | 0.878±0.01 | 0.821±0.02 | **0.983±0.003** |
>
> To address the reviewer’s concern, we have added a dedicated appendix section titled “Performance of centroid-based methods in datasets with multiple subclasses.”

---

### Official Review · Reviewer_7WhT · 2025-10-28

**Soundness:** 3
**Presentation:** 3
**Contribution:** 2
**Rating:** 6
**Confidence:** 3

**Summary:**

This paper introduces a generalized geometric theoretical framework called Geometric Discriminant Analysis (GDA) for linear classification. GDA unifies certain linear classifiers by defining them as geometric adjustments to a foundational Centroid Discriminant Basis (CDB0), the line connecting the centroids of two classes. Building on this framework, the authors propose a novel, efficient, and scalable linear classifier named Centroid Discriminant Analysis (CDA). CDA starts with CDB0 and iteratively refines the discriminant by performing performance-guided rotations on 2D planes, optimized through Bayesian optimization, achieving a lower quadratic time complexity compared to the cubic complexity of methods like LDA and SVM. The key contributions are the GDA framework itself, the high-performance CDA classifier, and a nonlinear extension of CDA via the kernel method.

**Strengths:**

The paper demonstrates strong originality on two main fronts. Firstly, the introduction of the GDA framework presents a novel conceptual lens for viewing centroid-based linear classifiers. Instead of treating methods like Linear Discriminant Analysis (LDA) and Minimum Distance Classifier (MDC) as disparate algorithms, the paper reframes them as specific instances within a unified system. This is achieved by defining a foundational "Centroid Discriminant Basis" (CDB0) and characterizing different classifiers by the unique geometric corrections they apply to this basis. This unification is a creative and insightful contribution that provides a new language for understanding and designing these models.

The mathematical derivation of the GDA framework is well-executed, particularly in how it deconstructs LDA into the CDB0 and a covariance-based correction term, with clear explanations for various special cases (e.g., equal covariance, no correlation). This provides a solid theoretical foundation for the proposed concepts.

The empirical evaluation is convincing. The authors test CDA against a comprehensive suite of strong baselines (including multiple fast SVM variants, LDA, and LR) across 27 diverse, real-world datasets spanning standard images, medical imaging, and chemical properties. The breadth of this evaluation strongly supports the claim that CDA is a robust and generally applicable classifier. Furthermore, the inclusion of a large-scale experiment on a 1.3-million-cell dataset directly addresses the critical aspect of scalability, providing powerful evidence that CDA's theoretical efficiency advantage (quadratic vs. cubic complexity) translates into significant real-world performance gains. The rigor of the experiments, utilizing multiple metrics and cross-validation, further enhances the quality and credibility of the results.

**Weaknesses:**

* The framework is primarily used in a descriptive capacity to show that LDA is a special case. The paper proposes that GDA "may inspire the design of new classifiers," but it stops short of demonstrating how. CDA itself is presented as a single, highly effective derivative, but the framework's utility in systematically generating other novel classifiers is not shown.

* Rationale for Bayesian Optimization (BO). The algorithm uses BO to perform a 1D search for the optimal rotation angle on each 2D plane. While BO is a powerful technique, its use for a simple 1D optimization problem is not justified. Given that the performance-score function is relatively inexpensive to evaluate for a single vector, simpler methods might have been sufficient and faster.

* The section on nonlinear kernel CDA feels underdeveloped compared to the thoroughness of the linear experiments. While the results on two datasets are promising, they are not sufficient to make a strong claim about its general effectiveness.

**Questions:**

* The GDA framework is presented as a unifying concept. However, a key claim is that the framework can "inspire the design of new classifiers." Please elaborate on how to use the GDA framework to design a new classifier systematically. A concrete, hypothetical example would greatly strengthen the claim of GDA's generative utility.

* The use of Bayesian Optimization (BO) for the 1D search of the optimal rotation angle on each 2D plane is a notable design choice. Could you provide a more detailed rationale for this? Given that the performance-score evaluation for a single vector is computationally inexpensive, simpler methods like a fine-grained grid search might seem sufficient and potentially faster in terms of overhead.

* The preliminary results for the nonlinear kernel CDA are promising but are limited to two datasets. Could you please provide more experimental evidence about this claim?

---

> ### Author Response · Authors · 2025-11-28
>
> **1. Reviewer**: The framework is primarily used in a descriptive capacity to show that LDA is a special case. The paper proposes that GDA "may inspire the design of new classifiers," but it stops short of demonstrating how. CDA itself is presented as a single, highly effective derivative, but the framework's utility in systematically generating other novel classifiers is not shown.
>
> **Reply**: We thank the Reviewer for this important point. We agree with the Reviewer and we added in the limitations the following sentence: “In this study, the GDA framework is used to show that LDA is a special centroid-based method case, and then is adopted to design CDA. CDA itself is presented as a single, highly effective derivative, but the GDA framework's utility in systematically generating other novel classifiers is not investigated enough. The mission to investigate the extent to which GDA framework may inspire the design of new classifiers will be a not trivial point to expand in future studies.”

---

> ### Author Response · Authors · 2025-11-28
>
> **2. Reviewer**: Rationale for Bayesian Optimization (BO). The algorithm uses BO to perform a 1D search for the optimal rotation angle on each 2D plane. While BO is a powerful technique, its use for a simple 1D optimization problem is not justified. Given that the performance-score function is relatively inexpensive to evaluate for a single vector, simpler methods might have been sufficient and faster.
>
> **Reply**: We thank the Reviewer for raising this important question regarding the necessity of using Bayesian Optimization (BO) for a one-dimensional search. We fully agree that, in principle, a 1D optimization problem does not inherently require BO, and that simpler methods could be sufficient. Because this choice is critical for CDA’s efficiency and robustness, we conducted a meticulous study comparing BO with five alternative 1D optimizers commonly used for black-box functions: 1) Random search 2) grid search 3) Fibonacci search 4) Brent’s method and 5) ternary search.
>
> We evaluated all optimizers in terms of (i) final CDA performance (ii) average running time (iii) average number of CDA iterations and (iv) average runtime per iteration, reported in the table below. Importantly, across all six optimizers, CDA consistently converges to a similar level of performance, demonstrating that the CDA algorithm is well-posed and stable, regardless of the choice of 1D optimizer. For average runtime per iteration in (iv), as expected, random search is the fastest due to zero structural overhead, whereas BO is the slowest due to the cost of maintaining its surrogate model. In CDA we applied an early-stop strategy by checking if the training performance across iteration is stable. We found that the results of (ii) and (iii) show that Fibonacci search help to stabilize the training process, with the least average running time and average number of CDA iterations.
>
> Based on our findings, we make the following recommendations. For maximum speed, Fibonacci search is the preferred choice, and BO is a strong second option. Be aware that BO seems to prefer log-transform of the data. For large-scale datasets, Fibonacci search remains the most efficient and reliable method.
>
> These results confirm that BO is not required for CDA to function correctly, but it remains a competitive choice under certain data conditions. We have added these comparisons and guidelines to the revised manuscript to clarify the rationale behind the optimizer selection.
>
> Please see full tables in:
>
> https://docs.google.com/spreadsheets/d/1qFeURphzLeBOF_YpDWCbos2ZlFz1uwep/edit?usp=sharing (original data)
>
> https://docs.google.com/spreadsheets/d/1cg7-EntqAsJRocYdkQtAlpKmtERmGmzH/edit?usp=sharing (log-transformed data)
>
> **Table 1. Comparison of Optimization Methods on Original and Log-Transformed Data**
> | Category | Metric | BO | Random search | Grid search | Fibonacci search | Brent's method | Ternary search |
> |----------|---------|------|----------------|--------------|-------------------|-----------------|----------------|
> | Original | Avg. performance (AUROC) | 0.758 | 0.755 | 0.751 | 0.757 | 0.753 | **0.760** |
> |          | Avg. runtime (s) | 4.681 | 4.339 | 4.137 | **2.473** | 2.838 | 3.230 |
> |          | Avg. number of rotations | 36.278 | 46.330 | 43.811 | 29.689 | **29.252** | 35.441 |
> |          | Time per iteration (s) (Avg. runtime / Avg. number of rotations) | 0.136 | **0.096** | 0.099 | 0.099 | 0.107 | 0.114 |
> | Log-transformed | Avg. performance (AUROC) | **0.764** | 0.745 | 0.761 | 0.755 | 0.754 | 0.758 |
> |                | Avg. runtime (s) | 4.693 | 4.313 | 4.003 | **2.503** | 2.546 | 3.355 |
> |                | Avg. number of rotations | 35.796 | 46.330 | 43.507 | 30.244 | **26.178** | 35.759 |
> |                | Time per iteration (s) (Avg. runtime / Avg. number of rotations) | 0.134 | **0.095** | 0.098 | 0.098 | 0.113 | 0.108 |
>
> Moreover, we applied CDA-Fibonacci for the large-scale mouse brain single-cell dataset to classifier cell types up to 1 million samples, and record the performance and running time. See results in the figure link below. Indeed, Fibonacci further enhances CDA efficiency for large-scale application. In contrast, SVM fast requires significantly longer running time.
>
> See figure with this link: https://drive.google.com/file/d/1dCq762QTiZaY2iX1VehVm6VvOEO7ZFgt/view?usp=sharing
>
> We updated these experiments comparing different optimizers with a section in Appendix titled “Optimizer Comparison for CDA Rotation”.

---

> ### Author Response · Authors · 2025-11-28
>
> **3. Reviewer**: The section on nonlinear kernel CDA feels underdeveloped compared to the thoroughness of the linear experiments. While the results on two datasets are promising, they are not sufficient to make a strong claim about its general effectiveness.
>
> **Reply (1/3)**: We thank the reviewer for raising this point.
>
> First, we addressed the Reviewer’s concern by adding this part of experiments and results in the revised manuscript with a section in Appendix titled “More about Linear CDA Classification”:
>
> “Although, we provide preliminary results that CDA can be formulated in the nonlinear versions via the kernel trick, we stress that in this paper we focused on linear CDA classification, because linear methods remain widely used in several scientific domains - particularly neuroscience, biomedicine, and social sciences - where explainability, stability, interpretability, and transparent decision boundaries are essential. In many neuroimaging studies, for example, linear classifiers are preferred [1-2] because they provide direct insight into which brain regions contribute to a behavioral or cognitive prediction, making them a standard analytical tool despite the inherent nonlinearity of the data. To further assess the capability of CDA under nonlinear data distributions, we conducted an additional experiment on a real fMRI-behavior prediction task, where the goal is to classify human subjects into four behavioral score categories (fluid intelligence (gF); 4 groups are low gF, low-middle gF, middle-high gF and high gF). Although the underlying neurobiological patterns are highly nonlinear, this setting reflects a realistic domain where linear models are still favored for interpretability and reproducibility [1-2]. The results (Table 2) show that CDA consistently outperforms all other linear classifiers, including LDA, logistic regression, and several SVM variants. This demonstrates that CDA can better capture discriminative structures even when the data distribution is nonlinear, while maintaining the transparency advantages of linear models.”
>
> **Table 2. Test set multiclass prediction performance on fMRI-behavior dataset (N=1076, M=64620, 4 classes)**
> | Method    | AUROC        | AUPR         | Fscore       | ACscore      |
> |-----------|--------------|--------------|--------------|--------------|
> | CDA       | **0.577±0.02** | **0.574±0.02** | **0.571±0.02** | **0.495±0.04** |
> | LDA       | 0.538±0.02   | 0.558±0.03   | 0.536±0.03   | 0.384±0.07   |
> | SVM fast  | 0.534±0.005  | 0.536±0.005  | 0.532±0.007  | 0.4±0.04     |
> | SVM SGD   | 0.544±0.005  | 0.547±0.007  | 0.545±0.007  | 0.424±0.04   |
> | SVM       | 0.538±0.006  | 0.539±0.007  | 0.538±0.007  | 0.419±0.03   |
> | LR fast   | 0.537±0.004  | 0.539±0.005  | 0.536±0.006  | 0.4±0.05     |
>
> [1] Shen, X., Finn, E., Scheinost, D. et al. Using connectome-based predictive modeling to predict individual behavior from brain connectivity. Nat Protoc 12, 506–518 (2017).
>
> [2] Misaki M., Kim Y., Bandettini P.A., Kriegeskorte N. “Comparison of multivariate classifiers and response normalizations for pattern‑information fMRI.” NeuroImage, 2010; 53(1): 103–118.

---

> ### Author Response · Authors · 2025-11-28
>
> **3. Reviewer**: The section on nonlinear kernel CDA feels underdeveloped compared to the thoroughness of the linear experiments. While the results on two datasets are promising, they are not sufficient to make a strong claim about its general effectiveness.
>
> **(Continued) Reply (2/3)**: Second, in response to the reviewer’s suggestion, we conducted experiments with kernel LDA and report the results in the table below. Addressing the request of the Reviewer, we also included a new tested dataset - fracturemnist 3d medical image dataset, in which linear CDA performance is low. The results in Table 3 show that kernel CDA outperforms on 2 out of 3 datasets, while the performance gap compared to the best one (nLDA) is not large. Importantly, kernel CDA improves on linear CDA substantially, suggesting the potential of applying kernel methods on linear CDA for complex data. These results have now been incorporated into Table 1 of the revised manuscript.
>
> **Table 3. Test set multiclass performance**
> | Dataset                | Method | AUROC         | AUPR          | Fscore        | ACscore       |
> |------------------------|--------|---------------|---------------|---------------|---------------|
> | SVHN subset (image)    | CDA    | 0.615±0.02    | 0.63±0.02     | 0.619±0.02    | 0.423±0.05    |
> |                        | nCDA   | 0.777±0.01    | 0.782±0.01    | 0.78±0.01     | 0.731±0.02    |
> |                        | SVM    | 0.555±0.01    | 0.568±0.007   | 0.551±0.006   | 0.273±0.05    |
> |                        | nSVM   | 0.736±0.02    | 0.776±0.009   | 0.756±0.008   | 0.654±0.03    |
> |                        | nLDA | **0.786±0.01** | **0.79±0.01** | **0.789±0.01** | **0.743±0.02** |
> | ClinTox (chemical)     | CDA    | 0.567         | 0.561         | 0.56          | 0.351         |
> |                        | nCDA | **0.625**      | **0.627**      | **0.627**      | **0.46**      |
> |                        | SVM    | 0.565         | 0.578         | 0.575         | 0.294         |
> |                        | nSVM   | 0.5           | 0.481         | 0.48          | 0             |
> |                        | nLDA   | 0.605         | 0.612         | 0.611         | 0.409         |
> | Fracture 3d (medical image) | CDA    | 0.518±0.01    | 0.526±0.02    | 0.486±0.02    | 0.279±0.05    |
> |                        | nCDA | **0.625±0.04** | 0.62±0.04 | **0.607±0.04** | **0.577±0.08** |
> |                        | SVM    | 0.576±0.009   | 0.579±0.008   | 0.577±0.008   | 0.505±0.05    |
> |                        | nSVM   | 0.608±0.06    | 0.591±0.07    | 0.586±0.07    | 0.44±0.2      |
> |                        | nLDA   | 0.608±0.03    | **0.626±0.02**    | 0.605±0.02    | 0.491±0.1     |

---

> ### Author Response · Authors · 2025-11-28
>
> **3. Reviewer**: The section on nonlinear kernel CDA feels underdeveloped compared to the thoroughness of the linear experiments. While the results on two datasets are promising, they are not sufficient to make a strong claim about its general effectiveness.
>
> **(Continued) Reply (3/3)**: Third, we added an experiment of kernel CDA on SVHN features extracted by a pretrained ResNet-18. We evaluate kernel CDA (Gaussian) as a nonlinear model and compare it to a single-hidden-layer MLP. Since kernel CDA involves computing a kernel matrix in quadratic to the number of samples, and due to the time limit we are not able to run the full dataset, we use the same subset of data (24k/99k) that we used in our first manuscript. The results in Table 4 show that kernel CDA outperforms the MLP baseline. We also compare with a CDA-initialized MLP. To compare fairly with CDA initialization, since after training CDA has 45 binary models, we set the hidden layer neurons to 45, therefore the MLP architecture is 512 input features -> 45 hidden units -> 10 output neurons. While achieving similar performance to random-initialized MLP on the test set, it has much faster training speed, as shown by the AAE and the training curve comparison in figure shown in the link below.
>
> **Table 4. Test set multiclass prediction performance on SVHN dataset (Using an ImageNet-pretrained ResNet-18 feature extractor followed by a classifier)**
> | Method                         | AUROC       | AUPR        | Fscore      | ACscore     | Acc   | AAE   |
> |--------------------------------|-------------|-------------|-------------|-------------|-------|-------|
> | **CDA initialized linear layer** | **0.79±0.01** | **0.797±0.01** | **0.795±0.01** | **0.748±0.02** | **0.664** | **0.637** |
> | Randomly initialized linear layer | 0.78±0.02 | 0.795±0.009 | 0.789±0.009 | 0.73±0.03 | 0.653 | 0.621 |
> |---|---|---|---|---|---|---|
> | **Gaussian CDA**                 | **0.815±0.02** | **0.816±0.009** | **0.815±0.009** | **0.783±0.03** | **0.683** | - |
> | CDA initialized MLP              | 0.796±0.02 | 0.799±0.01 | 0.798±0.01 | 0.757±0.03 | 0.674 | **0.824** |
> | Randomly initialized MLP         | 0.795±0.02 | 0.798±0.01 | 0.798±0.01 | 0.755±0.03 | 0.666 | 0.797 |
>
> **Hyperparameters:** 150 epochs; BatchSize=128; Init LR=0.005 with decay of 50% every 40 epochs; L2Regulation=1e-4. For nonlinear method tests (the last 3), we use a subset of 24 000 / 99 289 total samples due to time limits for kernel methods.
>
> Figure link: https://drive.google.com/file/d/1uqlygxGL8Q3cCDgAVrSAe3ah52oQpwQI/view?usp=sharing
>
> These experiments demonstrate that CDA is well-performing and flexible: it can (i) act as a strong initializer in neural networks, either for a simple linear layer or a MLP, improving both convergence speed and final performance, and (ii) be extended to kernelized nonlinear variants that outperform standard nonlinear baselines. We have updated this part in the revised manuscript with an Appendix subsection titled “Combining CDA with Neural Network”.

---

### Official Review · Reviewer_QeqZ · 2025-10-29

**Soundness:** 2
**Presentation:** 3
**Contribution:** 2
**Rating:** 4
**Confidence:** 4

**Summary:**

The work proposes a theoretical framework named geometric discriminant analysis (GDA), which includes the family of linear classifiers that can be expressed as function of a centroid discriminant basis (CDB0). A linear classifier named centroid discriminant analysis (CDA) is defined as a special case of GDA under a 2D plane geometric constraint.

**Strengths:**

1.	A geometric theoretical framework for classifiers is proposed.
2.	A scalable linear geometric classifier is designed.

**Weaknesses:**

1.	The authors claimed that the proposed GDA is a framework includes the family of linear classifiers, but in fact only on CDA is designed. The claimed way for this is not suitable.
2.	There are many classifier works, but the authors only compared LDA and SVM, which cannot reflect the sota of the propose method.
3.	In the experiments, only LDA, SVM and LR are used as compared methods, which has no persuasiveness. There are many L
4.	The proposed method uses kernel, so why not add kernel LDA for comparison?
5.	As we all known, linear classifier is hard to discuss nonlinear data. In fact, the distribution of the data is always nonlinear, so how about the generation of the proposed method for nonlinear data?
6.	Minor errors: Where in line 147; Eq. equation 6 in line 177 and similar errors in other places.

**Questions:**

Please see the Weaknesses.

---

> ### Author Response · Authors · 2025-11-28
>
> **1. Reviewer**: The authors claimed that the proposed GDA is a framework includes the family of linear classifiers, but in fact only on CDA is designed. The claimed way for this is not suitable.
>
> **Reply**: We understand the Reviewer concerns. We have now modified the text clarifying more explicitly that: “GDA in the current formulation is a framework that can include only centroid based classifiers, hence methods such as SVM that are based on large margin classification are not included in the GDA framework to explain and design classifiers based on linear centroid-based separability.”
>
> **2 & 3. Reviewer**: There are many classifier works, but the authors only compared LDA and SVM, which cannot reflect the sota of the propose method.
>
> In the experiments, only LDA, SVM and LR are used as compared methods, which has no persuasiveness. There are many L
>
> **Reply**: We thank the reviewer for highlighting the importance of comparing CDA with contemporary linear classifiers to more convincingly demonstrate its advantages in efficiency, predictive performance, and interpretability. In response to this valuable suggestion, we expanded our experimental study by adding a stronger representative linear classifier: We included the binary-version of Partial Least Squares Discriminant Analysis (PLSDA, Ref. [1]), a widely used and competitive linear method especially in chemometrics, genomics, and other high-dimensional applications. In addition, we considered also the inherent multiclass versions of LDA, SVM, and PLSDA, since multiclass is known for its efficiency regarding the number of classes.
>
> The results in Table 1 below show that, for binary classification (AUROC), across 27 real datasets, PLSDA does not outperform CDA. On average, CDA achieves the highest AUROC, followed closely by SVM variants and LDA, while PLSDA shows notably weaker performance. For multiclass prediction (AUROC) in Table 2, in the inherent multiclass setting, only multiclass LDA approaches CDA’s performance, while multiclass SVM and multiclass PLSDA exhibit limited prediction power. This makes sense since these methods are natively designed for binary classification such as the margin in SVM, and their optimization objective becomes less effective when extended to an inherent multiclass setting.
>
> Instead, CDA remains the top-performing method among all linear models evaluated. In addition to performance, CDA maintains a highly favorable computational profile. Its time complexity $O(NM+N\log N)$ makes it particularly efficient for large-scale, high-dimensional data, as demonstrated in our experiments on omics datasets.
>
> Since we have evidence that the inherent multiclass approaches for LDA and SVM performed worse than their one-versus-one scheme, we decided to leave the investigation the multiclass CDA to future study. We have updated the native multiclass comparison in the revised manuscript with an Appendix section titled “Comparison with Native Multiclass Approaches”.
>
> [1] Barker, M., & Rayens, W. (2003). Partial least squares for discrimination. Journal of Chemometrics, 17(3), 166–173.
>
> For full table please see link https://docs.google.com/spreadsheets/d/1CfwdEf0RJRXb4cunZ4g0pk0l9s-zP_jG/edit?usp=sharing
>
> **Table 1. AUROC (Binary classification performance)**
>   | Metric              | CDB0  | CDA      | LDA   | Fast SVM | SVM primal | SVM BO  | SVM SGD | SVM   | LR    | PLSDA |
> |---------------------|-------|----------|-------|----------|------------|---------|---------|-------|-------|-------|
> | Average performance | 0.749 | **0.800** | 0.758 | 0.765    | 0.775      | 0.782 | 0.761   | 0.775 | 0.767 | 0.663 |
>
> **Table 2. AUROC (Multiclass prediction performance)**
>   | Metric              | CDB0 | CDA | LDA | Inherent multiclass LDA | Fast SVM | SVM primal | SVM BO | SVM SGD | SVM | Inherent multiclass SVM | PLSDA | Inherent multiclass PLSDA | LR   |
> |---------------------|------|-----|-----|---------------------------|----------|------------|--------|---------|------|---------------------------|--------|------------------------------|------|
> | Average performance | 0.693 | **0.758** | 0.720 | 0.718 | 0.727 | 0.736 | 0.742 | 0.724 | 0.736 | 0.612 | 0.634 | 0.642 | 0.723 |

---

> ### Author Response · Authors · 2025-11-28
>
> **4. Reviewer**: The proposed method uses kernel, so why not add kernel LDA for comparison?
>
> **Reply**: We thank the reviewer for this insightful comment. Our initial goal in Section 5 was to provide a preliminary validation that CDA is compatible with kernel methods; for this reason, we included kernel SVM as a representative baseline and did not extend the comparison to additional kernelized classifiers.
>
> We agree, however, that kernel LDA is a meaningful and informative baseline. Unlike linear LDA, kernel LDA operates by mapping samples into a high-dimensional feature space and finding projection directions that maximize the ratio of between-class to within-class variance in that space. Given these distinct principles, kernel LDA offers a valuable comparison point and can further highlight the advantages of kernel CDA.
>
> In response to the reviewer’s suggestion, we conducted experiments with kernel LDA and report the results in the table below. Addressing the request of another Reviewer, we also included a new tested dataset - fracturemnist 3d medical image dataset, in which linear CDA performance is low. The results show that kernel CDA outperforms on 2 out of 3 datasets, while the performance gap compared to the best one (nLDA) is not large. Importantly, kernel CDA improves on linear CDA substantially, suggesting the potential of applying kernel methods on linear CDA for complex data. These results have now been incorporated into Table 1 of the revised manuscript.
>
> **Table 3. Test set multiclass performance**
> | Dataset                | Method | AUROC         | AUPR          | Fscore        | ACscore       |
> |------------------------|--------|---------------|---------------|---------------|---------------|
> | SVHN subset (image)    | CDA    | 0.615±0.02    | 0.63±0.02     | 0.619±0.02    | 0.423±0.05    |
> |                        | nCDA   | 0.777±0.01    | 0.782±0.01    | 0.78±0.01     | 0.731±0.02    |
> |                        | SVM    | 0.555±0.01    | 0.568±0.007   | 0.551±0.006   | 0.273±0.05    |
> |                        | nSVM   | 0.736±0.02    | 0.776±0.009   | 0.756±0.008   | 0.654±0.03    |
> |                        | nLDA | **0.786±0.01** | **0.79±0.01** | **0.789±0.01** | **0.743±0.02** |
> | ClinTox (chemical)     | CDA    | 0.567         | 0.561         | 0.56          | 0.351         |
> |                        | nCDA | **0.625**      | **0.627**      | **0.627**      | **0.46**      |
> |                        | SVM    | 0.565         | 0.578         | 0.575         | 0.294         |
> |                        | nSVM   | 0.5           | 0.481         | 0.48          | 0             |
> |                        | nLDA   | 0.605         | 0.612         | 0.611         | 0.409         |
> | Fracture 3d (medical image) | CDA    | 0.518±0.01    | 0.526±0.02    | 0.486±0.02    | 0.279±0.05    |
> |                        | nCDA | **0.625±0.04** | 0.62±0.04 | **0.607±0.04** | **0.577±0.08** |
> |                        | SVM    | 0.576±0.009   | 0.579±0.008   | 0.577±0.008   | 0.505±0.05    |
> |                        | nSVM   | 0.608±0.06    | 0.591±0.07    | 0.586±0.07    | 0.44±0.2      |
> |                        | nLDA   | 0.608±0.03    | **0.626±0.02**    | 0.605±0.02    | 0.491±0.1     |

---

> ### Author Response · Authors · 2025-11-28
>
> **5. Reviewer**: As we all known, linear classifier is hard to discuss nonlinear data. In fact, the distribution of the data is always nonlinear, so how about the generation of the proposed method for nonlinear data?
>
> **Reply**: We thank the reviewer for raising this point.
>
> We addressed the Reviewer’s concern by adding this part of experiments and results in the revised manuscript with a section in Appendix titled “More about Linear CDA Classification”:
>
> “While the core CDA algorithm is efficient, the kernelized version inherits the standard computational bottleneck of kernel matrix construction. This bottleneck is not specific to CDA, but is an inherent limitation of all kernel-based linear classifiers. When any linear method (e.g., LDA, SVM) is kernelized, the dominant cost comes from constructing and storing the N×N kernel matrix, which scales quadratically with the sample size. Thus, the kernelized CDA inherits the standard kernel-matrix bottleneck shared universally by kernel methods, rather than introducing a new one. Although, we provide preliminary results that CDA can be formulated in the nonlinear versions via the kernel trick, we stress that in this paper we focused on linear CDA classification, because linear methods remain widely used in several scientific domains - particularly neuroscience, biomedicine, and social sciences - where explainability, stability, interpretability, and transparent decision boundaries are essential. In many neuroimaging studies, for example, linear classifiers are preferred [1-2] because they provide direct insight into which brain regions contribute to a behavioral or cognitive prediction, making them a standard analytical tool despite the inherent nonlinearity of the data. To further assess the capability of CDA under nonlinear data distributions, we conducted an additional experiment on a real fMRI-behavior prediction task, where the goal is to classify human subjects into four behavioral score categories (fluid intelligence (gF); 4 groups are low gF, low-middle gF, middle-high gF and high gF). Although the underlying neurobiological patterns are highly nonlinear, this setting reflects a realistic domain where linear models are still favored for interpretability and reproducibility [1-2]. The results (Table 4) show that CDA consistently outperforms all other linear classifiers, including LDA, logistic regression, and several SVM variants. This demonstrates that CDA can better capture discriminative structures even when the data distribution is nonlinear, while maintaining the transparency advantages of linear models.”
>
> **Table 4. Test set multiclass prediction performance on fMRI-behavior dataset (N=1076, M=64620, 4 classes)**
> | Method    | AUROC        | AUPR         | Fscore       | ACscore      |
> |-----------|--------------|--------------|--------------|--------------|
> | CDA       | **0.577±0.02** | **0.574±0.02** | **0.571±0.02** | **0.495±0.04** |
> | LDA       | 0.538±0.02   | 0.558±0.03   | 0.536±0.03   | 0.384±0.07   |
> | SVM fast  | 0.534±0.005  | 0.536±0.005  | 0.532±0.007  | 0.4±0.04     |
> | SVM SGD   | 0.544±0.005  | 0.547±0.007  | 0.545±0.007  | 0.424±0.04   |
> | SVM       | 0.538±0.006  | 0.539±0.007  | 0.538±0.007  | 0.419±0.03   |
> | LR fast   | 0.537±0.004  | 0.539±0.005  | 0.536±0.006  | 0.4±0.05     |
>
> [1] Shen, X., Finn, E., Scheinost, D. et al. Using connectome-based predictive modeling to predict individual behavior from brain connectivity. Nat Protoc 12, 506–518 (2017).
>
> [2] Misaki M., Kim Y., Bandettini P.A., Kriegeskorte N. “Comparison of multivariate classifiers and response normalizations for pattern‑information fMRI.” NeuroImage, 2010; 53(1): 103–118.
>
> **6. Reviewer**: Minor errors: Where in line 147; Eq. equation 6 in line 177 and similar errors in other places.
>
> **Reply**: We thank the reviewer for carefully identifying these minor typographical inconsistencies. We have thoroughly reviewed the manuscript and corrected all occurrences of such errors. Specifically, we replaced redundant expressions such as “Eq. equation 6” with the appropriate format “Eq. 6” and revised sentences beginning with “Where” after equations to correctly use “where” in lowercase and connect them properly to the preceding sentences. Similar corrections have been applied throughout the text to ensure stylistic consistency, which are marked in the revised manuscript pdf with blue color.

---

### Official Review · Reviewer_K7aC · 2025-11-02

**Soundness:** 3
**Presentation:** 3
**Contribution:** 3
**Rating:** 6
**Confidence:** 2

**Summary:**

This paper introduces a Geometric Discriminant Analysis (GDA) framework, a unified geometric perspective for linear classifiers based on a Centroid Discriminant Basis (CDB0)—the vector connecting class centroids—adjusted by geometric corrections. Within this framework, the authors propose Centroid Discriminant Analysis (CDA), a linear classifier that iteratively rotates the discriminant in 2D planes using Bayesian optimization to maximize a performance score. CDA achieves quadratic time complexity, outperforming traditional methods like LDA and SVM (which have cubic complexity) in scalability.

Extensive experiments on 27 real-world datasets (images, medical, chemical) show CDA's superior performance, stability, and efficiency. A kernel-based nonlinear extension (nCDA) is also introduced and shown to improve upon linear CDA and kernel SVM on challenging datasets.

**Strengths:**

- computational efficiency of CDA, which achieves quadratic time complexity,
This potentially makes the CDA significantly fastethan traditional cubic-complexity methods like LDA and SVM.
 This is supported by  empirical evidence across 27 diverse real-world datasets, where CDA consistently outperforms established linear classifiers in terms of predictive performance, stability, and training speed, including on a large-scale single-cell dataset

- the method offers  interpretability due to its geometric foundation and performance-dependent learning process,

- the method can be extended to a kernelized nonlinear version (nCDA) demonstrates flexibility and improved capability on complex data, broadening its applicability.

**Weaknesses:**

-  The GDA framework is presented as a conceptual model and unifying perspective but does not seem to have rigorous theoretical guarantees?

- The training process relies on several heuristic components, including Bayesian optimization for rotations and a sample reweighting strategy, which are difficult to follow and are maybe not robust?

-  while the core CDA algorithm is efficient, the kernelized version inherits the standard computational bottleneck of kernel matrix construction, and the overall approach remains fundamentally designed for binary classification, relying on external methods  for multiclass problems.

**Questions:**

- how does the training time of CDA scale for a multiclass problem ?

- The paper uses flattened raw pixels and simple tokenization. Do you believe CDA's performance is currently limited by this pre-processing? Have you tested it on more sophisticated, learned features (e.g., from an autoencoder) to see if it can serve as a powerful final-layer classifier?

- In high-dimensional spaces, the 2D planes for rotation are constructed heuristically. How stable is the Bayesian Optimization process  especially in the initial rotations where the number of samples is low?

- Can you show that the method is not loosing too much performance compared to LDA if the data is (exactly) Gaussian?

- what about the performance on elliptic distributions? Maybe it could be interesting to show how this compare with "robust" methods build for elliptic distributions.

---

> ### Author Response · Authors · 2025-11-28
>
> **1. Reviewer**: The GDA framework is presented as a conceptual model and unifying perspective but does not seem to have rigorous theoretical guarantees?
>
> **Reply**: We thank the reviewer for this important point, because it allows us to contextualize better the GDA framwork. We agree that GDA is presented as a conceptual and unifying theoretical framework. Its purpose is to provide a general perspective that can encompass a wide family of linear and kernel classifiers. The framework makes the design of new classifiers straightforward.
>
> We address the reviewers concerns by clarifying in the revised text of the article that: “Since GDA is a framework rather than a specific algorithm, it does not itself provide global theoretical guarantees on performance or convergence. Instead, such guarantees must be established for each concrete classifier instantiated within the framework. Therefore, in Appendix Subsection “CDA Convergence Proof” we provide a section on CDA convergence proof, where we provide a formal mathematical proposition showing that, under the defined fitness function, CDA is guaranteed to converge. This convergence relies solely on the monotonicity and boundedness of the objective sequence and therefore holds independently of the specific 1D optimizer employed (e.g., BO, Brent’s method, etc)”. The proof has been added as a section in Appendix titled “CDA Convergence Proof”.
>
> *Proposition: Let* $w_{\rm CDB1}^{(k)} \in {\rm R}^{M}$ *be CDB1 at iteration k.
> In each CDA iteration the* $w_{\rm CDB1}^{(k)}$ *determines the CDB2*
> $w_{\rm CDB2}^{(k)}$ *by the CDA algorithm with sample weights update
> strategy* $w_{\rm CDB2}^{(k)} = g(X,w_{\rm CDB1}^{(k)})$*. Each discriminant is
> associated with a performance metric* $ps = f(\rm w_{CDB})$*, and this
> metric is bounded above up to 1. At current iteration the optimal
> vector, is found on the spanned 2d-plane by CDB1 and CDB2 by*
> $w_{\rm CDB1}^{(k + 1)} = \lambda_{1}^{\*}w_{\rm CDB1}^{(k)} + \lambda_{2}^{\*}w_{\rm CDB2}^{(k)}\ $
> *where*
> $\lambda_{1}^{\*},\lambda_{2}^{\*} = \arg\max_{\lambda_{1},\lambda_{2} \in {\rm R}}f\left( \lambda_{1}w_{\rm CDB1}^{(k)} + \lambda_{2}w_{\rm CDB2}^{(k)} \right)$*,
> which is then used as new CDB1 at iteration k+1. Since*
> $w_{\rm CDB1}^{(k)} \in \{\lambda_{1}w_{\rm CDB1}^{(k)} + \lambda_{2}w_{\rm CDB2}^{(k)}\}$
> *as a special case* $\lambda_{1} = 1$ *and* $\lambda_{2} = 0$*, this
> indicates that* $ps_{\rm CDB1}^{(k+1)} \geq ps_{\rm CDB1}^{(k)}$ *. Then
> the sequence* $ps_{\rm CDB1}^{(k)} = f(w_{\rm CDB1}^{(k)})$ *is non-decreasing
> and bounded above, and therefore converges to a finite limit*
> ${ps}^{\*}$*.*

---

> ### Author Response · Authors · 2025-11-28
>
> **2. Reviewer**: The training process relies on several heuristic components, including Bayesian optimization for rotations and a sample reweighting strategy, which are difficult to follow and are maybe not robust?
>
> **Reply**: We appreciate the Reviewer’s insightful comment regarding the heuristic components in the CDA training process. To clarify our design choice and evaluate robustness, we conducted a meticulous study comparing BO with five alternative 1D optimizers commonly used for black-box functions: 1) Random search 2) grid search 3) Fibonacci search 4) Brent’s method and 5) ternary search.
>
> We evaluated all optimizers in terms of (i) final CDA performance (ii) average running time (iii) average number of CDA iterations and (iv) average runtime per iteration, reported in the table below. Importantly, across all six optimizers, CDA consistently converges to a similar level of performance, demonstrating that the CDA algorithm is well-posed and stable, regardless of the choice of 1D optimizer. For average runtime per iteration in (iv), as expected, random search is the fastest due to zero structural overhead, whereas BO is the slowest due to the cost of maintaining its surrogate model. In CDA we applied an early-stop strategy by checking if the training performance across iteration is stable. We found that the results of (ii) and (iii) show that Fibonacci search help to stabilize the training process, with the least average running time and average number of CDA iterations.
>
> Based on our findings, we make the following recommendations. For maximum speed, Fibonacci search is the preferred choice, and BO is a strong second option. Be aware that BO seems to prefer log-transform of the data. For large-scale datasets, Fibonacci search remains the most efficient and reliable method.
>
> These results confirm that BO is not required for CDA to function correctly, but it remains a competitive choice under certain data conditions. We have added these comparisons and guidelines to the revised manuscript to clarify the rationale behind the optimizer selection.
>
> Please see full tables in:
>
> https://docs.google.com/spreadsheets/d/1qFeURphzLeBOF_YpDWCbos2ZlFz1uwep/edit?usp=sharing (original data)
>
> https://docs.google.com/spreadsheets/d/1cg7-EntqAsJRocYdkQtAlpKmtERmGmzH/edit?usp=sharing (log-transformed data)
>
> **Table 1. Comparison of Optimization Methods on Original and Log-Transformed Data**
> | Category | Metric | BO | Random search | Grid search | Fibonacci search | Brent's method | Ternary search |
> |----------|---------|------|----------------|--------------|-------------------|-----------------|----------------|
> | Original | Avg. performance (AUROC) | 0.758 | 0.755 | 0.751 | 0.757 | 0.753 | **0.760** |
> |          | Avg. runtime (s) | 4.681 | 4.339 | 4.137 | **2.473** | 2.838 | 3.230 |
> |          | Avg. number of rotations | 36.278 | 46.330 | 43.811 | 29.689 | **29.252** | 35.441 |
> |          | Time per iteration (s) (Avg. runtime / Avg. number of rotations) | 0.136 | **0.096** | 0.099 | 0.099 | 0.107 | 0.114 |
> | Log-transformed | Avg. performance (AUROC) | **0.764** | 0.745 | 0.761 | 0.755 | 0.754 | 0.758 |
> |                | Avg. runtime (s) | 4.693 | 4.313 | 4.003 | **2.503** | 2.546 | 3.355 |
> |                | Avg. number of rotations | 35.796 | 46.330 | 43.507 | 30.244 | **26.178** | 35.759 |
> |                | Time per iteration (s) (Avg. runtime / Avg. number of rotations) | 0.134 | **0.095** | 0.098 | 0.098 | 0.113 | 0.108 |
>
> Moreover, we applied CDA-Fibonacci for the large-scale mouse brain single-cell dataset to classifier cell types up to 1 million samples, and record the performance and running time. See results in the figure link below. Indeed, Fibonacci further enhances CDA efficiency for large-scale application. In contrast, SVM fast requires significantly longer running time.
>
> See figure with this link: https://drive.google.com/file/d/1dCq762QTiZaY2iX1VehVm6VvOEO7ZFgt/view?usp=sharing
>
> We updated these experiments comparing different optimizers with a section in Appendix titled “Optimizer Comparison for CDA Rotation”.

---

> ### Author Response · Authors · 2025-11-28
>
> **3. Reviewer**: while the core CDA algorithm is efficient, the kernelized version inherits the standard computational bottleneck of kernel matrix construction.
>
> **Reply**: We thank the reviewer for raising this point.
>
> We addressed the Reviewer’s concern by adding this part of experiments and results in the revised manuscript with a section in Appendix titled “More about Linear CDA Classification”:
>
> “While the core CDA algorithm is efficient, the kernelized version inherits the standard computational bottleneck of kernel matrix construction. This bottleneck is not specific to CDA, but is an inherent limitation of all kernel-based linear classifiers. When any linear method (e.g., LDA, SVM) is kernelized, the dominant cost comes from constructing and storing the N×N kernel matrix, which scales quadratically with the sample size. Thus, the kernelized CDA inherits the standard kernel-matrix bottleneck shared universally by kernel methods, rather than introducing a new one. Although, we provide preliminary results that CDA can be formulated in the nonlinear versions via the kernel trick, we stress that in this paper we focused on linear CDA classification, because linear methods remain widely used in several scientific domains - particularly neuroscience, biomedicine, and social sciences - where explainability, stability, interpretability, and transparent decision boundaries are essential. In many neuroimaging studies, for example, linear classifiers are preferred [1-2] because they provide direct insight into which brain regions contribute to a behavioral or cognitive prediction, making them a standard analytical tool despite the inherent nonlinearity of the data. To further assess the capability of CDA under nonlinear data distributions, we conducted an additional experiment on a real fMRI-behavior prediction task, where the goal is to classify human subjects into four behavioral score categories (fluid intelligence (gF); 4 groups are low gF, low-middle gF, middle-high gF and high gF). Although the underlying neurobiological patterns are highly nonlinear, this setting reflects a realistic domain where linear models are still favored for interpretability and reproducibility [1-2]. The results (Table 2) show that CDA consistently outperforms all other linear classifiers, including LDA, logistic regression, and several SVM variants. This demonstrates that CDA can better capture discriminative structures even when the data distribution is nonlinear, while maintaining the transparency advantages of linear models.”
>
> **Table 2. Test set multiclass prediction performance on fMRI-behavior dataset (N=1076, M=64620, 4 classes)**
> | Method    | AUROC        | AUPR         | Fscore       | ACscore      |
> |-----------|--------------|--------------|--------------|--------------|
> | CDA       | **0.577±0.02** | **0.574±0.02** | **0.571±0.02** | **0.495±0.04** |
> | LDA       | 0.538±0.02   | 0.558±0.03   | 0.536±0.03   | 0.384±0.07   |
> | SVM fast  | 0.534±0.005  | 0.536±0.005  | 0.532±0.007  | 0.4±0.04     |
> | SVM SGD   | 0.544±0.005  | 0.547±0.007  | 0.545±0.007  | 0.424±0.04   |
> | SVM       | 0.538±0.006  | 0.539±0.007  | 0.538±0.007  | 0.419±0.03   |
> | LR fast   | 0.537±0.004  | 0.539±0.005  | 0.536±0.006  | 0.4±0.05     |
>
> [1] Shen, X., Finn, E., Scheinost, D. et al. Using connectome-based predictive modeling to predict individual behavior from brain connectivity. Nat Protoc 12, 506–518 (2017).
>
> [2] Misaki M., Kim Y., Bandettini P.A., Kriegeskorte N. “Comparison of multivariate classifiers and response normalizations for pattern‑information fMRI.” NeuroImage, 2010; 53(1): 103–118.

---

> ### Author Response · Authors · 2025-11-28
>
> **4. Reviewer**: The overall approach remains fundamentally designed for binary classification, relying on external methods for multiclass problems.
>
> how does the training time of CDA scale for a multiclass problem ?
>
> **Reply**: The training time complexity of CDA for a multiclass problem is on the order of $C^2*O(NM+N\log N)$, where $C$ is the number of classes, $N$ is the number of samples per class, and M is the feature dimension. However, Since $C$ is typically much smaller than $NM$, it is neglible, and the overall complexity effectively becomes $O(NM+N\log N)$, similar to the binary case.
>
> Considering the reviewer’s question, we think that exploring an inherent multiclass extension of CDA is an important direction, especially given that other classifiers such as LDA and SVM have their inherent multiclass approach. In the cases where $C$ is large and not negligible, inherent multiclass approaches offer attractive lower computational complexity with respect to the number of classes ($C$ versus $C(C−1)/2$ discriminants in pairwise schemes).
>
> In the current study, we evaluated only binary versions of LDA, SVM, and LR, and relied on the external ECOC framework to handle multiclass prediction. To address the reviewer’s point, we additionally tested inherent multiclass LDA, SVM, and PLS-DA. Our results (Table 3) show that although these inherent multiclass methods indeed achieve faster training times, their predictive performance is consistently lower than that obtained by their binary counterparts combined with an external multiclass strategy. This makes sense since these methods are natively designed for binary classification such as the margin in SVM, and their optimization objective becomes less effective when extended to an inherent multiclass setting.
>
> Since we have evidence that the inherent multiclass approaches for LDA and SVM performed worse than their one-versus-one scheme, we decided to leave the investigation the multiclass CDA to future study. We have updated the native multiclass comparison in the revised manuscript with an Appendix section titled “Comparison with Native Multiclass Approaches”.
>
> For full table please see link https://docs.google.com/spreadsheets/d/1CfwdEf0RJRXb4cunZ4g0pk0l9s-zP_jG/edit?usp=sharing
>
> **Table 3. Average multiclass prediction performance (AUROC) across different classifiers**
> | Metric              | CDB0 | CDA | LDA | Inherent multiclass LDA | Fast SVM | SVM primal | SVM BO | SVM SGD | SVM | Inherent multiclass SVM | PLSDA | Inherent multiclass PLSDA | LR   |
> |---------------------|------|-----|-----|---------------------------|----------|------------|--------|---------|------|---------------------------|--------|------------------------------|------|
> | Average performance | 0.693 | **0.758** | 0.720 | 0.718 | 0.727 | 0.736 | 0.742 | 0.724 | 0.736 | 0.612 | 0.634 | 0.642 | 0.723 |

---

> ### Author Response · Authors · 2025-11-28
>
> **5. Reviewer**: The paper uses flattened raw pixels and simple tokenization. Do you believe CDA's performance is currently limited by this pre-processing? Have you tested it on more sophisticated, learned features (e.g., from an autoencoder) to see if it can serve as a powerful final-layer classifier?
>
> **Reply**: We thank the reviewer for this insightful question. To test this, we conducted two experiments using more sophisticated, learned features:
>
> 1)	Using CDA to initialize the final linear layer of a ResNet-18: We take an ImageNet-pretrained ResNet-18 and perform transfer learning on SVHN. Instead of using a randomly initialized final linear layer, we initialize it using class-wise averaged CDA weights. The results in the table below show clear improvements over the baseline: higher test accuracy and substantially faster training, as measured by area-across-epochs (AAE).
>
> 2) Kernel CDA vs. an MLP: We evaluate kernel CDA (Gaussian) as a nonlinear model and compare it to a single-hidden-layer MLP. Since kernel CDA involves computing a kernel matrix in quadratic to the number of samples, and due to the time limit we are not able to run the full dataset, we use the same subset of data (24k/99k) that we used in our first manuscript. The results in the table below show that kernel CDA outperforms the MLP baseline. We also compare with a CDA-initialized MLP. To compare fairly with CDA initialization, since after training CDA has 45 binary models, we set the hidden layer neurons to 45, therefore the MLP architecture is 512 input features -> 45 hidden units -> 10 output neurons. While achieving similar performance to random-initialized MLP on the test set, it has much faster training speed, as shown by the AAE and the training curve comparison in figure shown in the link below.
>
> **Table 4. Test set multiclass prediction performance on SVHN dataset (Using an ImageNet-pretrained ResNet-18 feature extractor followed by a classifier)**
> | Method                         | AUROC       | AUPR        | Fscore      | ACscore     | Acc   | AAE   |
> |--------------------------------|-------------|-------------|-------------|-------------|-------|-------|
> | **CDA initialized linear layer** | **0.79±0.01** | **0.797±0.01** | **0.795±0.01** | **0.748±0.02** | **0.664** | **0.637** |
> | Randomly initialized linear layer | 0.78±0.02 | 0.795±0.009 | 0.789±0.009 | 0.73±0.03 | 0.653 | 0.621 |
> |---|---|---|---|---|---|---|
> | **Gaussian CDA**                 | **0.815±0.02** | **0.816±0.009** | **0.815±0.009** | **0.783±0.03** | **0.683** | - |
> | CDA initialized MLP              | 0.796±0.02 | 0.799±0.01 | 0.798±0.01 | 0.757±0.03 | 0.674 | **0.824** |
> | Randomly initialized MLP         | 0.795±0.02 | 0.798±0.01 | 0.798±0.01 | 0.755±0.03 | 0.666 | 0.797 |
>
> **Hyperparameters:** 150 epochs; BatchSize=128; Init LR=0.005 with decay of 50% every 40 epochs; L2Regulation=1e-4. For nonlinear method tests (the last 3), we use a subset of 24 000 / 99 289 total samples due to time limits for kernel methods.
>
> Figure link:
> https://drive.google.com/file/d/1uqlygxGL8Q3cCDgAVrSAe3ah52oQpwQI/view?usp=sharing
>
> These experiments demonstrate that CDA is well-performing and flexible: it can (i) act as a strong initializer in neural networks, either for a simple linear layer or a MLP, improving both convergence speed and final performance, and (ii) be extended to kernelized nonlinear variants that outperform standard nonlinear baselines. We have updated this part in the revised manuscript with an Appendix subsection titled “Combining CDA with Neural Network”.

---

> ### Author Response · Authors · 2025-11-28
>
> **6. Reviewer**: In high-dimensional spaces, the 2D planes for rotation are constructed heuristically. How stable is the Bayesian Optimization process especially in the initial rotations where the number of samples is low?
>
> **Reply**: We thank the reviewer for this important question. To comprehensively demonstrate the stability and convergence of CDA following the BO-driven rotations, we assess the process from three complementary aspects.
>
>  + To assess the stability of the BO-driven rotation process when only a few initial samples are available, we compared two BO schemes that differ only in the number of initial samples (all other BO hyperparameters and total BO budget were kept the same). In the first BO sampling scheme, the number of sampling is set to min(10, max(4, Iter)) that gradually increase the number of sampling from 4 to 10 according to the current CDA iteration. In the second BO sampling scheme, the number of sampling is set to always 10. The goal is to test whether the early-stage sampling scarcity destabilizes convergence or substantially degrades multiclass performance.
>
> The results in Table 5 show that, across a wide set of images, medical and chemical datasets, performance is essentially unchanged between the two schemes (average AUROC 0.758 vs 0.754). The absolute average difference is 0.004, which is negligible for our tasks and indicates that BO-driven rotations are stable even when the initial sampling budget is small. It is even beneficial to use fewer sampling at initial CDA iterations. We update this experiment in the revised manuscript in Appendix.
>
> See the full table from the link: https://docs.google.com/spreadsheets/d/19VCIpJv7djU4t19r5Ti10Zcqy7UEAal4/edit?usp=sharing
>
> **Table 5. AUROC (Multiclass prediction performance)**
> | Metric              | BO (few initial sampling) | BO control (high initial sampling) |
> |---------------------|--------------------------|----------------------------------|
> | Average performance | **0.758**                | 0.754                            |
>
> + We are sorry that the previous Figure 3 did not clearly illustrate the CDA (BO version) convergence behavior with respect to the maximum number of rotations. We have now updated the figure (please see figure link below) to improve clarity. In the revised version, we investigated how CDA’s stopping iteration relates to classification performance and found two regimes. For tasks that converge within 50 iterations, there is a clear negative correlation (R = –0.48): harder datasets require more iterations, indicating that allowing up to 50 iterations is important for reliable convergence. Beyond 50 iterations, this relationship largely disappears (R = 0.184), showing that additional iterations do not meaningfully affect convergence. Consistently, when comparing performance distributions obtained with 50 vs. 150 maximum iterations, the results were statistically indistinguishable (Wilcoxon p = 0.398). This confirms that increasing the iteration limit beyond 50 yields no systematic performance gains. We have updated the Figure 3 in the revised manuscript. We also provide the same convergence analysis for CDA-Fibonacci with an Appendix subsection titled “CDA-Fibonacci Convergence”.
>
> Figure link: https://drive.google.com/file/d/1T5uG9u-WA7EtJXQ9BeFs3eQ5QXVFiJ7q/view?usp=sharing
>
> + We investigated the necessity of using Bayesian Optimization (BO) for a one-dimensional search, and tested other 5 different optimizers for single-parameter search apart from BO. Please see the replies in Point 2 of the previous comments above.
>
> We updated these experiments comparing different optimizers with a section in Appendix titled “Optimizer Comparison for CDA Rotation”.

---

> ### Author Response · Authors · 2025-11-28
>
> **7. Reviewer**: Can you show that the method is not loosing too much performance compared to LDA if the data is (exactly) Gaussian?
>
> what about the performance on elliptic distributions? Maybe it could be interesting to show how this compare with "robust" methods build for elliptic distributions.
>
> **Reply**: We thank the reviewer for these valuable suggestions. To address the two questions, we performed controlled simulations under three conditions: (1) Gaussian data, (2) elliptic data, and (3) log-transformed elliptic data. Each simulation consisted of 10,000 samples with 10 features, and the covariance structure was specified using a Toeplitz matrix. The group center distance was gradually increased, and performance was evaluated. We included SVM in the comparison, since its margin-based principle can provide some robustness to outliers.
>
> The results in the figure link below show that for Gaussian data, CDA performs comparably to LDA and SVM, demonstrating that it does not lose significant predictive power under ideal Gaussian conditions. For elliptic distributions, CDA is affected by the presence of outliers, as expected. However, applying a log-transform to the features effectively mitigates the bias introduced by extreme values. After log-transforming the elliptic data, CDA’s performance improves substantially. Interestingly, this preprocessing benefits CDA more than LDA or SVM.
>
> The results for synthetic data are shown in the google drive weblink https://drive.google.com/file/d/1eDVqt0XifI_pNCJd7wsLRu14SXe7z_8o/view?usp=sharing
>
> We further tested log-transform on 27 real datasets with the results shown in Table 6. For CDA we can see that both the winning rate increased from 0.519 to 0.741 and an increase in average performance; while LDA increases winning rate, the average performance drops; for SVM, log-transform leads to underperformance. These results suggest log-transform is adequate for estimating CDA centroids by promoting variance stabilization and outlier control, and CDA can indeed benefit more than other methods for the log-transform to enhance its performance.
>
> Complete table please see weblink: https://docs.google.com/spreadsheets/d/1mR1Yk7_tWpaFRHtKKNLbI8rmRTvgsP6E/edit?usp=sharing
>
> **Table 6. AUROC (Binary prediction performance)**
> | Metric              | CDA Original | CDA Log-transformed | \| | LDA Original | LDA Log-transformed | \| | SVM fast Original | SVM fast Log-transformed |
> |---------------------|--------------|-------------------|---|--------------|-------------------|---|-----------------|-------------------------|
> | Average performance | 0.800        | **0.805**         | \| | **0.758**    | 0.742             | \| | **0.765**       | 0.757                   |
> | Winning rate        | 0.519        | **0.741**         | \| | 0.481        | **0.688**         | \| | **0.704**       | 0.407                   |                   |                   |                   |
>
> Based on these findings, we recommend incorporating a log-transform preprocessing step when applying CDA to data with heavy-tailed or elliptic distributions. We have updated the corresponding section in the revised manuscript to include this recommendation, see Appendix section “Log-transforming the data for CDA to Handle Outliers”.

---

### Author Response · Authors · 2025-12-03
**General Comments (1/2)**

We sincerely thank all Reviewers for their valuable time and insightful feedback, which has been extremely helpful in further improving the quality and clarity of our paper.

We are grateful that the Reviewers recognized:

1.  The **novelty and originality of the Geometric Discriminant Analysis (GDA) framework** and its theoretical foundations, the mathematical derivation of which is well-executed (Reviewers 7WhT, PuwL).
2.  That the **experimental evaluation** on centroid discriminant analysis (CDA) **is comprehensive, convincing, and competitive** (Reviewers 7WhT, PuwL).
3.  CDA has **high efficiency and scalability** (all Reviewers). We are also pleased that CDA's runtime and scalability was acknowledged as **significant** (Reviewer 7WhT).
4.  **Extension of CDA to kernel CDA** to handle complex data better (Reviewers K7aC, PuwL).
5.  CDA has **good interpretability** (Reviewer K7aC).

To provide greater clarity on the revisions made and the additional experiments conducted to address reviewers' questions, we summarize the key modifications and new results below.

**Additional Experiments:**

*   We conducted **robustness tests of CDA under data with outliers,** which cause **poorly estimated centroids**. Introducing tests on synthetic ground truth multidimensional data with Gaussian (absence of outliers) and elliptic (presence of outliers) distribution. We offered solid results that by stabilizing the variance the log-transform preprocessing improves CDA performance significantly for outlier-prone distributions such as elliptic, because it improves the estimation of the centroids. The fact that log-transform is an effective preprocessing for improving the estimation of the centroids in CDA initialization was furthermore confirmed on the real datasets, where log-transform induced a marked CDA classification improvement. We added an appendix section "Log-transforming the data for CDA to Handle Outliers". (Reviewers K7aC, PuwL)

*   We compared **alternative 1D optimizers for CDA rotation** (Bayesian Optimization, Fibonacci, grid, random, ternary, Brent's method), providing guidance for practical usage and demonstrating that CDA's performance is robust across optimizers. We discovered that Bayesian optimization still retains the best performance among the different methods, and the Fibonacci search offers the best trade-off between speed and performance for very large datasets. We added an appendix section "Optimizer Comparison for CDA Rotation" (Reviewer 7WhT).

*   We offered empirical evidence of **Bayesian Optimization stability with fewer samplings in initial iterations**. We added an appendix section "Stability of BO in CDA with Initial Few Sampling" (Reviewer K7aC). Updated Fig. 1f-g with more efficient CDA-Fibonacci on large-scale mouse-brain single cell data.

*   We compared with **another SOTA linear classifier PLS-DA**, and extended evaluations in multiclass scenarios, comparing binary-based CDA with **inherent multiclass approaches for LDA, SVM, and PLS-DA**, showing that one-vs-one strategies remain more effective in practice than inherent multiclass. We added an appendix section "Comparison with Native Multiclass Approaches" (Reviewers QeqZ, PuwL).

*   Performed experiments demonstrating the use of **CDA to initialize neural network layers** to achieve faster training speed and better performance. Added an appendix subsection "Combining CDA with Neural Network" (Reviewer K7aC).

*   We compared **against kernel LDA** in kernel experiments, and **included an additional medical image dataset** fractureMNIST-3D, demonstrating better performance ranking of kernel CDA with respect to other methods. We updated Table 1 in the paper. (Reviewers QeqZ, 7WhT).

*   We examined the **impact of intra-class multimodality** (GTSRB dataset), showing that CDA's adaptive discriminant corrections outperform pure centroid-based methods in case of datasets with intra-class multimodality, being also the only centroid-based method that is competitive with SVM in this setup. We added an appendix section "Performance of centroid-based methods in datasets with multiple subclasses" (Reviewer PuwL).


**Clarifications and Theoretical Explanations:**

*   We made a **mathematical proof that CDA converges** with iterations, independent from the fitness function used. Added an appendix section "CDA Convergence Proof" (Reviewer K7aC).

*   We clarified the **role of correction terms in Eq. 10** by revising the text, better explaining that LDA corresponds to the first-order base term (CDB0) plus a second-order variance-based correction, while higher-order terms are zero (Reviewer PuwL).

(See the next comment)

---

### Author Response · Authors · 2025-12-03
**(Continued) General Comments (2/2)**

(Continued)

**Clarifications and Theoretical Explanations:**

*   We provided a **theoretical explanation of the discriminant bias**, describing how it can be computed post-hoc once the discriminant direction is fixed, with options including midpoint threshold, AUROC maximization, or empirical risk minimization. We added an appendix section "Explanation of the Discriminant Bias in the GDA Framework" (Reviewer PuwL).

*   We clarified that, despite providing preliminary and promising results on the applicability of CDA also in the nonlinear kernel case, **linear CDA remains the focus** of this paper. We provided detailed explanations on the real scenario (practical) reason to focus this study on linear classification. To this aim, we introduced a test on a new neuroscience case (fMRI-behavior datasets) that represents one of the fields that prefers solutions based on linear methods, because of the necessity to provide direct interpretability and explainability of the prediction results. We added an appendix section "More about Linear CDA Classification" (Reviewers K7aC, 7WhT, PuwL).

*   We added a further **discussion in the limitations** about having proven GDA's framework as useful to explain LDA as a centroid-based method, and to invent CDA as a new advanced centroid-based method. The limitation to address in future work is to adopt GDA's framework for inspiring other new centroid-based methods (Reviewers K7aC, QeqZ, 7WhT), which adopt diverse rationales and strategies for designing the correction terms of the GDA equation.


These changes, we believe, will comprehensively address the concerns and suggestions raised by the Reviewers, thereby enriching the overall quality and impact of our research.

---

### Meta-Review · Area_Chair_4xBe · 2026-01-07

**Summary:**

This work received 4, 4, 6, 6 scores, i.e. two borderline accept and two borderline reject recommendations from reviewers. The author rebuttals have quite satisfactorily addressed most of the reviewer concerns on e.g. the training time, flattened raw pixels, limited comparison to LDA/SVM, the initialization, extension to multi-classes. This work offers an unified geometric perspective for linear classifiers, GDA and demonstrates by extensive experiments on 27 datasets. ACs find this work interesting to share with the community and recommend acceptance.

**Reviewer Concerns:**

The same as above.

**Reviewer Scores:**

This work received 4, 4, 6, 6 scores, and the rebuttals have successfully addressed the issues.

---

### Decision · Program_Chairs · 2026-01-26

Accept (Poster)